# Active Continual Learning with Metaplastic Binary Bayesian Neural Networks

**Kellian Cottart** [1]  **Théo Ballet** [1]  **Djohan Bonnet** [1,2]  **Damien Querlioz** [1]

## Abstract

Always-on edge systems must keep learning as conditions change under tight compute budgets and must detect unreliable predictions. Bayesian binary neural networks are attractive in this setting, but mean-field Bernoulli posteriors can saturate on long non-stationary streams, wiping out epistemic uncertainty and freezing plasticity. We propose BiMU, derived from a bounded-memory variational objective that balances stability, plasticity, and forgetting. BiMU combines a data term with controlled relaxation toward the prior and an uncertainty-dependent step size that prevents saturation and sustains informative uncertainty. This non-degenerate posterior enables fully online, buffer-free active querying via Monte Carlo disagreement, reducing label queries and backpropagation updates under imbalance. BiMU sustains learning and strong OOD detection on 1000-tasks Permuted-MNIST, and on OpenLORIS-Object achieves up to 32× label/update savings at matched accuracy under class imbalance and feature compression.

## 1. Introduction

Always-on edge systems face a core tension: they must run inference continuously under tight energy budgets, yet still adapt online as conditions drift (new users, sensor aging, changing environments) and detect when predictions are unreliable (out-of-distribution inputs, rare events). Binary neural networks (BiNNs) (Hubara et al., 2016) are a natural fit: constraining activations and weights to $\{-1, +1\}$ reduces memory traffic and replaces multiply-accumulate operations with cheap bitwise arithmetic, which is critical when inference dominates an always-on device's lifetime compute. Bayesian BiNNs (Meng et al., 2020; Khan & Rue, 2023)

are even more attractive because they quantify epistemic uncertainty beyond point predictions, enabling reliability monitoring and OOD detection in safety- or mission-critical deployments.

However, Bayesian BiNNs trained with mean-field Bernoulli posteriors face a critical failure mode on long, non-stationary streams: the Bernoulli parameters saturate toward $0$ or $1$ as evidence accumulates. Equivalently, the natural parameters grow in magnitude, posterior samples become nearly deterministic, epistemic uncertainty collapses, and weight sign changes become exceedingly unlikely. This *posterior saturation* (often observed as "frozen synapses") has been reported in continual-learning practice for binary networks (Laborieux et al., 2021) and prevents sustained online adaptation. At the edge, this issue is compounded by two practical constraints: weight updates are expensive, and data streams are strongly imbalanced, so most samples are redundant while rare events and distribution shifts carry most of the learning signal.

In this work, we introduce **Binary Metaplasticity from Uncertainty (BiMU)**, a Bayesian continual-learning rule for mean-field Bernoulli synapses that prevents posterior degeneracy while maintaining usable epistemic uncertainty over long streams. BiMU is derived from the bounded-memory Bayesian learning-and-forgetting objective of (Bonnet et al., 2025), and we specialize it here to Bernoulli synapses to avoid saturation in binary posteriors. The update combines (i) a data-driven gradient term with (ii) an uncertainty-gated relaxation toward the prior that counteracts saturation, together with (iii) a bounded, state-dependent (metaplastic) step size that stabilizes consolidated synapses while keeping uncertain ones plastic. This yields a fully online, buffer-free Bayesian update for continual learning without task-boundary signals.

Preserving epistemic uncertainty is not only diagnostic: it enables fully online active continual learning. BiMU supports cheap Monte Carlo sampling of binary weights, and we use a one-pass threshold rule: an incoming example is labeled (and triggers backpropagation) only if a disagreement score exceeds a fixed threshold. This directly reduces label requests and parameter updates and is particularly effective under class imbalance, where most samples are uninformative, but rare events are critical. More broadly, BiMU

[1]Université Paris-Saclay, CNRS, Centre de Nanosciences et de Nanotechnologies, Palaiseau, France [2]Forschungszentrum Jülich, Germany. Correspondence to: Damien Querlioz <damien.querlioz@universite-paris-saclay.fr>.

*Proceedings of the 43rd International Conference on Machine Learning*, Seoul, South Korea. PMLR 306, 2026. Copyright 2026 by the author(s).

treats epistemic uncertainty as a finite resource that must be preserved to support future adaptation.

We evaluate BiMU in three regimes: (i) 1000-tasks Permuted-MNIST to stress long-horizon non-stationarity and OOD detection; (ii) OpenLORIS-Object continual recognition under distribution shifts with a frozen ImageNet feature extractor to isolate online adaptation under constrained trainable capacity; and (iii) imbalanced streaming scenarios with online querying to quantify label/update efficiency, including per-class effects.

**Contributions.**

- We derive BiMU: a bounded-memory variational update for mean-field Bernoulli synapses (Eq. (6)) with uncertainty-gated relaxation and a bounded metaplastic step size (Eq. (7)) that prevents posterior saturation.

- We couple BiMU with one-pass Monte Carlo disagreement to perform buffer-free online querying, so only informative samples are labeled and updated, reducing annotation and backprop cost under imbalance.

- We demonstrate sustained learning and useful uncertainty (OOD detection and querying) on 1000-tasks Permuted-MNIST, and OpenLORIS-Object with frozen/compressed features, achieving $32\times$ fewer label/update steps at matched accuracy.

## 2. Related work

### 2.1. Bayesian inference, uncertainty, and continual learning

Bayesian neural networks (BNNs) treat weights as random variables and quantify epistemic uncertainty through a posterior over parameters. In practice, BNNs are commonly trained with variational inference (VI), approximating the posterior with a tractable distribution $q(\boldsymbol{\omega})$ by minimizing a KL divergence or maximizing an ELBO (Jordan et al., 1999; Graves, 2011; Kingma et al., 2015; Blundell et al., 2015). This provides uncertainty estimates that are useful for reliability monitoring and OOD detection (Kendall & Gal, 2017; Gal et al., 2017).

Continual learning (CL) considers sequentially arriving data where past samples are typically unavailable, so naïve fine-tuning leads to catastrophic forgetting. A classical Bayesian approach reuses the previous approximate posterior as the next prior, yielding online/streaming VI updates (Broderick et al., 2013; Nguyen et al., 2018; Zeno et al., 2021). Regularization-based CL methods can also be interpreted through a Bayesian lens: Elastic Weight Consolidation (EWC) constrains important parameters via a Fisher-based quadratic penalty (Pascanu & Bengio, 2014; Kirkpatrick et al., 2017), and Synaptic Intelligence (SI)

accumulates parameter-importance scores online (Zenke et al., 2017). Online variants (e.g., Online EWC) introduce discounting to limit unbounded accumulation of importance (Schwarz et al., 2018). However, over very long horizons these methods can still become increasingly rigid as evidence/importance accumulates and plasticity decreases. Moreover, many such approaches require storing additional per-parameter state (e.g., Fisher/importance and/or reference parameters), which is non-trivial under tight edge-memory budgets.

### 2.2. Bounded-memory Bayesian learning and controlled forgetting

A direct way to prevent long-horizon rigidity is to explicitly bound the amount of retained information. Bayesian learning and forgetting (Bonnet et al., 2025) targets a bounded-memory posterior that adds evidence from the current batch while removing the contribution of the oldest batch in an effective window of size $N$:

$$p(\boldsymbol{\omega}|\mathcal{D}_{t-N:t}) = \frac{p(\mathcal{D}_t|\boldsymbol{\omega})\,p(\boldsymbol{\omega}|\mathcal{D}_{t-N-1:t-1})}{p(\mathcal{D}_t)}$$
$$\times \frac{p(\mathcal{D}_{t-N-1})}{p(\mathcal{D}_{t-N-1}|\boldsymbol{\omega})}. \quad (1)$$

This yields controlled information decay with memory independent of stream length and provides an explicit stability-plasticity-forgetting trade-off. In the Gaussian case, this principle leads to MESU (Bonnet et al., 2025), which sustains adaptation while preserving meaningful uncertainty over long streams.

### 2.3. Bayesian binary networks and posterior saturation

Our work builds on the same bounded-memory principle as MESU but targets binary weights with mean-field Bernoulli posteriors. In this setting, a key long-horizon failure mode is *Bernoulli posterior saturation*: probabilities collapse toward 0 or 1, destroying epistemic uncertainty and making weight sign changes exceedingly unlikely. This freezing effect is observed in continual-learning practice for binary networks (Laborieux et al., 2021) and also arises when applying mean-field Bernoulli BNN training rules (e.g., Bayes-BiNN (Meng et al., 2020; Khan & Rue, 2023)) naïvely in long non-stationary streams without explicit forgetting.

### 2.4. Active learning under streaming and continual shift

Active learning (AL) aims to reduce annotation and update cost by querying informative samples, which is particularly valuable under class imbalance and rare-event streams (MacKay, 1996; Khan et al., 2019; Ngartera et al., 2024). Bayesian models naturally support AL through uncertainty-based acquisition functions such as mutual information (BALD) (Houlsby et al., 2011) or predictive en-

tropy (Shannon, 1948). However, much of the AL literature assumes pool-based storage and ranking of unlabeled data, which is a poor match for always-on streaming systems with strict memory and latency constraints.

Fully online AL instead uses one-pass decision rules that query immediately based on an uncertainty score (Liu et al., 2015; Rajendran et al., 2023). Disagreement-based measures such as the variation ratio (Freeman, 1965) are attractive in this setting because they depend only on Monte Carlo predicted labels, avoiding information-theoretic computations given the same MC predictions. Existing continual AL methods often add replay buffers, task-boundary assumptions, or consolidation state (Perkonigg et al., 2021; Ash et al., 2021; Das et al., 2023; Vu et al., 2024; Park et al., 2025), increasing overhead.

Taken together, these works highlight two requirements for practical online continual AL with binary posteriors: (i) preventing Bernoulli posterior saturation so epistemic uncertainty remains informative over long horizons, and (ii) exposing a cheap uncertainty signal that supports one-pass thresholding without buffering. BiMU addresses (i) through bounded-memory Bernoulli updates and metaplastic step sizing, and leverages (ii) via Monte Carlo disagreement thresholding to reduce both label queries and backpropagation updates in streaming continual learning.

## 3. Forgetting in binary Bayesian neural networks

### 3.1. Bernoulli variational parameterization

We approximate the posterior over $s$ binary weights $\boldsymbol{\omega} \in \{-1, +1\}^s$ with a mean-field Bernoulli distribution parameterized by natural parameters $\boldsymbol{\lambda} \in \mathbb{R}^s$. For each synapse $i \in [1, \dots, s]$,

$$\omega^{(i)} \sim 2\,\mathrm{Ber}\Big(\sigma(2\lambda^{(i)})\Big) - 1. \quad (2)$$

Equivalently (Prop. A.1),

$$q(\boldsymbol{\omega}|\boldsymbol{\lambda}) = \prod_{i=1}^{s} \frac{\exp\big(\lambda^{(i)}\omega^{(i)}\big)}{2\cosh\big(\lambda^{(i)}\big)}. \quad (3)$$

Here $\lambda^{(i)} = 0$ corresponds to maximal uncertainty $p(\omega^{(i)} = +1) = 1/2$, while large $|\lambda^{(i)}|$ yields near-deterministic weights. We denote by $\boldsymbol{\lambda}_{\mathrm{prior}}$ the natural parameters of the initialization prior $p(\boldsymbol{\omega})$.

### 3.2. Binary Metaplasticity from Uncertainty

At time step $t$ (one online update per batch $\mathcal{D}_t$), we seek an approximate posterior that retains only a bounded amount of information from the past, corresponding to a sliding window of roughly the last $N$ updates. Following the controlled-forgetting posterior (Eq. (1)), we compute $\boldsymbol{\lambda}_t$ by variational

projection:

$$\boldsymbol{\lambda}_t = \arg\min_{\boldsymbol{\lambda}} \mathcal{D}_{KL}(q(\boldsymbol{\omega}|\boldsymbol{\lambda}) \,\|\, p(\boldsymbol{\omega} \mid \mathcal{D}_{t-N:t})). \quad (4)$$

Using the variational decomposition (Prop. A.2), the objective is the sum of a data term $\mathcal{L}(\boldsymbol{\lambda}, \mathcal{D}_t)$ and KL regularizers. Under the mean-field Bernoulli approximation, the KL terms decouple across synapses (while the data term still couples parameters through the network). This yields a coordinate-wise update for each $\lambda^{(i)}$:

$$\lambda_t^{(i)} = \arg\min_{\lambda^{(i)}} \Big[ \mathcal{L}(\boldsymbol{\lambda}, \mathcal{D}_t) + \tfrac{1}{N} \mathcal{D}_{KL}\Big(q(\omega^{(i)}|\lambda^{(i)})\|p(\omega^{(i)})\Big)$$
$$+ \big(1 - \tfrac{1}{N}\big) \mathcal{D}_{KL}\Big(q(\omega^{(i)}|\lambda^{(i)})\|q(\omega^{(i)}|\lambda_{t-1}^{(i)})\Big) \Big]. \quad (5)$$

This form makes the roles explicit (Fig. 1): the data term drives *plasticity*, the KL-to-previous term provides *stability*, and the KL-to-prior term implements *controlled forgetting*, preventing unbounded evidence accumulation in long streams.

The window size $N$ should be interpreted as a forgetting horizon: smaller $N$ relaxes faster toward the prior, while larger $N$ approaches cumulative learning and can induce rigidity (Appendix G).

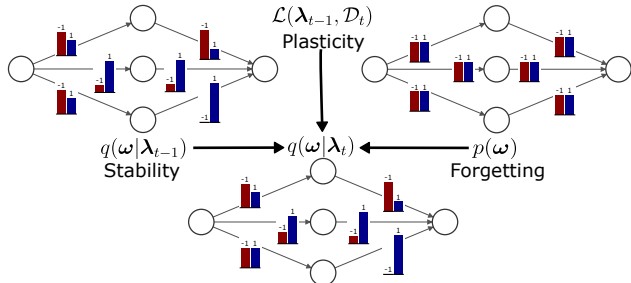

*Figure 1.* Schematic of the BiMU update. The next variational state is shaped jointly by the current loss (plasticity), the previous posterior (stability), and the prior (forgetting). Bars show Bernoulli probabilities for $\omega \in \{-1, +1\}$.

Eqs. (1)-(3) constitute the background that allows deriving BiMU. Combining the closed-form Bernoulli KL (Prop. A.3) with a second-order expansion around $\boldsymbol{\lambda}_{t-1}$ yields a per-synapse update (Theorem A.4):

$$\lambda_t^{(i)} = \lambda_{t-1}^{(i)} - \eta(\lambda_{t-1}^{(i)}) \left[ \frac{\partial \mathcal{L}}{\partial \lambda^{(i)}}\Big|_{\boldsymbol{\lambda}_{t-1}} + \frac{\lambda_{t-1}^{(i)} - \lambda_{\mathrm{prior}}^{(i)}}{N \cosh^2(\lambda_{t-1}^{(i)})} \right]. \quad (6)$$

The gradient term $\partial \mathcal{L}/\partial \lambda^{(i)}$ implements *plasticity*. The second term implements *controlled forgetting* via a relaxation toward the prior. This relaxation term is uncertainty-gated: since $\mathrm{Var}_q(\omega) = 1 - \tanh^2(\lambda) = \cosh^{-2}(\lambda)$, the pull

toward the prior is strongest for uncertain synapses and vanishes as $|\lambda| \to \infty$. Accordingly, the mechanism primarily prevents runaway growth of $|\lambda|$ that would otherwise drive Bernoulli probabilities toward 0 or 1 and collapse epistemic uncertainty.

The *stability* mechanism appears in the metaplastic step size $\eta(\cdot)$. In the exact second-order update (Theorem A.4), $\eta(\cdot)$ depends on a curvature term involving $\partial^2 \mathcal{L} / \partial(\lambda^{(i)})^2$. Estimating this curvature online is challenging for Bernoulli synapses: unlike mean-field Gaussians, Bernoulli posteriors do not expose a continuous dispersion parameter that enables MESU's Stein-based diagonal-Hessian estimator, and naive Fisher/Hessian approximations can be unstable in single-pass regimes. We therefore avoid explicit curvature estimation and use a bounded surrogate that preserves the qualitative effect of curvature while guaranteeing positivity and a maximum step size (Prop. A.5). Concretely, we use the element-wise learning rate $\eta(\boldsymbol{\lambda}_{t-1})$ defined by:

$$\frac{1}{\eta(\lambda_{t-1}^{(i)})} = \frac{1}{\cosh^2(\lambda_{t-1}^{(i)})} + 2\tanh(\lambda_{t-1}^{(i)}) \frac{\partial \mathcal{L}}{\partial \lambda^{(i)}}\Big|_{\boldsymbol{\lambda}_{t-1}}$$
$$+ \frac{1}{\alpha_{\max}} + 2\Big|\frac{\partial \mathcal{L}}{\partial \lambda^{(i)}}\Big|_{\boldsymbol{\lambda}_{t-1}}\Big|, \qquad (7)$$

The gradient-dependent terms make the step size *asymmetric*: consolidation steps that reinforce the current synaptic sign are full, whereas de-consolidation steps that push against the current sign are inhibited. This effect is seen in Fig. 2. Three regimes emerge: **(i) Uncertain synapses** ($\lambda^{(i)} \approx 0$): updates are conservatively bounded. **(ii) Consolidation** ($\lambda^{(i)} g^{(i)} < 0$): consistent gradients reinforce the current weight sign and $\eta$ approaches its upper bound, enabling fast consolidation. **(iii) De-consolidation** ($\lambda^{(i)} g^{(i)} > 0$): gradients opposing the current sign reduce $|\lambda^{(i)}|$, but $\eta$ shrinks, making sign changes difficult unless such gradients persist. This consolidation/de-consolidation asymmetry is consistent with the qualitative metaplastic behavior reported in Laborieux et al. (2021), but here it emerges directly from the Bayesian formulation and the bounded learning-rate construction.

Together, Eqs. (6)-(7) define **BiMU**: a fully online update that mitigates Bernoulli posterior saturation while preserving actionable epistemic uncertainty over long, non-stationary streams.

### 3.3. Uncertainty Estimation and Online Querying

BiMU can use posterior uncertainty as a control signal for fully online active continual learning: an example is labeled and triggers an update only when the current posterior has epistemic disagreement. Since $q(\boldsymbol{\omega}|\boldsymbol{\lambda})$ is Bernoulli, uncertainty can be estimated by sampling binary networks and running Monte Carlo (MC) forward passes, with no buffer or revisiting past data.

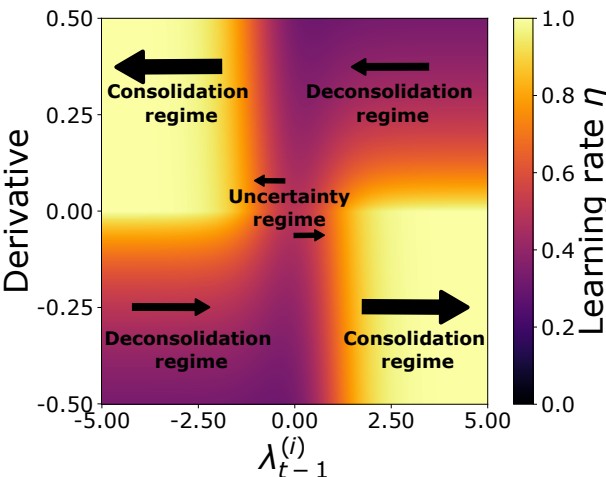

*Figure 2.* Color map of the learning rate $\eta$ ($\alpha_{\max} = 1$) as a function of the synaptic state $\lambda_{t-1}^{(i)}$ and the derivative $\partial \mathcal{L} / \partial \lambda^{(i)}$.

For an incoming (possibly unlabeled) example $x_j$, we draw $K$ independent weight samples $\{\boldsymbol{\omega}_k\}_{k=1}^K \sim q(\boldsymbol{\omega}|\boldsymbol{\lambda})$ and compute predictive distributions $\mathcal{V}_j = \{p(y|x_j, \boldsymbol{\omega}_k)\}_{k=1}^K$. Given $\mathcal{V}_j$, we compute an uncertainty score $u(\mathcal{V}_j) \in \mathbb{R}$ on-the-fly and use it for one-pass querying decisions.

**Predictive uncertainty decomposition.** In Bayesian predictive models, epistemic uncertainty can be quantified as the mutual information between predictions and parameters (Gal et al., 2017),

$$\mathcal{I}(\boldsymbol{\omega}, y|x_j) = H[p(y|x_j)] - \mathbb{E}_{q(\boldsymbol{\omega}|\boldsymbol{\lambda})}[H[p(y|x_j, \boldsymbol{\omega})]], \quad (8)$$

($H(\cdot)$ denotes Shannon entropy.), i.e., predictive uncertainty minus expected predictive entropy (aleatoric). We use these scores primarily for analysis and OOD detection. For online querying, we prefer a cheaper disagreement score.

**Variation ratio.** We adopt the *variation ratio* (VR) (Freeman, 1965), which measures disagreement among $K$ Monte Carlo predictors. Let $\hat{y}_k = \arg\max_{c \in \mathcal{C}} p(y_c|x_j, \boldsymbol{\omega}_k)$ be the predicted class under sample $k$, and let $f_{\text{mode}} = \max_{c \in \mathcal{C}} \sum_{k=1}^K \mathbb{1}[\hat{y}_k = c]$ be the number of samples predicting the modal class. We define $\text{VR} = 1 - f_{\text{mode}}/K$.

When labels are available, we also report the oracle diagnostic $\text{VR-True} = 1 - \frac{1}{K}\sum_{k=1}^K \mathbb{1}[\hat{y}_k = y_j]$.

The unlabeled scores above (VR, predictive uncertainty, aleatoric uncertainty, and epistemic uncertainty) are computed before observing $y_j$. Therefore, if the input distribution remains familiar but the task function changes, i.e., $p(y|x)$ shifts while $p(x)$ does not, unlabeled scores may fail to trigger a query because the input does not appear out-of-distribution. VR-True is particularly useful for diagnosing and adapting to such pure labeling-function shifts, but it

requires labels for the corresponding training samples.

**One-pass threshold querying.** At time $t$, we query the label of $x_j$ iff $u(\mathcal{V}_j) \geq \tau$, with a fixed threshold $\tau \in \mathbb{R}$. Only queried samples trigger backpropagation and a BiMU update. Computing the query score requires $K$ Monte Carlo forward passes, but these are particularly cheap in binary networks (bit-level sampling and low-cost binary arithmetic), whereas backpropagation and weight writes dominate on-device training cost.

## 4. Experiments

We evaluate BiMU along three axes: *long-horizon plasticity* under sustained non-stationarity without task-boundary signals; *uncertainty quality* for OOD detection; and *label/update efficiency* via online, buffer-free active querying. Methodological details, including the hyperparameter tuning procedure, are provided in Appendix B. For BiMU, the main tuned hyperparameters are the memory window $N$, the maximum metaplastic step size $\alpha_{\max}$, and the scaling coefficients of the likelihood and KL terms; for active learning, the query threshold $\tau$ controls the realized label/update budget.

### 4.1. Gradient estimation for Bernoulli synapses

BiMU provides a closed-form update for the stability and forgetting terms in the natural parameters $\boldsymbol{\lambda}$ (Eqs. (6)-(7)). In all experiments, we estimate stochastically $\nabla_{\boldsymbol{\lambda}} \mathcal{L}(\boldsymbol{\lambda}, \mathcal{D}_t)$ using a differentiable Bernoulli relaxation (Concrete / Gumbel-softmax reparameterization) (Maddison et al., 2017), following Algorithm 1 (Appendix O). Concretely, we draw relaxed weight samples, backpropagate through the relaxation, and average gradients over $K$ MC samples.

### 4.2. 1000-tasks Permuted-MNIST: long-horizon continual learning

We stress-test long-horizon online continual learning on 1000-tasks Permuted-MNIST, where each task applies a fixed random pixel permutation to MNIST. Training is strictly online (one epoch per task; batch size 1; no replay). Methods that require explicit task-boundary signals (Table 1, *Task bounds* = yes) are provided the task boundaries; BiMU and all *Task bounds* = no baselines are run without them.

We use a compact MLP with one hidden layer of 100 units to emphasize plasticity limits in a low-capacity binary regime. As a sanity check, the last column of Table 1 reports *single-task* accuracy. In this setting, BiMU and BayesBiNN (Meng et al., 2020; Khan & Rue, 2023) perform best, ahead of STE (Hubara et al., 2016) and Synaptic Metaplasticity (Laborieux et al., 2021), confirming that Bayesian binary training

ing is already competitive in this small-network regime.

*Table 1.* 1000-tasks Permuted-MNIST (online; 100-unit MLP). Mean accuracy over the last 5 tasks; OOD AUC (Permuted MNIST vs. Fashion-MNIST); MMRR; Single-task accuracy. Mean±std over 5 runs.

| METHOD | TASK BOUNDS | MEAN ACC. 5 TASKS (%) | OOD DET. (AUC) | MMRR | ACC. 1 TASK (%) |
|---|---|---|---|---|---|
| *Binary neural networks* | | | | | |
| BiMU | NO | $90.30 \pm 0.38$ | $0.99 \pm 0.00$ | $139.47$ | $94.67 \pm 0.11$ |
| BAYESBiNN | YES | $41.12 \pm 1.62$ | $0.57 \pm 0.12$ | $2.04$ | $93.22 \pm 0.09$ |
| SYN. META. | YES | $10.27 \pm 0.01$ | - | $1.64$ | $71.40 \pm 1.48$ |
| STE | NO | $29.35 \pm 0.96$ | $0.69 \pm 0.04$ | $9.32$ | $77.56 \pm 1.35$ |
| *Real-valued neural networks* | | | | | |
| MESU | NO | $91.69 \pm 0.58$ | $0.95 \pm 0.03$ | $261.10$ | $96.10 \pm 0.18$ |
| EWC O. | YES | $81.78 \pm 0.82$ | $0.66 \pm 0.11$ | $6.63$ | $96.06 \pm 0.11$ |
| SI | YES | $74.41 \pm 1.19$ | $0.66 \pm 0.17$ | $5.11$ | $95.55 \pm 0.28$ |
| SGD | NO | $66.64 \pm 2.70$ | $0.87 \pm 0.05$ | $43.50$ | $96.03 \pm 0.34$ |

After training on all 1000 tasks, we report three quantities (Table 1). (i) *MEAN ACC. (last 5 tasks)* averages test accuracy over the final five tasks and directly measures late-stream adaptability (ability to keep learning after a very long sequence of shifts). (ii) *MMRR* (Appendix D) summarizes rigidity accumulation by comparing late-stream performance to each method's best observed per-task performance over the stream. (iii) *OOD AUC* evaluates uncertainty-based discrimination between in-distribution Permuted-MNIST and Fashion-MNIST (see Appendix B for the uncertainty score used per method). Appendix C also reports backward transfer (BWT) and explains why late-stream accuracy and MMRR are more informative than backward transfer for diagnosing loss of plasticity in very long non-stationary streams.

Among binary-weight methods, BiMU is the only approach that remains accurate at the end of the 1000-tasks stream: it reaches $90.30\%$ mean accuracy on the last five tasks, whereas BayesBiNN, Synaptic Metaplasticity, and STE drop to $41.12\%$, $10.27\%$, and $29.35\%$, respectively. BiMU also has the highest MMRR among binary methods, indicating far less rigidity accumulation.

The gap to BayesBiNN reflects two mechanisms. First, BiMU implements bounded-memory forgetting. Second, BiMU uses the uncertainty-dependent metaplastic step size in Eq. (7), which changes how evidence is consolidated or de-consolidated online. The $N$ ablation supports this distinction: even when $N = 100{,}000$, where BiMU approaches cumulative learning, it remains above BayesBiNN in the 100-tasks study ($83.70\%$ vs. $67.91\%$ accuracy, $0.94$ vs. $0.75$ OOD AUC, MMRR $13.13$ vs. $5.03$; Appendix G). The same ablation study shows that the results of BiMU remain strong over a broad range of $N$ values. $N$ acts as an interpretable

stability-plasticity-forgetting control parameter, analogous to a replay size or discount factor.

STE, lacking a continual-learning mechanism, suffers both catastrophic forgetting and some rigidity from latent real-valued weight divergence over long training. Synaptic Metaplasticity has the strongest rigidity of all tested methods, by construction, due to its nearly irreversible metaplasticity mechanism.

Among real-valued baselines, MESU achieves the best late-stream accuracy (91.69%), narrowly above BiMU, but relies on a higher-dimensional posterior parameterization ($\mu, \sigma$). EWC Online and Synaptic Intelligence reach intermediate performance (81.78% and 74.41%), while SGD degrades substantially at the tail of the stream (66.64%).

Finally, BiMU yields the strongest OOD detection (0.99 AUC), ahead of MESU (0.95) and SGD (0.87). We do not report OOD AUC for Synaptic Metaplasticity because its near-chance late-stream accuracy makes its predictive uncertainty uninformative for in-vs-out discrimination in this setting. We further find that the Reverse Binary Gate activation sharpens disagreement between posterior samples on out-of-distribution inputs, improving OOD separability (Appendix H). BiMU's gains persist with increased network capacity (Appendix E) while retaining a minimal, constant training-state memory footprint (Appendix F).

### 4.3. OpenLORIS-Object: lifelong learning under nuisance-factor shifts

We next evaluate BiMU on OpenLORIS-Object, a lifelong recognition benchmark designed around robotic nuisance factors. We follow the sequential factors analysis protocol (12 tasks: illumination, occlusion, pixel corruption, and clutter at three difficulty levels) without active querying. To isolate online adaptation under constrained trainable capacity, we freeze an ImageNet-pretrained VGG19 feature extractor and train only an online linear classifier on top of its $512 \times 7 \times 7 = 25,088$-dimensional representation. To emulate edge-like feature-compression constraints, we additionally train the head on a fixed random subset of 8,192 and 1,024 features ($\approx 3\times$ and $25\times$ compression).

After training on all 12 tasks in a single pass (no replay), we report the mean accuracy averaged across the 12 task test sets (Table 2). For OOD evaluation, we remove the toy class during training and treat it as out-of-distribution at test time. We report ROC-AUC using aleatoric and epistemic uncertainty estimates.

Under strong compression (1,024 features), BiMU reaches 73.61% mean accuracy, improving substantially over STE, Synaptic Metaplasticity, and non-Bayesian real-valued base-

*Table 2.* OpenLORIS-Object (online; linear head on frozen VGG19). Mean accuracy over 12 tasks and OOD AUC (held-out toy); results for 1,024/8,192/25,088 features. Mean±std over 5 runs.

| METHOD | FEATURES | MEAN ACC. (%) | ALEATORIC (AUC) | EPISTEMIC (AUC) |
|---|---|---|---|---|
| *Binary neural networks* | | | | |
| BiMU | 1,024 | **73.61 ± 1.53** | **0.96 ± 0.01** | **1.00 ± 0.00** |
| | 8,192 | **89.19 ± 0.19** | **0.99 ± 0.00** | **1.00 ± 0.00** |
| | 25,088 | **90.62 ± 0.22** | **0.93 ± 0.00** | **0.90 ± 0.00** |
| BayesBiNN | 1,024 | 72.01 ± 1.69 | 0.93 ± 0.01 | **1.00 ± 0.00** |
| | 8,192 | 86.93 ± 0.41 | **0.99 ± 0.00** | **1.00 ± 0.00** |
| | 25,088 | 89.37 ± 0.77 | 0.92 ± 0.00 | **0.90 ± 0.01** |
| SYN. META. | 1,024 | 62.82 ± 2.31 | 0.72 ± 0.04 | – |
| | 8,192 | 88.03 ± 0.38 | 0.63 ± 0.03 | – |
| | 25,088 | 86.72 ± 0.34 | 0.55 ± 0.00 | – |
| STE | 1,024 | 52.88 ± 3.39 | 0.73 ± 0.02 | – |
| | 8,192 | 79.12 ± 1.39 | 0.61 ± 0.05 | – |
| | 25,088 | 83.79 ± 1.13 | 0.55 ± 0.00 | – |
| *Real-valued neural networks* | | | | |
| MESU | 1,024 | 80.82 ± 0.94 | 0.90 ± 0.01 | **0.99 ± 0.00** |
| | 8,192 | 87.01 ± 0.69 | **1.00 ± 0.00** | **0.98 ± 0.00** |
| | 25,088 | 87.84 ± 0.11 | **0.83 ± 0.00** | **0.86 ± 0.00** |
| EWC ONLINE | 1,024 | 75.72 ± 0.87 | **0.97 ± 0.01** | – |
| | 8,192 | 87.18 ± 0.58 | **1.00 ± 0.00** | – |
| | 25,088 | 88.23 ± 0.05 | 0.79 ± 0.00 | – |
| SI | 1,024 | 70.75 ± 0.57 | 0.95 ± 0.03 | – |
| | 8,192 | 86.44 ± 0.64 | **1.00 ± 0.00** | – |
| | 25,088 | 88.04 ± 0.03 | **0.83 ± 0.00** | – |
| SGD | 1,024 | 62.31 ± 1.80 | 0.50 ± 0.00 | – |
| | 8,192 | 86.27 ± 0.51 | **1.00 ± 0.00** | – |
| | 25,088 | 88.04 ± 0.03 | **0.83 ± 0.00** | – |

lines (SGD, SI), while remaining below real-valued MESU and EWC Online in this low-dimensional regime. As the feature dimension increases, BiMU scales smoothly to 89.19% at 8,192 features and 90.62% at 25,088 features, slightly above the real-valued baselines in this raw-feature setting. BiMU also remains consistently ahead of BayesBiNN, with a smaller gap than on 1000-tasks Permuted-MNIST, consistent with the shorter horizon (12 tasks) where Bernoulli posterior saturation is less pronounced.

Uncertainty remains informative under compression: BiMU attains near-perfect epistemic OOD separability (AUC = 1.00) at 1,024 and 8,192 features. At full dimensionality, epistemic AUC drops to 0.90 (a trend shared by BayesBiNN), indicating reduced uncertainty-based separability in this linear-head setting as capacity increases. When the extracted features are standardized using OpenLORIS statistics (offline control), real-valued methods regain their advantage (Appendix J), highlighting BiMU's robustness to unnormalized features in the streaming regime.

### 4.4. Animals: one-pass active learning under class imbalance

We first isolate the *active learning* component (without continual distribution shift) in a strongly imbalanced image-classification stream built from a 20-class subset of the Animals dataset (Jana, 2023) (Appendix B.5). We designate as *low-frequency* the classes with fewer than 200 training images and group the remaining classes as *high-frequency*. Images are encoded with a frozen ImageNet-pretrained VGG19 feature extractor (25,088-dim), and we train only an online linear classifier in a single pass (batch size 1; no replay buffer). Active learning follows our one-pass thresholding protocol: for each incoming sample, we compute an uncertainty score and query its label (and perform backpropagation) only if the score exceeds a fixed threshold; otherwise the example is skipped (no label request, no update). Therefore, the queried-label fraction directly measures both the *annotation budget* and the *update budget*. We compare random querying to predictive, aleatoric, epistemic uncertainty, and variation ratio (VR), averaging over five runs (with $K = 10$ MC predictors for Bayesian methods). We additionally report an oracle diagnostic, *VR-True*, which compares sampled predictions to the ground-truth label; it is not deployable because it assumes labels for all samples.

Fig. 3 shows final test accuracy as a function of the queried-label fraction, evaluated on a balanced test set (50 images per class). Selective querying consistently improves over random querying at matched budgets, indicating that one-pass active learning is effective in this low-label regime. The horizontal line shows *100% update baseline under the same one-pass protocol* (i.e., always querying/updating on every sample), reaching 86.28%. VR is the most effective criterion, outperforming entropy scores at comparable budgets: it reaches 84.46% accuracy while querying only 11% of labels, and reaches 87.12% at 18% labels, slightly exceeding the 100% update baseline, consistent with updates being concentrated on informative and under-represented examples rather than redundant majority-class samples.

Figs. 3(b,c) break down results by class frequency. Gains are driven primarily by low-frequency classes while high-frequency accuracy is largely preserved. At the 11% labeling budget, VR improves low-frequency accuracy by 41 absolute points over random querying, yielding substantially better balanced performance under severe imbalance.

Fig. 4 compares VR thresholding across Bayesian methods; for the deterministic STE baseline we use aleatoric uncertainty for querying. In this stationary setting, inherent class confusability makes entropy-based querying already useful, but posterior sampling further improves the accuracy-label trade-off when the disagreement signal is well calibrated.

Among Bayesian baselines, MESU is sensitive to feature scaling on raw VGG19 features; per-dimension standardization substantially improves MESU and can be competitive over a narrow range of query budgets (Appendix I).

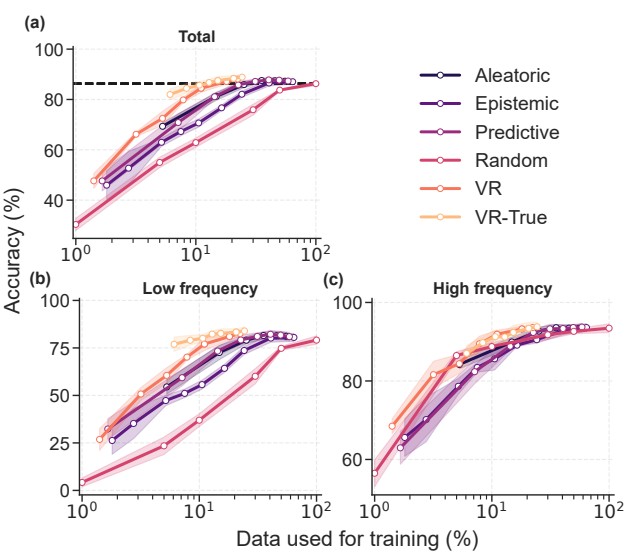

*Figure 3.* Imbalanced Animals active learning (frozen VGG19 features). Accuracy vs. queried-label fraction: (a) overall, (b) high-frequency classes, (c) low-frequency classes. Horizontal line: 100% update baseline. Mean over five runs.

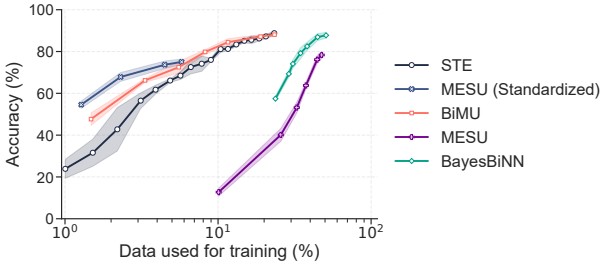

*Figure 4.* Variation ratio-based and STE aleatoric-based active learning on the Animals dataset.

In contrast, BiMU remains consistently strong without feature standardization, indicating that its disagreement signal remains informative under the native VGG19 feature distribution, a practical advantage for edge-like streams where reliable normalization may be unavailable or drift over time.

BayesBiNN performs markedly worse under VR querying, requiring substantially more labels to reach a given accuracy and often not exceeding random querying (Fig. 8, Appendix I). Empirically, its posterior-sample disagreement is less aligned with learning progress in this single-pass, imbalanced regime, causing VR to select less informative samples at matched budgets.

### 4.5. OpenLORIS-Object: active continual learning under shift and imbalance

We finally test whether BiMU's one-pass active-learning gains persist under *continual* nuisance-factor shifts. We use OpenLORIS-Object under the sequential factors analysis protocol (12 tasks) with a frozen VGG19 feature extractor and an online linear head, using a fixed random subset of 8,192 features from the original 25,088-dimensional representation. To induce a long-tailed stream, we define low-frequency classes as those with fewer than 700 training examples and randomly remove 50-80% of their training images. This makes rare categories appear more sparsely over time and increases the share of redundant majority-class samples. Active learning follows the same one-pass rule as in Sec. 4.4: for each incoming sample, we compute an uncertainty score and query its label only if it exceeds a fixed threshold. Only queried samples trigger labeling and backpropagation updates, so the queried fraction directly measures both annotation and update budgets.

Fig. 5 reports accuracy versus queried-label fraction (overall, low frequency, and high frequency classes). With VR thresholding (computed with $K = 10$ posterior-sampled predictors unless stated otherwise), BiMU reaches 88.70% while updating on only 3.1% of the stream, corresponding to a 32× reduction in labeled samples and gradient updates relative to the 100% update baseline (87.76%, equivalent to training on the full stream in this one-pass setting). With a 4.0% update budget, accuracy increases to 90.91% (a 25× reduction). The improvement over the 100% update baseline is consistent with the imbalanced stream: skipping redundant majority-class samples implicitly reweights updates toward informative and under-represented examples. The gains in the low-label regime are indeed driven primarily by improved performance on low-frequency categories (Fig. 5c), while avoiding redundant updates on frequent classes (Fig. 5b), yielding higher overall accuracy than random querying at matched budgets. Appendix M further shows that VR queries concentrate shortly after task shifts

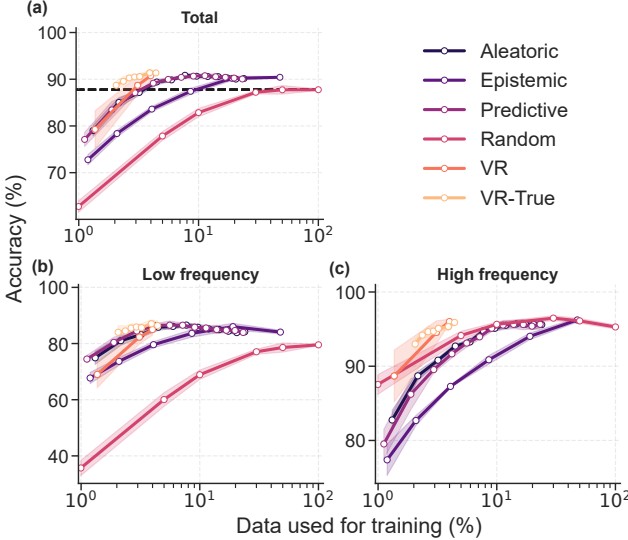

*Figure 5.* Active continual learning on OpenLORIS-Object under class imbalance (sequential factor analysis; frozen VGG19 features subsampled to 8,192 dims; online linear head). Accuracy vs. queried-label fraction for (a) overall accuracy, (b) frequent classes, and (c) rare classes. Horizontal line: 100% update baseline. Mean over five runs.

while remaining non-zero later in each task.

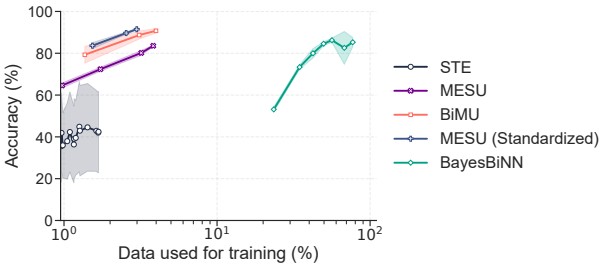

*Figure 6.* OpenLORIS-Object active continual learning (8,192 frozen VGG19 features): accuracy vs. queried-label fraction for BiMU, MESU, and BayesBiNN (VR thresholding) and STE (aleatoric thresholding). Results averaged over five runs.

Fig. 6 compares VR thresholding across Bayesian methods at matched queried-label fractions. BiMU achieves higher accuracy than MESU and BayesBiNN, indicating that its posterior-sample disagreement remains more useful for one-pass querying under feature compression and continual shift. Appendix K provides additional details for other algorithms. Appendix L further shows that small $K$ already recovers most of the label/update savings, supporting low-overhead deployment. Overall, BiMU enables effective active continual learning without replay buffers and without requiring task-identity/boundary signals by sustaining informative epistemic uncertainty in a binary-weight model.

## 5. Conclusion

We introduced BiMU, a Bayesian continual-learning rule for binary neural networks that prevents mean-field Bernoulli posteriors from saturating while preserving informative epistemic uncertainty on long, non-stationary streams. Starting from a bounded-memory Bayesian learning-and-forgetting objective, BiMU yields a fully online update in the natural parameters that combines a data-driven term with controlled relaxation toward the prior and a bounded, uncertainty-dependent (metaplastic) step size. This sustains plasticity without replay buffers and without requiring task-identity or task-boundary signals.

Empirically, BiMU maintains long-horizon adaptation where existing binary baselines lose plasticity. On 1000-tasks Permuted-MNIST, it achieves high final accuracy and near-perfect OOD detection, while BayesBiNN, Synaptic Metaplasticity, and STE degrade substantially as training proceeds. On OpenLORIS-Object with frozen VGG19 features and aggressive dimensionality reduction, BiMU remains competitive in the streaming setting while retaining a compact binary posterior and useful uncertainty estimates. Preserving epistemic uncertainty is also operationally beneficial: BiMU enables fully online active continual learning via one-pass Monte Carlo disagreement (variation ratio) thresholding. On OpenLORIS-Object under class imbalance, BiMU reaches $88.70\%$ accuracy while updating on only $3.1\%$ of samples (a $32\times$ reduction in labels and gradient updates), with gains driven primarily by improved performance on under-represented classes while avoiding redundant updates on frequent ones.

Appendix N further shows that the fixed threshold $\tau$ can be replaced by a fully online budget-driven controller that targets a desired query/update fraction. Appendix P quantifies the inference-update trade-off on an STM32 microcontroller unit and shows that active BiMU reduces expected compute despite the extra Monte Carlo forward passes.

BiMU makes uncertainty evaluation practical on energy-constrained edge platforms because its Monte Carlo inference is intrinsically cheaper than in real-valued Bayesian networks. Sampling Bernoulli synapses can be implemented with simple bit-level draws and comparisons, and each sampled forward pass leverages binary arithmetic (e.g., XNOR/popcount or bitwise accumulations) rather than real-valued multiplications, with substantially reduced data movement due to compact weight representations. This shifts the cost profile of uncertainty estimation toward operations that are well matched to microcontroller-class deployments (Cerutti et al., 2020) and becomes even more compelling on dedicated hardware (Conti et al., 2018), where emerging devices and circuits can provide stochastic sampling primitives at very low overhead (Querlioz & Vianello, 2025). When an update is required, however, backpropagation is still comparatively expensive; BiMU's one-pass querying mitigates this by triggering learning only on samples that are predicted to be informative under the current posterior. This makes the average training-time budget compatible with always-on operation, while retaining calibrated uncertainty for both OOD monitoring and selective supervision.

Overall, BiMU shows that binary Bayesian models can combine edge-efficient inference with sustained plasticity and actionable uncertainty, providing a practical route to reliable, data-efficient lifelong learning under tight memory, compute, and annotation budgets.

## Code availability

The code to reproduce our experiments is available at `https://github.com/kellian-cottart/active-continual-learning-bayesianbinn`.

## Acknowledgments

This work was supported by the Horizon Europe program (EIC Pathfinder METASPIN, grant number 326101098651). It also benefited from a France 2030 government grant managed by the French National Research Agency (ANR-23-PEIA-0002). The authors would like to thank Emre Neftci for discussion and invaluable feedback.

## Impact Statement

BiMU targets continual learning on resource-constrained devices by enabling fully online adaptation with binary Bayesian neural networks while preserving epistemic uncertainty over long, non-stationary streams. This can reduce energy use, labeling effort, and unnecessary gradient updates, and it can improve reliability through uncertainty-aware monitoring (e.g., selective supervision and out-of-distribution detection) in applications such as robotics and embedded vision.

At the same time, continual adaptation can amplify biases present in streaming and imbalanced data, and uncertainty estimates may become unreliable under unmodeled shifts or adversarial conditions. Practical deployments should include privacy and data-governance protections, ongoing evaluation across subpopulations, and human oversight, especially in safety-critical settings. Our contribution is methodological; its societal impact depends on how the approach is integrated, validated, and governed in downstream systems.

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

# A. Mathematical derivations

**Proposition A.1** (Exponential family product of Bernoulli)**.** *The joint distribution of $\boldsymbol{\omega}$ can be written in exponential form as*

$$q(\boldsymbol{\omega}|\boldsymbol{\lambda}) = \prod_{i=1}^{s} \frac{\exp\left(\lambda^{(i)}\omega^{(i)}\right)}{2\cosh\left(\lambda^{(i)}\right)}, \tag{9}$$

*where the natural parameters $\lambda^{(i)} \in \mathbb{R}$ are related to the Bernoulli parameters by*

$$p^{(i)} = \sigma\left(2\lambda^{(i)}\right). \tag{10}$$

*Proof.* We begin by using a change of variable that maps the standard Bernoulli support $\{0, 1\}$ to $\{-1, 1\}$. Let $v^{(i)} \sim$ Bernoulli$(p^{(i)})$ with $v^{(i)} \in \{0, 1\}$, and define

$$\omega^{(i)} = 2v^{(i)} - 1. \tag{11}$$

This transformation ensures that $\omega^{(i)} \in \{-1, 1\}$ and preserves the Bernoulli probability, with $\mathbb{P}(\omega^{(i)} = 1) = p^{(i)}$ and $\mathbb{P}(\omega^{(i)} = -1) = 1 - p^{(i)}$. For a single synapse $\omega^{(i)} \in \{-1, 1\}$, the Bernoulli probability mass function can be written as

$$q(\omega^{(i)}) = \left(p^{(i)}\right)^{\frac{1+\omega^{(i)}}{2}} \left(1 - p^{(i)}\right)^{\frac{1-\omega^{(i)}}{2}}. \tag{12}$$

Taking the logarithm yields

$$\log q(\omega^{(i)}) = \frac{1}{2}\left[(1 + \omega^{(i)})\log p^{(i)} + (1 - \omega^{(i)})\log(1 - p^{(i)})\right] \tag{13}$$

$$= \frac{1}{2}\left[\log\left(p^{(i)}(1 - p^{(i)})\right) + \omega^{(i)}\log\left(\frac{p^{(i)}}{1 - p^{(i)}}\right)\right]. \tag{14}$$

Defining the natural parameter

$$\lambda^{(i)} = \frac{1}{2}\log\left(\frac{p^{(i)}}{1 - p^{(i)}}\right), \tag{15}$$

The Bernoulli parameter $p^{(i)}$ can be recovered by inverting the definition of the natural parameter:

$$\lambda^{(i)} = \frac{1}{2}\log\left(\frac{p^{(i)}}{1 - p^{(i)}}\right) \iff \frac{p^{(i)}}{1 - p^{(i)}} = e^{2\lambda^{(i)}} \tag{16}$$

$$\iff p^{(i)} = \frac{e^{2\lambda^{(i)}}}{1 + e^{2\lambda^{(i)}}} = \sigma\left(2\lambda^{(i)}\right), \tag{17}$$

where $\sigma(\cdot)$ denotes the logistic sigmoid function. The distribution is rewritten as

$$q(\omega^{(i)}) = \exp\left(\lambda^{(i)}\omega^{(i)} + \frac{1}{2}\log\left(p^{(i)}(1 - p^{(i)})\right)\right). \tag{18}$$

Using the identity

$$p^{(i)}(1 - p^{(i)}) = \frac{1}{\left(2\cosh(\lambda^{(i)})\right)^2}, \tag{19}$$

the remaining log term simplifies to

$$\exp\left(\frac{1}{2}\log\left(p^{(i)}(1 - p^{(i)})\right)\right) = \frac{1}{2\cosh(\lambda^{(i)})}. \tag{20}$$

Hence, for each variable,

$$q(\omega^{(i)}) = \frac{\exp\left(\lambda^{(i)}\omega^{(i)}\right)}{2\cosh(\lambda^{(i)})}. \tag{21}$$

Finally, independence of the synapses implies that the joint distribution factorizes as

$$q(\boldsymbol{\omega}|\boldsymbol{\lambda}) = \prod_{i=1}^{s} q(\omega^{(i)}) = \prod_{i=1}^{s} \frac{\exp\left(\lambda^{(i)}\omega^{(i)}\right)}{2\cosh(\lambda^{(i)})}, \tag{22}$$

which completes the proof. $\qquad\square$

**Proposition A.2** (Variational free-energy decomposition under controlled forgetting)**.** *Let $p(\boldsymbol{\omega})$ be the prior distribution over binary synaptic variables $\boldsymbol{\omega}$, and let $\{\mathcal{D}_t\}_{t>0}$ denote a stream of data batches. Assume that all past batches $\mathcal{D}_{t-N-1}, \ldots, \mathcal{D}_{t-1}$ have equal marginal likelihood. Then, approximating the posterior $p(\boldsymbol{\omega}|\mathcal{D}_{t-N:t})$ with a variational distribution $q(\boldsymbol{\omega}|\boldsymbol{\lambda})$, the variational free energy at time $t$ decomposes as*

$$\mathcal{D}_{KL}\left(q(\boldsymbol{\omega}|\boldsymbol{\lambda}) \,\|\, p(\boldsymbol{\omega}|\mathcal{D}_{t-N:t})\right) \approx \left(1 - \frac{1}{N}\right) \mathcal{D}_{KL}\left(q(\boldsymbol{\omega}|\boldsymbol{\lambda}) \,\|\, q(\boldsymbol{\omega}|\boldsymbol{\lambda}_{t-1})\right) + \frac{1}{N}\mathcal{D}_{KL}\left(q(\boldsymbol{\omega}|\boldsymbol{\lambda}) \,\|\, p(\boldsymbol{\omega})\right) + \mathcal{L}(\boldsymbol{\lambda}, \mathcal{D}_t), \tag{23}$$

*where*

$$\mathcal{L}(\boldsymbol{\lambda}, \mathcal{D}_t) = -\mathbb{E}_{q(\boldsymbol{\omega}|\boldsymbol{\lambda})}[\log p(\mathcal{D}_t|\boldsymbol{\omega})] \tag{24}$$

*is the negative log-likelihood of the current batch.*

*Proof.* By Bayes' rule with controlled forgetting, the posterior over the window $\mathcal{D}_{t-N:t}$ satisfies

$$p(\boldsymbol{\omega}|\mathcal{D}_{t-N:t}) \propto p(\mathcal{D}_t|\boldsymbol{\omega})\, p(\boldsymbol{\omega}|\mathcal{D}_{t-N-1:t-1}) \left(\frac{p(\boldsymbol{\omega})}{p(\boldsymbol{\omega}|\mathcal{D}_{t-N-1:t-1})}\right)^{\frac{1}{N}}, \tag{25}$$

where the equal-marginal-likelihood assumption implies

$$p(\mathcal{D}_{t-N-1:t-1}|\boldsymbol{\omega}) = \prod_{i=t-N-1}^{t-1} p(\mathcal{D}_i|\boldsymbol{\omega}) = \left[p(\mathcal{D}_{t-N-1}|\boldsymbol{\omega})\right]^N. \tag{26}$$

We now consider the variational objective

$$\mathcal{D}_{KL}\left(q(\boldsymbol{\omega}|\boldsymbol{\lambda}) \,\|\, p(\boldsymbol{\omega}|\mathcal{D}_{t-N:t})\right) = \mathbb{E}_q[\log q(\boldsymbol{\omega}|\boldsymbol{\lambda})] - \mathbb{E}_q[\log p(\boldsymbol{\omega}|\mathcal{D}_{t-N:t})]. \tag{27}$$

Substituting the expression of the posterior and expanding the logarithm yields

$$\mathcal{D}_{KL}\left(q(\boldsymbol{\omega}|\boldsymbol{\lambda}) \,\|\, p(\boldsymbol{\omega}|\mathcal{D}_{t-N:t})\right) = \mathbb{E}_q[\log q(\boldsymbol{\omega}|\boldsymbol{\lambda})] - \mathbb{E}_q[\log p(\boldsymbol{\omega}|\mathcal{D}_{t-N-1:t-1})]$$
$$- \frac{1}{N}\mathbb{E}_q\left[\log \frac{p(\boldsymbol{\omega})}{p(\boldsymbol{\omega}|\mathcal{D}_{t-N-1:t-1})}\right] - \mathbb{E}_q[\log p(\mathcal{D}_t|\boldsymbol{\omega})]. \tag{28}$$

We multiply and divide by the variational distribution

$$\mathcal{D}_{KL}\left(q(\boldsymbol{\omega}|\boldsymbol{\lambda}) \,\|\, p(\boldsymbol{\omega}|\mathcal{D}_{t-N:t})\right) = \mathbb{E}_q[\log q(\boldsymbol{\omega}|\boldsymbol{\lambda})] - \mathbb{E}_q[\log p(\boldsymbol{\omega}|\mathcal{D}_{t-N-1:t-1})]$$
$$- \mathbb{E}_q\left[\log\left(\left[\frac{p(\boldsymbol{\omega})}{p(\boldsymbol{\omega}|\mathcal{D}_{t-N-1:t-1})} \frac{q(\boldsymbol{\omega}|\boldsymbol{\lambda})}{q(\boldsymbol{\omega}|\boldsymbol{\lambda})}\right]^{\frac{1}{N}}\right)\right] - \mathbb{E}_q[\log p(\mathcal{D}_t|\boldsymbol{\omega})]. \tag{29}$$

Rearranging terms, we obtain

$$\mathcal{D}_{KL}\left(q(\boldsymbol{\omega}|\boldsymbol{\lambda}) \,\|\, p(\boldsymbol{\omega}|\mathcal{D}_{t-N:t})\right) = \left(1 - \frac{1}{N}\right) \mathcal{D}_{KL}\left(q(\boldsymbol{\omega}|\boldsymbol{\lambda}) \,\|\, p(\boldsymbol{\omega}|\mathcal{D}_{t-N-1:t-1})\right)$$
$$+ \frac{1}{N}\mathcal{D}_{KL}\left(q(\boldsymbol{\omega}|\boldsymbol{\lambda}) \,\|\, p(\boldsymbol{\omega})\right) - \mathbb{E}_q[\log p(\mathcal{D}_t|\boldsymbol{\omega})]. \tag{30}$$

Finally, since the true posterior $p(\boldsymbol{\omega}|\mathcal{D}_{t-N-1:t-1})$ is intractable, we approximate it by the previous variational solution $q(\boldsymbol{\omega}|\boldsymbol{\lambda}_{t-1})$, assuming it minimized the divergence at the previous time step. Defining

$$\mathcal{L}(\boldsymbol{\lambda}, \mathcal{D}_t) = -\mathbb{E}_{q(\boldsymbol{\omega}|\boldsymbol{\lambda})}[\log p(\mathcal{D}_t|\boldsymbol{\omega})], \tag{31}$$

the stated decomposition follows. $\qquad\square$

**Proposition A.3** (Closed-form KL divergence for Bernoulli variables). *Let $q(\omega|\theta)$ and $q(\omega|\xi)$ be Bernoulli distributions in exponential-family form with support $\mathcal{X} = \{-1, 1\}$, $\omega \in \mathcal{X}$:*

$$q(\omega|\theta) = \frac{\exp(\theta\,\omega)}{2\cosh(\theta)}, \quad q(\omega|\xi) = \frac{\exp(\xi\,\omega)}{2\cosh(\xi)}. \tag{32}$$

*The KL divergence between these two distributions admits the closed-form expression:*

$$\mathcal{D}_{KL}\big(q(\omega|\theta)\,\|\,q(\omega|\xi)\big) = \mathbb{E}_{q(\omega|\theta)}\left[\log\frac{q(\omega|\theta)}{q(\omega|\xi)}\right]$$

$$= (\theta - \xi)\tanh(\theta) - \log\frac{\cosh(\theta)}{\cosh(\xi)}. \tag{33}$$

*Proof.* By definition, the KL divergence is given by

$$\mathcal{D}_{KL}\big(q(\omega|\theta)\,\|\,q(\omega|\xi)\big) := \sum_{\mathcal{X}} q(\omega|\theta)\log\frac{q(\omega|\theta)}{q(\omega|\xi)}. \tag{34}$$

Substituting the exponential-family forms:

$$q(\omega|\theta)\log\frac{q(\omega|\theta)}{q(\omega|\xi)} = \frac{\exp(\theta\,\omega)}{2\cosh(\theta)}\log\frac{\exp(\theta\,\omega)/2\cosh(\theta)}{\exp(\xi\,\omega)/2\cosh(\xi)}$$

$$= \frac{\exp(\theta\,\omega)}{2\cosh(\theta)}\left[(\theta - \xi)\,\omega - \log\frac{\cosh(\theta)}{\cosh(\xi)}\right]. \tag{35}$$

Summing over $\mathcal{X}$:

$$\mathcal{D}_{KL}\big(q(\omega|\theta)\,\|\,q(\omega|\xi)\big) = \frac{\exp(\theta)}{2\cosh(\theta)}\left[(\theta - \xi) - \log\frac{\cosh(\theta)}{\cosh(\xi)}\right]$$

$$+ \frac{\exp(-\theta)}{2\cosh(\theta)}\left[-(\theta - \xi) - \log\frac{\cosh(\theta)}{\cosh(\xi)}\right]$$

$$= (\theta - \xi)\frac{\exp(\theta) - \exp(-\theta)}{2\cosh(\theta)} - \log\frac{\cosh(\theta)}{\cosh(\xi)}$$

$$= (\theta - \xi)\tanh(\theta) - \log\frac{\cosh(\theta)}{\cosh(\xi)}. \tag{36}$$

This completes the derivation of the closed-form. $\qquad\square$

**Theorem A.4** (Memory-limited second-order asymmetric update for binary synapses). *Consider a Bayesian neural network with synaptic weights $\boldsymbol{\omega} = (\omega^{(1)}, \ldots, \omega^{(s)}) \in \mathcal{X}^s$, with $\mathcal{X} = \{-1, 1\}$ Assume a mean-field Bernoulli variational approximation $q(\boldsymbol{\omega}|\boldsymbol{\lambda}) = \prod_{i=1}^{s} q(\omega^{(i)}|\lambda^{(i)})$, where each synapse follows the exponential-family form*

$$q(\omega^{(i)}|\lambda^{(i)}) = \frac{\exp\big(\lambda^{(i)}\omega^{(i)}\big)}{2\cosh(\lambda^{(i)})}. \tag{37}$$

*Let $\mathcal{D}_t$ denote the current batch of data and $\mathcal{L}(\boldsymbol{\lambda}, \mathcal{D}_t) = -\mathbb{E}_{q(\boldsymbol{\omega}|\boldsymbol{\lambda})}[\log p(\mathcal{D}_t|\boldsymbol{\omega})]$ the negative log-likelihood. Assume a memory window of size $N$ and a prior parameter $\lambda_{prior}^{(i)}$.*

*The second-order expansion of the variational free energy yields the following asymmetric optimal update rule:*

$$\lambda_t^{(i)} = \lambda_{t-1}^{(i)} - \frac{\left.\dfrac{\partial\mathcal{L}(\boldsymbol{\lambda}, \mathcal{D}_t)}{\partial\lambda^{(i)}}\right|_{\boldsymbol{\lambda}=\boldsymbol{\lambda}_{t-1}} + \dfrac{\lambda_{t-1}^{(i)} - \lambda_{prior}^{(i)}}{N\,\cosh^2(\lambda_{t-1}^{(i)})}}{\dfrac{1}{\cosh^2(\lambda_{t-1}^{(i)})} + 2\tanh(\lambda_{t-1}^{(i)})\left.\dfrac{\partial\mathcal{L}(\boldsymbol{\lambda}, \mathcal{D}_t)}{\partial\lambda^{(i)}}\right|_{\boldsymbol{\lambda}=\boldsymbol{\lambda}_{t-1}} + \left.\dfrac{\partial^2\mathcal{L}(\boldsymbol{\lambda}, \mathcal{D}_t)}{\partial\lambda^{(i)2}}\right|_{\boldsymbol{\lambda}=\boldsymbol{\lambda}_{t-1}}}. \tag{38}$$

*This update corresponds to a memory-modulated second-order step with an asymmetric, state-dependent learning rate.*

*Proof.* We derive the optimal update for the variational parameter $\lambda^{(i)}$ by minimizing the variational free energy associated with synapse $i$ under the memory-limited Bayesian continual learning objective.

**Variational objective.** Following the derivation of the memory-limited posterior in Prop. A.2, the variational free energy for a single synapse $i$ is given by

$$\mathcal{F}(\lambda^{(i)}, \mathcal{D}_t) = \left(1 - \frac{1}{N}\right) \mathcal{D}_{KL}\Big(q(\omega^{(i)}|\lambda^{(i)}) \,\|\, q(\omega^{(i)}|\lambda^{(i)}_{t-1})\Big) + \frac{1}{N}\mathcal{D}_{KL}\Big(q(\omega^{(i)}|\lambda^{(i)}) \,\|\, p(\omega^{(i)})\Big) + \mathcal{L}(\boldsymbol{\lambda}, \mathcal{D}_t), \quad (39)$$

where $\mathcal{L}(\boldsymbol{\lambda}, \mathcal{D}_t) = -\mathbb{E}_{q(\boldsymbol{\omega}|\boldsymbol{\lambda})}[\log p(\mathcal{D}_t|\boldsymbol{\omega})]$ is the negative log-likelihood.

**Exponential-family form.** Following Prop. A.1, each synapse follows a Bernoulli distribution over $\mathcal{X}$, written in exponential-family form as

$$q(\omega^{(i)}|\lambda^{(i)}) = \frac{\exp(\lambda^{(i)}\omega^{(i)})}{2\cosh(\lambda^{(i)})}. \quad (40)$$

The prior $p(\omega^{(i)})$ is of the same form with natural parameter $\lambda^{(i)}_{\text{prior}}$.

**Closed-form KL divergence.** Prop. A.3 states that for any Bernoulli distributions over $\omega \in \mathcal{X}$ in exponential-family form $q(\omega|\theta)$ and $q(\omega|\xi)$:

$$q(\omega|\theta) = \frac{\exp(\theta\,\omega)}{2\cosh(\theta)}, \quad q(\omega|\xi) = \frac{\exp(\xi\,\omega)}{2\cosh(\xi)}, \quad (41)$$

the KL divergence between these two distributions admits the closed-form expression:

$$\begin{aligned}
\mathcal{D}_{KL}\big(q(\omega|\theta) \,\|\, q(\omega|\xi)\big) &= \mathbb{E}_{q(\omega|\theta)}\left[\log\frac{q(\omega|\theta)}{q(\omega|\xi)}\right] \\
&= (\theta - \xi)\tanh(\theta) - \log\frac{\cosh(\theta)}{\cosh(\xi)}.
\end{aligned} \quad (42)$$

**Closed-form free-energy.** Substituting the closed-form KL divergence of Bernoulli distributions from (Eq. (42)) into the single-synapse variational free energy (39), we obtain:

$$\begin{aligned}
\mathcal{F}(\lambda^{(i)}, \mathcal{D}_t) &= \left(1 - \frac{1}{N}\right)\left[(\lambda^{(i)} - \lambda^{(i)}_{t-1})\tanh(\lambda^{(i)}) - \log\frac{\cosh(\lambda^{(i)})}{\cosh(\lambda^{(i)}_{t-1})}\right] \\
&\quad + \frac{1}{N}\left[(\lambda^{(i)} - \lambda^{(i)}_{\text{prior}})\tanh(\lambda^{(i)}) - \log\frac{\cosh(\lambda^{(i)})}{\cosh(\lambda^{(i)}_{\text{prior}})}\right] + \mathcal{L}(\boldsymbol{\lambda}, \mathcal{D}_t).
\end{aligned} \quad (43)$$

We reorganize the terms to isolate contributions from previous parameter values $\lambda^{(i)}_{t-1}$ and the prior $\lambda^{(i)}_{\text{prior}}$. First, note that

$$(\lambda^{(i)} - \lambda^{(i)}_{\text{prior}}) = (\lambda^{(i)} - \lambda^{(i)}_{t-1}) + (\lambda^{(i)}_{t-1} - \lambda^{(i)}_{\text{prior}}),$$

so that

$$\frac{1}{N}(\lambda^{(i)} - \lambda^{(i)}_{\text{prior}})\tanh(\lambda^{(i)}) = \frac{1}{N}(\lambda^{(i)} - \lambda^{(i)}_{t-1})\tanh(\lambda^{(i)}) + \frac{1}{N}(\lambda^{(i)}_{t-1} - \lambda^{(i)}_{\text{prior}})\tanh(\lambda^{(i)}). \quad (44)$$

Similarly, the logarithmic term can be split:

$$-\frac{1}{N}\log\frac{\cosh(\lambda^{(i)})}{\cosh(\lambda^{(i)}_{\text{prior}})} = -\frac{1}{N}\log\frac{\cosh(\lambda^{(i)})}{\cosh(\lambda^{(i)}_{t-1})} - \frac{1}{N}\log\frac{\cosh(\lambda^{(i)}_{t-1})}{\cosh(\lambda^{(i)}_{\text{prior}})}. \quad (45)$$

Combining these expansions with the first KL term weighted by $1 - 1/N$, we factor the contributions of $(\lambda^{(i)} - \lambda_{t-1}^{(i)}) \tanh(\lambda^{(i)})$ and $-\log(\cosh(\lambda^{(i)})/\cosh(\lambda_{t-1}^{(i)}))$:

$$
\mathcal{F}(\lambda^{(i)}, \mathcal{D}_t) = (\lambda^{(i)} - \lambda_{t-1}^{(i)}) \tanh(\lambda^{(i)}) - \log \frac{\cosh(\lambda^{(i)})}{\cosh(\lambda_{t-1}^{(i)})}
$$

$$
+ \frac{1}{N} \left[ (\lambda_{t-1}^{(i)} - \lambda_{\text{prior}}^{(i)}) \tanh(\lambda^{(i)}) - \log \frac{\cosh(\lambda_{t-1}^{(i)})}{\cosh(\lambda_{\text{prior}}^{(i)})} \right]
$$

$$
+ \mathcal{L}(\boldsymbol{\lambda}, \mathcal{D}_t). \tag{46}
$$

**First-order optimality condition.** Differentiating (46) with respect to $\lambda^{(i)}$ yields

$$
\frac{\partial \mathcal{F}}{\partial \lambda^{(i)}} = \frac{1}{\cosh^2(\lambda^{(i)})} \left[ (\lambda^{(i)} - \lambda_{t-1}^{(i)}) + \frac{\lambda_{t-1}^{(i)} - \lambda_{\text{prior}}^{(i)}}{N} \right] + \frac{\partial \mathcal{L}(\boldsymbol{\lambda}, \mathcal{D}_t)}{\partial \lambda^{(i)}}. \tag{47}
$$

The optimal update $\lambda_t^{(i)}$ satisfies

$$
\frac{\partial \mathcal{F}}{\partial \lambda^{(i)}} = 0 \iff \lambda^{(i)} - \lambda_{t-1}^{(i)} = -\cosh^2(\lambda^{(i)}) \frac{\partial \mathcal{L}(\boldsymbol{\lambda}, \mathcal{D}_t)}{\partial \lambda^{(i)}} - \frac{\lambda_{t-1}^{(i)} - \lambda_{\text{prior}}^{(i)}}{N} \tag{48}
$$

**Second-order expansion.** To retrieve terms in $\lambda_{t-1}^{(i)}$, we apply a second-order Taylor expansion around $\lambda_{t-1}^{(i)}$ for $\cosh^2(\lambda^{(i)}) \frac{\partial \mathcal{L}(\boldsymbol{\lambda}, \mathcal{D}_t)}{\partial \lambda^{(i)}}$, with $\mathcal{L}$ twice differentiable, and $\cosh^2(\lambda^{(i)})$ differentiable.

$$
\cosh^2(\lambda_t^{(i)}) \frac{\partial \mathcal{L}(\boldsymbol{\lambda}, \mathcal{D}_t)}{\partial \lambda^{(i)}} \approx \cosh^2(\lambda_{t-1}^{(i)}) \frac{\partial \mathcal{L}(\boldsymbol{\lambda}, \mathcal{D}_t)}{\partial \lambda^{(i)}} \Big|_{\boldsymbol{\lambda} = \boldsymbol{\lambda}_{t-1}}
$$

$$
+ (\lambda_t^{(i)} - \lambda_{t-1}^{(i)}) \left[ \cosh^2(\lambda_{t-1}^{(i)}) \frac{\partial^2 \mathcal{L}(\boldsymbol{\lambda}, \mathcal{D}_t)}{\partial (\lambda^{(i)})^2} \Big|_{\boldsymbol{\lambda} = \boldsymbol{\lambda}_{t-1}} \right.
$$

$$
\left. + 2\sinh(\lambda_{t-1}^{(i)}) \cosh(\lambda_{t-1}^{(i)}) \frac{\partial \mathcal{L}(\boldsymbol{\lambda}, \mathcal{D}_t)}{\partial (\lambda^{(i)})} \Big|_{\boldsymbol{\lambda} = \boldsymbol{\lambda}_{t-1}} \right] \tag{49}
$$

**Solving for the update.**

Using the identity $\tanh(x) = \sinh(x)/\cosh(x)$, substituting (49) into the stationary condition (48) and collecting terms linear in $\Delta \lambda^{(i)} = \lambda_t^{(i)} - \lambda_{t-1}^{(i)}$ yields

$$
\lambda_t^{(i)} = \lambda_{t-1}^{(i)} - \frac{\frac{\partial \mathcal{L}(\boldsymbol{\lambda}, \mathcal{D}_t)}{\partial \lambda^{(i)}} \Big|_{\boldsymbol{\lambda} = \boldsymbol{\lambda}_{t-1}} + \frac{\lambda_{t-1}^{(i)} - \lambda_{\text{prior}}^{(i)}}{N \cosh^2(\lambda_{t-1}^{(i)})}}{\frac{1}{\cosh^2(\lambda_{t-1}^{(i)})} + 2\tanh(\lambda_{t-1}^{(i)}) \frac{\partial \mathcal{L}(\boldsymbol{\lambda}, \mathcal{D}_t)}{\partial \lambda^{(i)}} \Big|_{\boldsymbol{\lambda} = \boldsymbol{\lambda}_{t-1}} + \frac{\partial^2 \mathcal{L}(\boldsymbol{\lambda}, \mathcal{D}_t)}{\partial (\lambda^{(i)})^2} \Big|_{\boldsymbol{\lambda} = \boldsymbol{\lambda}_{t-1}}}. \tag{50}
$$

This concludes the proof. $\qquad \square$

**Proposition A.5** (Bounded Learning Rate with a Curvature Surrogate). *Consider the binary Bayesian update rule with learning rate*

$$
\eta(\lambda_{t-1}^{(i)}) = \frac{1}{\frac{1}{\cosh^2(\lambda_{t-1}^{(i)})} + 2\tanh(\lambda_{t-1}^{(i)}) \frac{\partial \mathcal{L}(\boldsymbol{\lambda}, \mathcal{D}_t)}{\partial \lambda^{(i)}} \Big|_{\boldsymbol{\lambda} = \boldsymbol{\lambda}_{t-1}} + \frac{\partial^2 \mathcal{L}(\boldsymbol{\lambda}, \mathcal{D}_t)}{\partial (\lambda^{(i)})^2} \Big|_{\boldsymbol{\lambda} = \boldsymbol{\lambda}_{t-1}}}. \tag{51}
$$

*In large-scale neural networks, the exact evaluation of the second derivative term $\partial^2 \mathcal{L}/\partial (\lambda^{(i)})^2$ is computationally prohibitive. Let $f : \mathbb{R} \to \mathbb{R}$ be a surrogate curvature function that replaces this second derivative in the denominator.*

*Fix a maximum learning rate $\alpha_{\max} > 0$. If $f$ satisfies*

$$f(\lambda_{t-1}^{(i)}) \geq \frac{1}{\alpha_{\max}} - \frac{1}{\cosh^2(\lambda_{t-1}^{(i)})} - 2\tanh(\lambda_{t-1}^{(i)})\frac{\partial\mathcal{L}(\boldsymbol{\lambda}, \mathcal{D}_t)}{\partial\lambda^{(i)}}\bigg|_{\boldsymbol{\lambda}=\boldsymbol{\lambda}_{t-1}}, \tag{52}$$

*then the resulting learning rate satisfies*

$$0 < \eta(\lambda_{t-1}^{(i)}) \leq \alpha_{\max} \quad \forall\lambda_{t-1}^{(i)} \in \mathbb{R},$$

*and therefore defines a valid bounded gradient descent step.*

*In particular, the choice*

$$f(\lambda_{t-1}^{(i)}) = \frac{1}{\alpha_{\max}} + 2\left|\frac{\partial\mathcal{L}(\boldsymbol{\lambda}, \mathcal{D}_t)}{\partial\lambda^{(i)}}\bigg|_{\boldsymbol{\lambda}=\boldsymbol{\lambda}_{t-1}}\right| \tag{53}$$

*is a computationally efficient surrogate that guarantees positivity and boundedness of the learning rate without requiring second-order derivatives.*

*Proof.* Fix $\alpha_{\max} > 0$. By definition of the update rule, replacing the second derivative with a surrogate curvature function $f$ yields

$$\frac{1}{\eta(\lambda_{t-1}^{(i)})} = \frac{1}{\cosh^2(\lambda_{t-1}^{(i)})} + 2\tanh(\lambda_{t-1}^{(i)})\frac{\partial\mathcal{L}(\boldsymbol{\lambda}, \mathcal{D}_t)}{\partial\lambda^{(i)}}\bigg|_{\boldsymbol{\lambda}=\boldsymbol{\lambda}_{t-1}} + f(\lambda_{t-1}^{(i)}). \tag{54}$$

The constraint $0 < \eta(\lambda_{t-1}^{(i)}) \leq \alpha_{\max}$ is equivalent to requiring

$$\frac{1}{\eta(\lambda_{t-1}^{(i)})} \geq \frac{1}{\alpha_{\max}}. \tag{55}$$

Thus, a sufficient condition is

$$\frac{1}{\cosh^2(\lambda_{t-1}^{(i)})} + 2\tanh(\lambda_{t-1}^{(i)})\frac{\partial\mathcal{L}(\boldsymbol{\lambda}, \mathcal{D}_t)}{\partial\lambda^{(i)}}\bigg|_{\boldsymbol{\lambda}=\boldsymbol{\lambda}_{t-1}} + f(\lambda_{t-1}^{(i)}) \geq \frac{1}{\alpha_{\max}}. \tag{56}$$

We now derive a uniform lower bound for the terms that do not involve $f$. Since $1/\cosh^2(x) \geq 0$ and $|\tanh(x)| \leq 1$ for all $x \in \mathbb{R}$, we obtain

$$2\tanh(\lambda_{t-1}^{(i)})\frac{\partial\mathcal{L}(\boldsymbol{\lambda}, \mathcal{D}_t)}{\partial\lambda^{(i)}}\bigg|_{\boldsymbol{\lambda}=\boldsymbol{\lambda}_{t-1}} \geq -2\left|\frac{\partial\mathcal{L}(\boldsymbol{\lambda}, \mathcal{D}_t)}{\partial\lambda^{(i)}}\bigg|_{\boldsymbol{\lambda}=\boldsymbol{\lambda}_{t-1}}\right|. \tag{57}$$

Consequently,

$$\frac{1}{\cosh^2(\lambda_{t-1}^{(i)})} + 2\tanh(\lambda_{t-1}^{(i)})\frac{\partial\mathcal{L}(\boldsymbol{\lambda}, \mathcal{D}_t)}{\partial\lambda^{(i)}}\bigg|_{\boldsymbol{\lambda}=\boldsymbol{\lambda}_{t-1}} \geq -2\left|\frac{\partial\mathcal{L}(\boldsymbol{\lambda}, \mathcal{D}_t)}{\partial\lambda^{(i)}}\bigg|_{\boldsymbol{\lambda}=\boldsymbol{\lambda}_{t-1}}\right|. \tag{58}$$

Substituting (58) into (56), it suffices to define the surrogate curvature term as

$$f(\lambda_{t-1}^{(i)}) \geq \frac{1}{\alpha_{\max}} + 2\left|\frac{\partial\mathcal{L}(\boldsymbol{\lambda}, \mathcal{D}_t)}{\partial\lambda^{(i)}}\bigg|_{\boldsymbol{\lambda}=\boldsymbol{\lambda}_{t-1}}\right|. \tag{59}$$

This choice of $f$ depends only on first-order information and can be evaluated efficiently even in large-scale neural networks. With this surrogate, the denominator in (54) is strictly positive and uniformly lower bounded by $1/\alpha_{\max}$ for all $\lambda_{t-1}^{(i)} \in \mathbb{R}$, ensuring $0 < \eta(\lambda_{t-1}^{(i)}) \leq \alpha_{\max}$. $\qquad\square$

# B. Main results: Experiments methodology

## B.1. Implemented Objective

In practice, we optimize a scaled bounded-memory variational objective for BiMU, which explicitly balances data fitting, stability to previous parameters, and forgetting toward the prior. Given an online batch $\mathcal{D}_t$, we minimize

$$\mathcal{S}_t(\boldsymbol{\lambda}) = \beta_{\mathrm{L}}\, \mathcal{L}(\boldsymbol{\lambda}, \mathcal{D}_t) + \beta_{\mathrm{KL}} \left[ \left(1 - \frac{1}{N}\right) \mathrm{KL}(q(\boldsymbol{\omega} \mid \boldsymbol{\lambda}) \,\|\, q(\boldsymbol{\omega} \mid \boldsymbol{\lambda}_{t-1})) + \frac{1}{N}\mathrm{KL}(q(\boldsymbol{\omega} \mid \boldsymbol{\lambda}) \,\|\, p(\boldsymbol{\omega})) \right], \tag{60}$$

where $\boldsymbol{\lambda}$ denotes the Bernoulli natural parameters, $\mathcal{L}(\boldsymbol{\lambda}, \mathcal{D}_t) = -\mathbb{E}_{q(\boldsymbol{\omega}|\boldsymbol{\lambda})}[\log p(\mathcal{D}_t \mid \boldsymbol{\omega})]$ is the negative log-likelihood, $p(\boldsymbol{\omega})$ is the prior, $N$ controls the effective memory window, likelihood coefficient $\beta_{\mathrm{L}}$ controls the scale of the asymmetry, and KL coefficient $\beta_{\mathrm{KL}}$ allows for regularization strength.

The second-order approximation of the optimal update of $\mathcal{S}_t$ yields the metaplastic learning rate

$$\boldsymbol{\eta}(\boldsymbol{\lambda}_{t-1}) = \frac{1}{\beta_{\mathrm{KL}}\big(1 - \tanh^2(\boldsymbol{\lambda}_{t-1})\big) + 2\beta_{\mathrm{L}} \tanh(\boldsymbol{\lambda}_{t-1})\, g + 2\beta_{\mathrm{L}}|g| + \alpha_{\max}^{-1}}, \tag{61}$$

where $g = \partial\mathcal{L}(\boldsymbol{\lambda}, \mathcal{D}_t)/\partial\boldsymbol{\lambda}|_{\boldsymbol{\lambda}=\boldsymbol{\lambda}_{t-1}}$ and $\alpha_{\max}$ bounds the maximum step size. We prefer $1 - \tanh^2(\boldsymbol{\lambda}_{t-1})$ to $\cosh^{-2}(\boldsymbol{\lambda}_{t-1})$, which is mathematically equivalent but allows factoring the computation of $\tanh(\boldsymbol{\lambda}_{t-1})$. The resulting online update of the natural parameters is

$$\boldsymbol{\lambda}_t = \boldsymbol{\lambda}_{t-1} - \boldsymbol{\eta}(\boldsymbol{\lambda}_{t-1}) \odot \left[ \gamma\, \beta_{\mathrm{L}}\, g + \frac{\beta_{\mathrm{KL}}}{N}(\boldsymbol{\lambda}_{t-1} - \boldsymbol{\lambda}_{\mathrm{prior}}) \big(1 - \tanh^2(\boldsymbol{\lambda}_{t-1})\big) \right], \tag{62}$$

where $\gamma$ is an added learning rate scaling the gradient, and $\odot$ denotes element-wise multiplication. This formulation is used in all experiments unless stated otherwise.

## B.2. Hyperparameters tuning

Hyperparameter optimization (HPO) is performed using Optuna (Akiba et al., 2019), with 200 optimization trials per method. We define a task-dependent objective $\mathcal{H}$, evaluated at each task, as

$$\mathcal{H}_t = w_1 \cdot \mathrm{Acc}_0 + w_2 \cdot \overline{\mathrm{Acc}} + w_3 \cdot \mathrm{Acc}_t, \tag{63}$$

where $\mathrm{Acc}_0$ is the accuracy on the first task, $\overline{\mathrm{Acc}}$ denotes the mean accuracy over all tasks learned so far, and $\mathrm{Acc}_t$ is the accuracy on the current task. The objective $\mathcal{H}$ jointly captures initial-task retention, overall performance, and adaptability to the most recent task, with their relative importance controlled by weights $(w_1, w_2, w_3)$. Maximizing $\mathcal{H}$ therefore promotes balanced hyperparameter configurations that neither over-emphasize stability nor plasticity, preventing HPO from artificially amplifying catastrophic forgetting or catastrophic remembering in the different algorithms.

## B.3. 1000-tasks Permuted MNIST

**Experimental setup.** We evaluate all methods on the Permuted MNIST benchmark with 1000 sequential tasks, where each task applies a distinct random pixel permutation to the MNIST images, inducing continual distribution shifts. Training follows a fully online protocol: each sample is observed once and no replay is used.

All approaches share a common continual learning setup. We use a compact MLP with a single hidden layer of 100 neurons, train for one epoch per task with batch size 1, and evaluate with batch size 100. Inputs are standardized. Monte Carlo sampling is used for Bayesian methods. BiMU and BayesBiNN employ 5 posterior samples for both inference and gradient estimation, while MESU uses 10 samples. Out-of-distribution detection is evaluated using Fashion-MNIST (Xiao et al., 2017) as OOD data. Performance is measured by ROC-AUC, computed over 1000 decision thresholds to discriminate in-distribution from OOD samples.

**Hyperparameter optimization.**

Hyperparameters are computed on 10 tasks of Permuted MNIST with different permutations as the ones presented in the main paper as validation. Hyperparameters are obtained by maximizing the hyperparameter tuning cost function $\mathcal{H}$ (see Appendix B.2).

*Table 3.* Hyperparameter configurations for all evaluated methods on the 1000-tasks Permuted MNIST benchmark with 100 neurons.

| METHOD | ACTIVATION | LEARNING RATE(S) | ADDITIONAL PARAMETERS | MC SAMPLES |
|---|---|---|---|---|
| *Binary Neural Networks* | | | | |
| BIMU | REVERSE BINARY GATE | $\gamma = 4.9$ | $\alpha_{\text{MAX}} = 0.0023$, $\beta_{\text{L}} = 161.3$, $\beta_{\text{KL}} = 3.76$, $N = 700$ | 5 |
| BAYESBINN | REVERSE BINARY GATE | 0.77 | PRIOR STRENGTH $= 1.25 \times 10^{-5}$ | 5 |
| SYNAPTIC METAPLASTICITY | SIGN | $3.7 \times 10^{-5}$ | METAPLASTICITY $= 7.2$, WEIGHT DECAY $= 2.9 \times 10^{-9}$ | 1 |
| STE (BASELINE) | SIGN | $1 \times 10^{-4}$ | WEIGHT DECAY $= 2.2 \times 10^{-9}$ | 1 |
| *Real-Valued Neural Networks* | | | | |
| MESU | RELU | $\alpha_{\mu} = 1.6$, $\alpha_{\sigma} = 2.1$ | $\sigma_{\text{PRIOR}} = 0.3$, $N = 600{,}000$, CLAMP GRAD $= 0.36$ | 10 |
| ONLINE EWC | RELU | 0.003 | IMPORTANCE $= 4.1$, DOWNWEIGHTING $= 0.7$ | 1 |
| SYNAPTIC INTELLIGENCE | RELU | 0.003 | COEFF. $= 1 \times 10^{-4}$, DAMPING $= 1.0$ | 1 |
| SGD (BASELINE) | RELU | $9 \times 10^{-4}$ | – | 1 |

## B.4. OpenLORIS-Object

**Experimental setup.** We evaluate all methods on the OpenLORIS-Object dataset (She et al., 2020) using the sequential factor analysis protocol. The benchmark comprises 12 sequential tasks defined by four nuisance factors (illumination, occlusion, pixel corruption, and clutter) each evaluated at three difficulty levels, covering 19 object classes. Learning follows a fully online continual setting, where each sample is processed once without replay.

We adopt a frozen ImageNet-pretrained VGG19 (Deng et al., 2009; Simonyan & Zisserman, 2014) network as a feature extractor. Images are preprocessed using the standard VGG19 pipeline, and features are taken from the final convolutional block after removing the classifier, yielding 512×7×7 representations. To study adaptation under constrained capacity, we randomly subsample these features to 8,192 and 1,024 dimensions when specified.

All methods share a unified continual learning configuration. A linear classifier is trained online with no hidden layers, for one epoch per task, using batch size 1 for training and batch size 4 for evaluation. No input normalization is applied. Bayesian methods (MESU, BiMU, and BayesBiNN) use 10 Monte Carlo samples for inference and gradient estimation.

### B.4.1. CONTINUAL LEARNING

**Additional experimental setup.**

We evaluate all algorithms on 18 classes instead of 19. The 'toy' class is removed from the dataset. We evaluate out-of-distribution performance using images taken from the 'toy' class. The ROC-AUC is computed through 1000 threshold points to distinguish between in and out-of-distribution data.

**Hyperparameter optimization.**

Hyperparameters are computed on the validation set of OpenLORIS-Object. Hyperparameters are obtained by maximizing the hyperparameter tuning cost function $\mathcal{H}$ (see Appendix B.2).

*Table 4.* Hyperparameter configurations for OpenLORIS-Object for 25,088, 8,192, and 1,024 input features.

| METHOD | LEARNING RATE(S) | ADDITIONAL PARAMETERS | MC SAMPLES |
|---|---|---|---|
| **25,088 INPUT FEATURES** | | | |
| *Binary Neural Networks* | | | |
| BIMU | $\gamma = 2.16$ | $\alpha_{\text{MAX}} = 0.06$, $\beta_{\text{L}} = 164.9$, $\beta_{\text{KL}} = 0.156$, $N = 6900$ | 10 |
| BAYESBINN | 0.011 | PRIOR STRENGTH $= 1.3 \times 10^{-6}$ | 10 |
| SYNAPTIC METAPLASTICITY | $3 \times 10^{-4}$ | METAPLASTICITY $= 5.6$, WEIGHT DECAY $= 3.3 \times 10^{-12}$ | 1 |
| STE (BASELINE) | $1 \times 10^{-4}$ | – | 1 |
| *Real-Valued Neural Networks* | | | |
| MESU | $\alpha_\mu = 71.91$, $\alpha_\sigma = 0.015$ | $\sigma_{\text{PRIOR}} = 0.69$, $N = 449{,}000$, CLAMP GRAD $= 0.61$ | 10 |
| ONLINE EWC | $3.9 \times 10^{-4}$ | IMPORTANCE $= 0.12$, DOWNWEIGHTING $= 0.03$ | 1 |
| SYNAPTIC INTELLIGENCE | 0.002 | COEFF. $= 7 \times 10^{-5}$, DAMPING $= 12.6$ | 1 |
| SGD (BASELINE) | 0.002 | – | 1 |
| **8,192 INPUT FEATURES** | | | |
| *Binary Neural Networks* | | | |
| BIMU | $\gamma = 33.2$ | $\alpha_{\text{MAX}} = 0.08$, $\beta_{\text{L}} = 187.8$, $\beta_{\text{KL}} = 1.5$, $N = 6700$ | 10 |
| BAYESBINN | 0.86 | PRIOR STRENGTH $= 1 \times 10^{-6}$ | 10 |
| SYNAPTIC METAPLASTICITY | $6 \times 10^{-4}$ | METAPLASTICITY $= 6$, WEIGHT DECAY $= 6.4 \times 10^{-12}$ | 1 |
| STE (BASELINE) | 0.003 | – | 1 |
| *Real-Valued Neural Networks* | | | |
| MESU | $\alpha_\mu = 16.34$, $\alpha_\sigma = 0.04$ | $\sigma_{\text{PRIOR}} = 0.9$, $N = 460{,}000$, CLAMP GRAD $= 0.9$ | 10 |
| ONLINE EWC | 0.001 | IMPORTANCE $= 0.069$, DOWNWEIGHTING $= 0.2$ | 1 |
| SYNAPTIC INTELLIGENCE | 0.001 | COEFF. $= 0.025$, DAMPING $= 0.15$ | 1 |
| SGD (BASELINE) | 0.01 | – | 1 |
| **1,024 INPUT FEATURES** | | | |
| *Binary Neural Networks* | | | |
| BIMU | $\gamma = 0.87$ | $\alpha_{\text{MAX}} = 0.092$, $\beta_{\text{L}} = 27.26$, $\beta_{\text{KL}} = 1.6$, $N = 8900$ | 10 |
| BAYESBINN | 0.3 | PRIOR STRENGTH $= 6.25 \times 10^{-6}$ | 10 |
| SYNAPTIC METAPLASTICITY | $6 \times 10^{-4}$ | METAPLASTICITY $= 2.15$, WEIGHT DECAY $= 5.22 \times 10^{-6}$ | 1 |
| STE (BASELINE) | 0.0035 | – | 1 |
| *Real-Valued Neural Networks* | | | |
| MESU | $\alpha_\mu = 2.96$, $\alpha_\sigma = 11.27$ | $\sigma_{\text{PRIOR}} = 0.92$, $N = 275{,}000$, CLAMP GRAD $= 0.34$ | 10 |
| ONLINE EWC | 0.005 | IMPORTANCE $= 0.028$, DOWNWEIGHTING $= 0.6$ | 1 |
| SYNAPTIC INTELLIGENCE | 0.006 | COEFF. $= 0.003$, DAMPING $= 0.13$ | 1 |
| SGD (BASELINE) | 89.5 | – | 1 |

### B.4.2. IMBALANCED ACTIVE CONTINUAL LEARNING

**Additional experimental setup.**

From the 19 classes of the OpenLORIS-Object dataset (She et al., 2020) under the sequential factor analysis protocol, we designate as low frequency the classes containing fewer than 700 training samples. To induce further class imbalance, we randomly remove between 50% and 80% of the training images from each low frequency class.

For evaluation, the test set is balanced by subsampling all classes to the minimum class cardinality, enabling separate and fair assessment of performance on high and low frequency classes. In the main results, we perform an active learning ablation study, comparing epistemic, aleatoric, predictive, variation ratio (VR), and oracle VR-True strategies across a range of uncertainty thresholds and multiple algorithms.

**Hyperparameter optimization.**

Hyperparameter optimization is conducted without active learning, and the resulting performance is reported as the baseline. Hyperparameters are computed on the validation set of OpenLORIS-Object. They are obtained by maximizing the hyperparameter tuning cost function $\mathcal{H}$ (see Appendix B.2).

*Table 5.* Hyperparameters for algorithms on the OpenLORIS-Object dataset with an imbalanced training set with 8,192 input features.

| METHOD | LEARNING RATE(S) | ADDITIONAL PARAMETERS | MC SAMPLES |
|---|---|---|---|
| *Binary Neural Networks* | | | |
| BIMU | $\gamma = 48.7$ | $\alpha_{\mathrm{MAX}} = 0.065,\ \beta_{\mathrm{L}} = 16.7,\ \beta_{\mathrm{KL}} = 0.53,\ N = 1600$ | 10 |
| BAYESBINN | 0.066 | PRIOR STRENGTH $= 4.6 \times 10^{-5}$ | 10 |
| STE | 0.005 | WEIGHT DECAY $= 5 \times 10^{-7}$ | 1 |
| *Real-valued Neural Networks* | | | |
| MESU | $\alpha_\mu = 60.16,\ \alpha_\sigma = 2.95$ | $\mu_{\mathrm{PRIOR}} = 0,\ \sigma_{\mathrm{PRIOR}} = 0.36,\ N = 328{,}000,\ \mathrm{CLAMP\ GRAD} = 0.7$ | 10 |
| MESU (STANDARDIZED) | $\alpha_\mu = 20.2,\ \alpha_\sigma = 0.005$ | $\mu_{\mathrm{PRIOR}} = 0,\ \sigma_{\mathrm{PRIOR}} = 0.61,\ N = 485{,}000,\ \mathrm{CLAMP\ GRAD} = 0.05$ | 10 |

### B.5. Animals Detection: Imbalanced active learning

In the main text, we evaluate and compare active learning strategies on a subset of the Animals Detection dataset (Jana, 2023). From the 80 available classes, we select 20 classes exhibiting strong class imbalance (see Table 6). Classes with fewer than 200 training samples are considered low frequency. For fair evaluation, we construct a balanced test set by uniformly sampling 50 examples per class, allowing us to explicitly assess the impact of class imbalance on low and high frequency classes.

Images are processed using a frozen VGG19 network (Simonyan & Zisserman, 2014) pretrained on ImageNet (Deng et al., 2009). Standard VGG19 preprocessing is applied, and features are extracted from the final convolutional block after removing the classifier, yielding 512×7×7 representations. All methods share a common experimental configuration. A linear classifier is trained online with no hidden layers, for a single epoch, using batch size 1 for training and batch size 50 for evaluation. No input normalization is applied. Bayesian methods (MESU, BiMU, and BayesBiNN) use 10 Monte Carlo samples for inference and gradient estimation.

*Table 6.* Train/test splits for selected animal classes. Classes under 200 training examples are considered as being under-represented classes.

| CLASS | NUMBER OF TRAIN EXAMPLES | NUMBER OF TEST EXAMPLES |
|---|---|---|
| BUTTERFLY | 1997 | 50 |
| LIZARD | 1412 | 50 |
| FISH | 1404 | 50 |
| MONKEY | 1043 | 50 |
| SPIDER | 1015 | 50 |
| EAGLE | 849 | 50 |
| FROG | 617 | 50 |
| JELLYFISH | 501 | 50 |
| PENGUIN | 390 | 50 |
| WHALE | 291 | 50 |
| ZEBRA | 164 | 50 |
| CROCODILE | 136 | 50 |
| LEOPARD | 132 | 50 |
| SHEEP | 125 | 50 |
| RACCOON | 106 | 50 |
| RAVEN | 91 | 50 |
| PANDA | 62 | 50 |
| LYNX | 66 | 50 |
| BULL | 72 | 50 |
| SCORPION | 76 | 50 |

**Hyperparameter optimization.**

To decouple model selection from the active-learning mechanism, we tune all training hyperparameters once in a fully supervised streaming run (i.e., the model updates on every incoming example; no querying is applied during HPO). In the active-learning runs, we do not retune hyperparameters for any acquisition function or labeling budget: accuracy-budget curves are obtained solely by sweeping the uncertainty threshold $\tau$ with these fixed settings. This protocol ensures that differences between acquisition strategies reflect the querying criterion rather than strategy-specific hyperparameter tuning.

*Table 7.* Baseline hyperparameter configurations for the Animals Detection dataset with imbalanced classes.

| METHOD | LEARNING RATE(S) | ADDITIONAL PARAMETERS | MC SAMPLES |
|---|---|---|---|
| *Binary Neural Networks* | | | |
| BIMU | $\gamma = 5.8$ | $\alpha_{\text{MAX}} = 0.054$, $\beta_{\text{L}} = 14$, $\beta_{\text{KL}} = 0.16$, $N = 4600$ | 10 |
| BAYESBINN | 10 | PRIOR STRENGTH $= 1 \times 10^{-6}$ | 10 |
| STE | 0.0001 | WEIGHT DECAY $= 2.4 \times 10^{-9}$ | 1 |
| *Real-Valued Neural Networks* | | | |
| MESU | $\alpha_{\mu} = 42.4$, $\alpha_{\sigma} = 97.7$ | $\sigma_{\text{PRIOR}} = 0.56$, $N = 61{,}000$, CLAMP GRAD $= 0.1$ | 10 |
| MESU (STANDARDIZED) | $\alpha_{\mu} = 65.6$, $\alpha_{\sigma} = 4.7$ | $\sigma_{\text{PRIOR}} = 0.86$, $N = 397{,}000$, CLAMP GRAD $= 0.41$ | 10 |

## C. Backward transfer clarifications

Backward transfer (BWT) (Lopez-Paz & Ranzato, 2017) is a standard metric for quantifying forgetting in continual learning. If $a_{t,i}$ denotes the test accuracy on task $i$ after training up to task $t$, a common definition is

$$\text{BWT} = \frac{1}{T-1} \sum_{i=1}^{T-1} \left( a_{T,i} - a_{i,i} \right). \tag{64}$$

BWT measures how much performance on previously learned tasks changes after subsequent training. A negative value indicates stronger forgetting, while a value close to zero is usually interpreted as better backward stability. Positive values indicate positive transfer.

The goal of our work is to both mitigate catastrophic forgetting of old tasks, as well as progressive loss of plasticity due to posterior saturation on new tasks. In Binary networks, the latent variable (or natural parameter) can grow in magnitude over a long stream, driving synapses to saturation. Once this happens, weights are nearly deterministic and sign changes become increasingly unlikely.

BWT is conditional on the amount of task-specific performance that was acquired in the first place. A method that learns a task well and later forgets it will have a negative BWT. In contrast, a method that fails to learn, or that becomes rigid and remains uniformly poor, can obtain a near-zero BWT simply because there is little performance left to lose. Therefore, on very long non-stationary streams, BWT can confound two qualitatively different regimes: useful stability after learning, and apparent stability caused by loss of plasticity.

*Table 8.* BWT on the 1000-tasks Permuted-MNIST stream. Synaptic Metaplasticity has the least negative BWT, but it reaches only chance-level accuracy. BiMU has a more negative BWT, but remains highly plastic and accurate after 1000 tasks.

| METHOD | MEAN ACC. 5 TASKS (%) | MMRR | BWT |
|---|---|---|---|
| BIMU | **90.30 ± 0.38** | **139.47** | $-0.8187 \pm 0.0014$ |
| BAYESBINN | $41.12 \pm 1.62$ | 2.04 | $-0.3897 \pm 0.0021$ |
| SYNAPTIC METAPLASTICITY | $10.27 \pm 0.01$ | 1.64 | $\mathbf{-0.0009 \pm 0.0001}$ |
| STE | $29.35 \pm 0.96$ | 9.32 | $-0.5689 \pm 0.0105$ |

In the main paper, we report the mean accuracy on the last five tasks and Maximum Memory Rigidity Resilience (MMRR) instead of BWT, which appears in Table 8. Late-stream accuracy directly measures whether the model can still acquire new tasks after a long history of distribution shifts. MMRR complements this by measuring how much of the method's best attainable performance is preserved at the end of the stream. Together, these quantities distinguish a model that remains both stable and plastic from one that stops changing.

Specifically, Synaptic Metaplasticity has the least negative BWT among the binary methods. However, it is the weakest method at the end of the stream, with only $10.27\%$ mean accuracy over the last five tasks. This value is close to chance for MNIST classification, showing that the near-zero BWT neither reflects good performance in continual learning nor how much loss of plasticity occurred in the model.

This supplementary analysis emphasizes that a better BWT is not necessarily a good sign of well-performing continual learning, specifically for our use-case. When posterior saturation or synaptic rigidity prevents new learning, BWT can become artificially favorable. To solve the challenges we presented in the introduction, the relevant question is whether the method can still learn after hundreds or thousands of distribution shifts. Late-stream accuracy and MMRR directly answer this question, while BWT only measures one aspect of backward stability.

The OpenLORIS-Object results provide a complementary case, due to not facing the same rigidity as for the 1000-tasks Permuted MNIST. As shown in Table 9, BiMU achieves the highest mean accuracy, $90.62\%$, while having BWT comparable to BayesBiNN, Synaptic Metaplasticity and within range of other continual learning methods. STE is worse both in accuracy and in BWT.

*Table 9.* OpenLORIS-Object with 25,088 frozen VGG19 features. We report the mean accuracy over all tasks and backward transfer (BWT).

| METHOD | MEAN ACC. OVER ALL TASKS (%) | BWT |
|---|---|---|
| *Binary Neural Networks* | | |
| BIMU | **90.62 ± 0.22** | −0.0525 ± 0.0027 |
| BAYESBINN | 89.37 ± 0.77 | **−0.0399 ± 0.0078** |
| SYNAPTIC METAPLASTICITY | 86.72 ± 0.34 | −0.0485 ± 0.0021 |
| STE (BASELINE) | 83.79 ± 1.13 | −0.0790 ± 0.0157 |
| *Real-Valued Neural Networks* | | |
| MESU | 87.84 ± 0.11 | −0.0515 ± 0.0012 |
| ONLINE EWC | 88.23 ± 0.05 | −0.0452 ± 0.0003 |
| SYNAPTIC INTELLIGENCE | 88.04 ± 0.03 | −0.0517 ± 0.0004 |
| SGD (BASELINE) | 88.04 ± 0.03 | −0.0517 ± 0.0004 |

So far, this is consistent with our main claim: BiMU deliberately forgets part of the accumulated posterior information in order to avoid rigidity and preserve plasticity. If the information is not useful currently and lies outside of the memory window $N$, it is discarded. A loss in backward transfer is exchanged for higher overall accuracy on late stream: it tends to indicate that the method discards non-essential or outdated representations.

## D. Maximum Memory Rigidity Resilience (MMRR): A measure of plasticity loss

In long-horizon continual learning, models may progressively lose plasticity as evidence accumulates, becoming increasingly resistant to parameter updates. This rigidity can degrade performance on new tasks even in the absence of explicit forgetting. Standard metrics such as average accuracy (ACC), Forward Transfer (FWT), or Backward Transfer (BWT) (Lopez-Paz & Ranzato, 2017) do not explicitly capture this phenomenon, as they focus on task-to-task retention rather than long-term adaptability under prolonged non-stationarity.

To quantify plasticity loss, we build on the Memory Rigidity Resilience (MRR) metric of (Bonnet et al., 2025) and introduce Maximum Memory Rigidity Resilience (MMRR), a scalar diagnostic that measures performance degradation relative to the model's own best observed performance.

Let $a_t \in [0, 1]$ denote the accuracy of the model evaluated on task $t$, and let

$$a_{\max} = \max_{t' \leq T} a_{t'}$$

be the maximum accuracy achieved by the model on any task over the full training horizon $T$. We define the Maximum Memory Rigidity Resilience (MMRR) at time $t$ as

$$\mathrm{MMRR}_t = \frac{1}{a_{\max} - a_t + \varepsilon},$$

where $\varepsilon > 0$ ensures numerical stability. Larger values indicate smaller performance degradation and thus higher resistance to rigidity.

MMRR captures the inability of a model to sustain its peak performance as new tasks arrive, independently of explicit forgetting. Unlike the original MRR, which anchors rigidity to early-task performance and can be misleading when performance improves over time, MMRR evaluates each model relative to its maximum attainable accuracy over the stream. This ensures stability and interpretability in long-horizon settings.

In our experiments, MMRR clearly separates methods that maintain long-term adaptability from those that become rigid: BayesBiNN, Synaptic Metaplasticity, and STE exhibit rapid MMRR collapse due to posterior saturation, while BiMU maintains high MMRR over extended non-stationary streams, indicating sustained plasticity.

## E. 1000-tasks Permuted MNIST: Effect of Network Capacity

This appendix investigates whether BiMU's advantages persist when network capacity is substantially increased. While the main paper focuses on compact architectures to emphasize bounded-memory effects, scaling the model allows us to disentangle representational limitations from loss of plasticity and synaptic rigidity.

We repeat the 1000-tasks Permuted MNIST experiment using a single-hidden-layer network with 2000 neurons, keeping the online continual learning protocol unchanged. Training is strictly online, with each sample observed once. Inputs are standardized, training uses batch size 1, evaluation uses batch size 100, and out-of-distribution (OOD) detection is assessed on Fashion-MNIST using ROC-AUC over 1000 thresholds. Bayesian methods employ Monte Carlo sampling during inference: BiMU and BayesBiNN use 5 samples, while MESU uses 10. All other methods are deterministic.

Table 10 reports results averaged over the last five tasks after training on all 1000 permutations. Increasing capacity improves accuracy for all methods, confirming that additional representational capacity partly mitigates long-horizon continual learning challenges.

BiMU benefits most from scaling, achieving a mean accuracy of 95.20%, the best performance among binary-weight models and higher than all real-valued baselines. In contrast, other binary approaches fail to fully exploit the additional capacity: BayesBiNN improves relative to the low-capacity setting but remains constrained by progressive rigidity, STE shows only moderate gains, and Synaptic Metaplasticity collapses to near-chance performance. This indicates that capacity alone is insufficient without mechanisms that actively regulate plasticity.

Among real-valued models, scaling preserves the relative performance trends observed in smaller networks. MESU achieves strong results (92.99%), followed by EWC Online and Synaptic Intelligence. Notably, BiMU surpasses all real-valued baselines despite operating under a more constrained synaptic posterior, highlighting that its advantage stems from synapse-level control of learning dynamics rather than architectural flexibility.

The Maximum Memory Rigidity Resilience (MMRR) metric further emphasizes these differences. BiMU achieves the highest resilience (862.09), indicating minimal degradation between early and late tasks and sustained plasticity across the entire stream. All other methods exhibit substantially lower resilience, confirming that they progressively lose adaptability even with increased capacity.

Scaling also accentuates differences in uncertainty estimation. BiMU achieves perfect OOD discrimination on Fashion-MNIST (AUC = 1.00), while MESU remains strong but inferior (AUC = 0.86), and other baselines lag behind. This demonstrates that BiMU preserves meaningful epistemic uncertainty over extremely long task horizons, even as capacity increases.

*Table 10.* Benchmark on 1000 tasks Permuted MNIST in the Online Continual Learning context (2000 neurons). The mean accuracy is taken after training on all tasks, computing the average of the accuracy on the last five tasks test datasets. Memory rigidity resilience is defined as the inverse of the absolute differential in accuracy between the first and the last task. Mean ± standard deviation are given over five runs. Methods are grouped by whether they use binary weights or not.

| METHOD | MEAN ACC. 5 TASKS (%) | OOD DET. (AUC) | MMRR |
|---|---|---|---|
| *Binary neural networks* | | | |
| BiMU | **95.20 ± 0.26** | **1.00 ± 0.00** | **862.09** |
| BAYESBiNN | 86.61 ± 0.27 | 0.80 ± 0.10 | 9.86 |
| SYN. META. | 10.34 ± 0.19 | 0.00 ± 0.04 | 1.33 |
| STE | 47.39 ± 1.35 | 0.61 ± 0.10 | 39.12 |
| *Real-valued neural networks* | | | |
| MESU | **92.99 ± 0.71** | 0.86 ± 0.02 | **171.82** |
| EWC O. | 91.47 ± 0.31 | 0.44 ± 0.14 | 139.66 |
| SI | 92.82 ± 0.49 | 0.96 ± 0.04 | 89.77 |
| SGD | 90.61 ± 1.32 | **0.98 ± 0.02** | 117.65 |

**Hyperparameter optimization.**

Hyperparameters are computed on 10 tasks of Permuted MNIST with different permutations as the ones presented in the main paper as validation. They are obtained by maximizing the hyperparameter tuning cost function $\mathcal{H}$ (see Appendix B.2).

*Table 11.* Hyperparameter configurations for all evaluated methods on the 1000-tasks Permuted MNIST benchmark with 2000 neurons.

| METHOD | ACTIVATION | LEARNING RATE(S) | ADDITIONAL PARAMETERS | MC SAMPLES |
|---|---|---|---|---|
| *Binary Neural Networks* | | | | |
| BiMU | REVERSE BINARY GATE | 60 | $\alpha_{\text{MAX}} = 0.005$, $\beta_{\text{L}} = 280.7$, $\beta_{\text{KL}} = 6.2$, $N = 900$ | 5 |
| BAYESBiNN | REVERSE BINARY GATE | 0.05 | PRIOR STRENGTH $= 1.1 \times 10^{-6}$ | 5 |
| SYNAPTIC METAPLASTICITY | SIGN | $1 \times 10^{-5}$ | METAPLASTICITY $= 6.3$, WEIGHT DECAY $= 1.46 \times 10^{-6}$ | 1 |
| STE (BASELINE) | SIGN | $1 \times 10^{-4}$ | WEIGHT DECAY $= 9.7 \times 10^{-5}$ | 1 |
| *Real-Valued Neural Networks* | | | | |
| MESU | ReLU | $\alpha_\mu = 2.2$, $\alpha_\sigma = 0.47$ | $\sigma_{\text{PRIOR}} = 0.46$, $N = 1.6 \times 10^6$, CLAMP GRAD $= 0.83$ | 10 |
| ONLINE EWC | ReLU | 0.006 | IMPORTANCE $= 4.0$, DOWNWEIGHTING $= 0.4$ | 1 |
| SYNAPTIC INTELLIGENCE | ReLU | 0.0015 | COEFF. $= 1 \times 10^{-4}$, DAMPING $= 1.0$ | 1 |
| SGD (BASELINE) | ReLU | 0.001 | – | 1 |

# F. 1000-tasks Permuted MNIST: Training-State Memory Overhead

Table 12 reports the training-state memory overhead of all methods evaluated on the 1000-tasks Permuted MNIST benchmark. We measure the total persistent memory required during training, including model parameters and all auxiliary variables needed to perform parameter updates, excluding activations, capturing the dominant constraints in long-horizon learning scenarios.

BiMU is the only continual learning method among the others that is fully online, not requiring stored past parameters, task-specific statistics, replay buffers, or optimizer momentum. Its updates rely solely on the current Bernoulli posterior parameters and the incoming batch, so the training-state memory footprint is identical to the inference-state memory footprint and remains constant over time.

In the binary setting, BayesBiNN stores both Bernoulli parameters and associated the posterior of the previous task, resulting in a larger memory overhead than BiMU. STE and Synaptic Metaplasticity are trained with Adam, which requires storing two additional momentum matrices per parameter. In addition, Synaptic Metaplasticity maintains task-specific Batch Normalization parameters, which accumulate linearly with the number of tasks and dominate the memory overhead in long-horizon regimes.

In the real-valued setting, MESU attaches an explicit variance parameter to each synapse, increasing the memory footprint relative to SGD. EWC Online and Synaptic Intelligence require storing both an importance matrix and a copy of the parameters from the previous task, resulting in persistent additional storage even in their online variants.

*Table 12.* Training-state memory overhead on the 1000-tasks Permuted MNIST benchmark. Reported values correspond to the total persistent memory required during training, including model parameters and all auxiliary update variables, but excluding transient activations.

| METHOD | TRAINING-STATE MEMORY OVERHEAD (MB) |
|---|---|
| *Binary neural networks* | |
| BiMU | **0.32** |
| BAYESBiNN | 0.64 |
| SYN. META. | 1.84 |
| STE | 0.95 |
| *Real-valued neural networks* | |
| MESU | 0.64 |
| EWC O. | 0.95 |
| SI | 0.95 |
| SGD | **0.32** |

## G. 100-tasks Permuted MNIST: Effect of the memory window on stability and forgetting

### G.1. Forgetting, rigidity, controlled regimes

Consider BiMU's update of the natural parameters $\lambda_t^{(i)}$ defined in Eq. (6), obtained from the bounded-memory Bayesian objective with memory window $N$. For a fixed data stream and learning-rate bound $\alpha_{\max}$, the memory window $N$ explicitly controls the forgetting-stability trade-off as follows.

**Small memory window (low $N$): forgetting-dominated regime.** When $N$ is small, the forgetting term

$$\frac{\lambda_{t-1}^{(i)} - \lambda_{\text{prior}}^{(i)}}{N \cosh^2(\lambda_{t-1}^{(i)})}$$

rapidly relaxes the parameters toward the prior. Past evidence is, therefore, quickly discarded, preventing sustained accumulation of information in $\lambda^{(i)}$ and leading to excessive forgetting.

**Large memory window (high $N$): stability-dominated regime.** As $N$ increases, the forgetting term vanishes, and updates approach cumulative Bayesian learning. In this regime, the magnitude of $\lambda^{(i)}$ grows monotonically as evidence accumulates, driving Bernoulli probabilities toward deterministic values. This induces synaptic degeneracy and increases memory rigidity, reflected by low MMRR values, inducing a situation akin to the one observed on BayesBiNN.

**Intermediate memory window: controlled forgetting.** For intermediate values of $N$, forgetting counterbalances evidence accumulation, bounding the magnitude of $\lambda^{(i)}$ over time. This prevents synaptic degeneracy while preserving sufficient information for learning, yielding a favorable trade-off between plasticity and stability.

Overall, the memory window $N$ acts as a direct control parameter for forgetting, regulating both the long-term magnitude of the synaptic parameters and the degree of rigidity in the learned representation.

### G.2. Memory window ablation study

Table 13 quantitatively illustrates the effect of the memory window $N$ on forgetting, memory rigidity, and uncertainty in 100-tasks Permuted-MNIST. For very small $N = 100$, the model falls into the forgetting-dominated regime: final accuracy is only 27.09%, OOD detection remains near-perfect (0.99), and MMRR is high (216.45), indicating rapid weight fluctuations induced by forgetting, unstable posterior retention circumventing stability. Increasing $N$ to 400 allows more evidence accumulation, improving accuracy to 73.46%, with OOD AUC stable (1.00) and a high MMRR (574.72), reflecting higher retention than $N = 100$ without accumulating rigidity.

Intermediate windows ($N = 700$–$1000$) achieve the best balance: accuracy peaks at 90.29% ($N = 700$) while OOD detection remains strong (0.99–0.98) and MMRR is moderate (215.52–46.30), consistent with controlled forgetting that stabilizes synapses without compromising plasticity. For larger windows ($N = 1300$–$1900$), final accuracy slightly declines (86.55%–87.67%) while MMRR steadily decreases to 28.36–21.01, indicating increased rigidity, and slower adaptation to

recent tasks.

Finally, a large memory window ($N = 100,000$) leads to further drops in accuracy, at about $83.70\%$ and OOD AUC (0.94), with minimal MMRR (13.13), consistent with synaptic degeneracy.

Overall, Table 13 confirms that $N$ acts as a direct control mechanism over forgetting and rigidity: small $N$ causes rapid forgetting, very large $N$ induces rigidity and synaptic degeneracy, and intermediate $N$ preserves a favorable stability-plasticity trade-off while maintaining informative epistemic uncertainty. Contrary to MESU (Bonnet et al., 2025) that suggests using $N$ as the number of batches, here the exponential dependency prevents from asserting a similar statement.

*Table 13.* Effect of the memory window $N$ of BiMU, and of BayesBiNN on forgetting, rigidity, and uncertainty in 100-tasks Permuted-MNIST (OCL; 100-hidden-unit MLP). MEAN ACC. (last 5) is averaged over the final five tasks. OOD AUC is ROC-AUC for Permuted MNIST vs Fashion-MNIST. MMRR: Maximum Memory Rigidity Resilience (see main text). Mean±std over 5 runs.

| MEMORY WINDOW $N$ | MEAN ACC. 5 TASKS (%) | OOD DET. (AUC) | MMRR |
|---|---|---|---|
| 100 | $27.09 \pm 0.57$ | $0.99 \pm 0.00$ | 216.45 |
| 400 | $73.46 \pm 2.59$ | $\mathbf{1.00 \pm 0.00}$ | **574.72** |
| 700 | $\mathbf{90.29 \pm 0.24}$ | $0.99 \pm 0.01$ | 215.52 |
| 1000 | $89.07 \pm 0.28$ | $0.98 \pm 0.01$ | 46.30 |
| 1300 | $87.67 \pm 0.33$ | $0.97 \pm 0.02$ | 28.36 |
| 1600 | $86.94 \pm 0.29$ | $0.96 \pm 0.02$ | 24.91 |
| 1900 | $86.55 \pm 0.24$ | $0.96 \pm 0.02$ | 21.01 |
| 100000 | $83.70 \pm 0.30$ | $0.94 \pm 0.05$ | 13.13 |
| BAYESBINN | $67.91 \pm 0.96$ | $0.75 \pm 0.20$ | 5.03 |

We further illustrate the effect of the memory window $N$ on synaptic weight distributions at the end of training. Figure 7 shows the histograms of the probability of each synapse for all values of $N$. Small $N$ results in broad, low-confidence distributions, reflecting rapid forgetting without stability. Intermediate $N$ produces well-formed, moderately peaked distributions, indicating a favorable balance between plasticity and stability. Very large $N$ leads to sharply peaked distributions near 0 or 1, highlighting distribution degeneracy and loss of plasticity, a behaviour also observed on BayesBiNN due to the absence of forgetting for the algorithm.

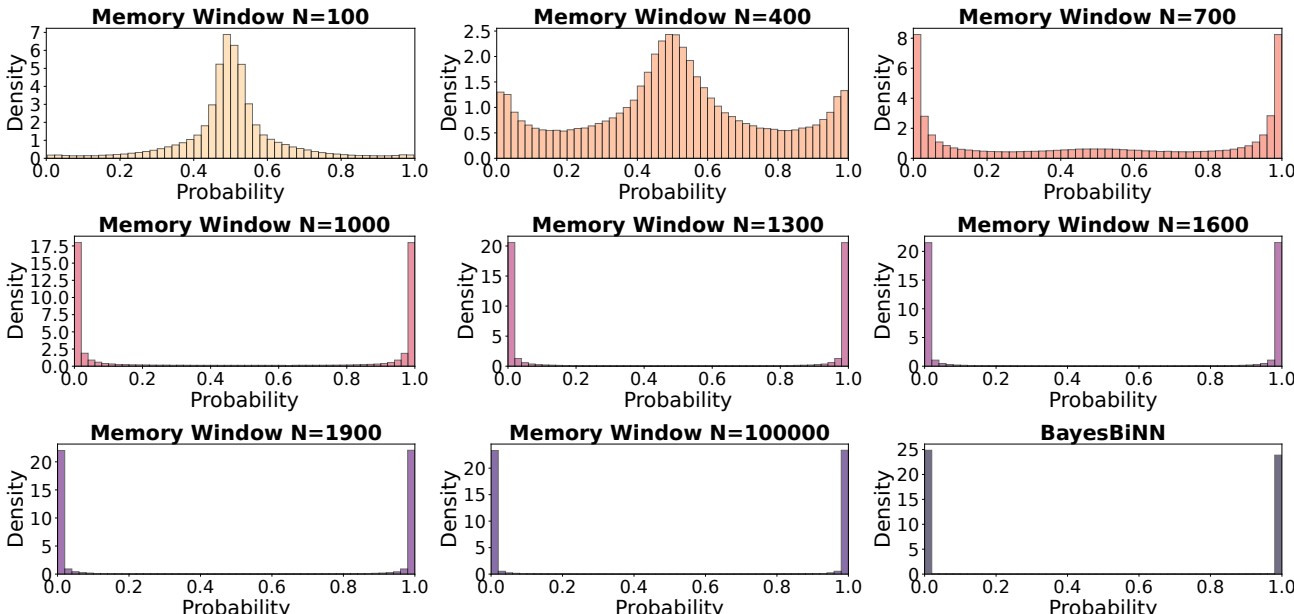

*Figure 7.* Histograms of the synaptic probabilities $\boldsymbol{p} = \text{sigmoid}(2\boldsymbol{\lambda})$ at the end of training for different memory windows $N$ and BayesBiNN in 100-tasks Permuted-MNIST (OCL; 100-hidden-unit MLP). Small $N$ shows broad distributions indicative of forgetting-dominated dynamics, intermediate $N$ yields moderate peaks reflecting a balanced stability-plasticity-forgetting trade-off, and very large $N$ exhibits sharp peaks near 0 and 1, consistent with synaptic degeneracy and rigidity, highlighting too high stability, akin to BayesBiNN's behaviour.

## H. 100-tasks Permuted MNIST: Activation function ablation study

Binary Neural Networks (BiNNs) traditionally rely on the sign activation function (Hubara et al., 2016; Laborieux et al., 2021), together with a surrogate hardtanh during backpropagation to overcome the non-differentiability of the binary non-linearity:

$$\text{sign}(x) = \begin{cases} 1 & \text{if } x > 0, \\ -1 & \text{if } x < 0, \\ 0 & \text{otherwise,} \end{cases} \qquad \text{hardtanh}(x) = \begin{cases} 1 & \text{if } x > 1, \\ x & \text{if } -1 \le x \le 1, \\ -1 & \text{if } x < -1. \end{cases} \tag{65}$$

In this setting, neurons are activated according to the *sign* of their pre-activations. Combined with instance normalization (Ulyanov et al., 2016), gradients are only propagated through neurons whose pre-activations lie in the narrow neutral region $[-1, 1]$. Consequently, learning is dominated by the most frequent pre-activation values, which can be suboptimal in continual learning scenarios where selective and stable parameter updates are required.

**Reverse Binary Gate.** We introduce the Reverse Binary Gate (RBG) activation function and its surrogate sRBG. Unlike the sign function, RBG selects neurons based on the *magnitude* of their pre-activations rather than their sign:

$$\text{RBG}(x, a) = \begin{cases} 0 & \text{if } |x| < \frac{a}{2}, \\ 1 & \text{otherwise,} \end{cases} \qquad \text{sRBG}(x, a) = \begin{cases} -x & \text{if } -\frac{3a}{2} < x < -\frac{a}{2}, \\ x & \text{if } \frac{a}{2} < x < \frac{3a}{2}, \\ 0 & \text{if } |x| < \frac{a}{2}, \\ 1 & \text{otherwise.} \end{cases} \tag{66}$$

This design inverts the usual gating logic: neurons with the most frequent activations are suppressed, while only the moderate to low activations contribute to the forward and backward passes. As a result, a majority of synapses are explicitly

deactivated, promoting sparse and selective updates. The RBG function can be interpreted as a reversed version of the Elephant activation (Lan & Mahmood, 2024) in the limit case.

**Disentangling activation functions from continual learning.** Table 14 reports an ablation study on 100-tasks Permuted MNIST that isolates the effect of the activation function from the continual learning mechanism. By comparing sign-based and RBG-based variants of the same algorithms, we disentangle the respective contributions of activation design and learning rules for the Permuted MNIST experiment.

*Table 14.* Disentangling the effect of the activation function on continual learning performance (100-tasks Permuted MNIST). Mean accuracy is computed over the last five tasks; OOD detection is ROC-AUC for Permuted MNIST vs Fashion-MNIST. Maximum Resilience corresponds to the Maximum Memory Rigidity Resilience (MMRR). Mean±std over 5 runs.

| METHOD | ACTIVATION | MEAN ACC. 5 TASKS (%) | OOD DET. (AUC) | MMRR | ACC. 1 TASK (%) |
|---|---|---|---|---|---|
| BAYESBiNN | SIGN | $66.40 \pm 0.97$ | $0.54 \pm 0.19$ | 5.85 | $90.27 \pm 0.32$ |
| BAYESBiNN | RBG | $67.41 \pm 1.03$ | $0.76 \pm 0.17$ | 4.99 | $93.22 \pm 0.09$ |
| BiMU | SIGN | $81.78 \pm 0.58$ | $0.76 \pm 0.08$ | 22.25 | $92.89 \pm 0.22$ |
| BiMU | RBG | $90.29 \pm 0.24$ | $0.99 \pm 0.01$ | 215.52 | $94.67 \pm 0.11$ |
| MESU | ReLU | $93.51 \pm 0.18$ | $0.91 \pm 0.03$ | 500.02 | $96.10 \pm 0.18$ |
| MESU | RBG | $92.35 \pm 0.21$ | $0.80 \pm 0.05$ | 943.44 | $93.01 \pm 0.20$ |
| SYN. META. | SIGN | $10.22 \pm 0.05$ | - | 9.92 | $71.40 \pm 1.48$ |
| SYN. META. | RBG | $11.35 \pm 0.00$ | - | 26.29 | $13.20 \pm 1.38$ |

To isolate the impact of the activation function from long-horizon continual learning effects, we provide an ablation study conducted on 100-tasks Permuted MNIST. This reduced setting compared to the main enables a clearer separation between optimization effects induced by the activation function and degradation caused by prolonged memory saturation.

The results in Table 14 reveal heterogeneous sensitivities to the choice of activation across methods. For BayesBiNN, replacing the sign activation with the Reverse Binary Gate yields a small but consistent improvement in continual performance: mean accuracy over the last five tasks increases from $66.40\%$ to $67.41\%$, while OOD detection improves markedly from $0.54$ to $0.76$ ROC-AUC. One-task accuracy also increases from $90.27\%$ to $93.22\%$, indicating that RBG provides a stronger single-task accuracy. However, memory resilience slightly decreases ($5.85$ to $4.99$), suggesting that while magnitude-based gating improves representation quality, it does not fundamentally resolve the rigidity and plasticity limitations of BayesBiNN.

In contrast, BiMU benefits substantially from the Reverse Binary Gate. Switching from sign to RBG increases the mean accuracy from $81.78\%$ to $90.29\%$, improves OOD detection from $0.76$ to $0.99$ ROC-AUC, and dramatically boosts MMRR from $22.25$ to $215.52$. Importantly, one-task accuracy also improves from $92.89\%$ to $94.67\%$, confirming that the gains are not solely due to reduced forgetting but to which neurons are going to be activated during the forward and backward passes. These results indicate that selecting neurons based on activation magnitude rather than sign leads to more stable Bayesian updates and mitigates interference between tasks.

For MESU, the activation change does not provide a clear advantage. While RBG substantially increases memory rigidity ($500.02$ to $943.44$), it degrades OOD detection ($0.91$ to $0.80$ ROC-AUC), slightly reduces mean continual accuracy ($93.51\%$ to $92.35\%$), and lowers one-task accuracy from $96.10\%$ to $93.01\%$. This indicates that the Reverse Binary Gate increases rigidity in real-valued networks, harming both generalization and predictive confidence. As a result, RBG does not replace the role of structured Bayesian updates in MESU.

Finally, for Synaptic Metaplasticity, one-task accuracy collapses from $71.40\%$ to $13.20\%$. This highlights a fundamental incompatibility between the method and magnitude-based gating, exacerbating known sensitivities to network size, batching, and training regime.

Overall, this ablation demonstrates that selecting synapses based on activation magnitude rather than sign can substantially improve both single-task performance and continual learning robustness in Bayesian binary neural networks, particularly for BiMU. The effect is strongly method-dependent: it is moderate for BayesBiNN, ineffective for MESU and Synaptic Metaplasticity. While it improves performance, it does not prevent memory rigidity nor does it work better than a ReLU for real-valued networks.

# I. Animals Imbalanced Active Learning: Complementary Figures

Complementary to the main text, we provide additional figures illustrating the behavior of each acquisition strategy on the Animals benchmark under imbalanced active learning. These figures correspond to the aggregated results reported in Fig. 4 and are presented here to facilitate a more detailed, per-method analysis. Certain strategies are not appearing on figures due to the threshold not being activated, very small measures of uncertainty or points overlapping.

Fig. 8 reports the performance of BayesBiNN on the Animals dataset. All acquisition strategies perform below random sampling, indicating that the uncertainty estimates are not sufficiently informative to support effective active learning. Unlike settings with a very large number of tasks such as Permuted MNIST, where weight divergence naturally emerges, the Animals dataset contains relatively few training examples per task (10,549 images in total, corresponding to roughly one sixth of a MNIST task). As a result, the model's weights do not have sufficient opportunity to diverge across tasks. Moreover, in the absence of a metaplastic learning rate mechanism, BayesBiNN is unable to learn efficiently enough from these limited examples to consolidate its weights. This prevents the model from reliably distinguishing between low-frequency and high-frequency samples. While this leads to high overall uncertainty and consequently strong out-of-distribution detection, the quality of the uncertainty estimates remains poor, as predictions are primarily influenced by the most recently observed samples rather than by the dataset as a whole, in contrast to methods incorporating metaplasticity such as BiMU.

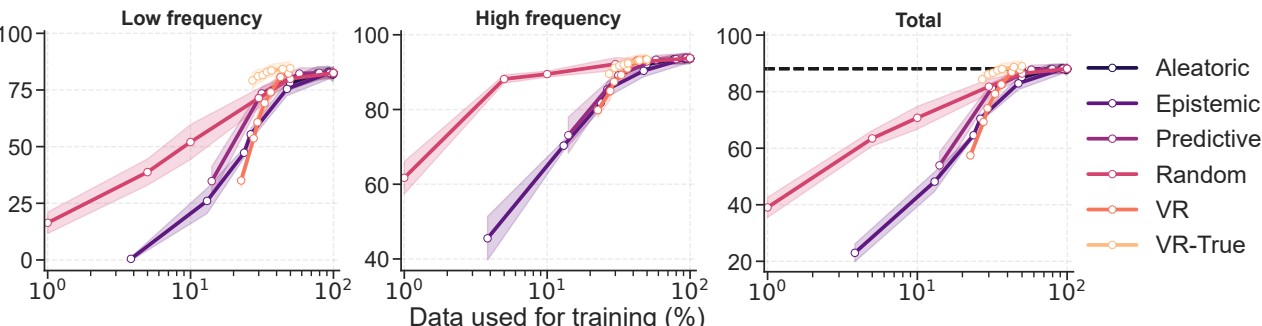

*Figure 8.* Animals dataset under imbalanced active learning using BayesBiNN. Classification accuracy as a function of the fraction of queried samples, highlighting the behavior of Bayesian binary weights under uncertainty-driven querying.

Fig. 9 and Fig. 10 show the results for MESU, without and with input standardization, respectively. Without standardization, all querying strategies other than VR and VR-True collapse early in training, reflecting the inability to learn with low amount of data and poorly conditioned features. While both VR variants benefit from uncertainty-driven sampling, standardization markedly improves performance, yielding lower data used for training and more balanced exploration of minority classes, consistently outperforming random acquisition. Among all criteria, epistemic uncertainty achieves the strongest trade-off between data and accuracy, reaching the random-query baseline with the smallest fraction of labeled samples. This confirms that when uncertainty estimates are well calibrated, they enable highly selective supervision, and further highlights the sensitivity of real-valued Bayesian methods such as MESU to input preprocessing.

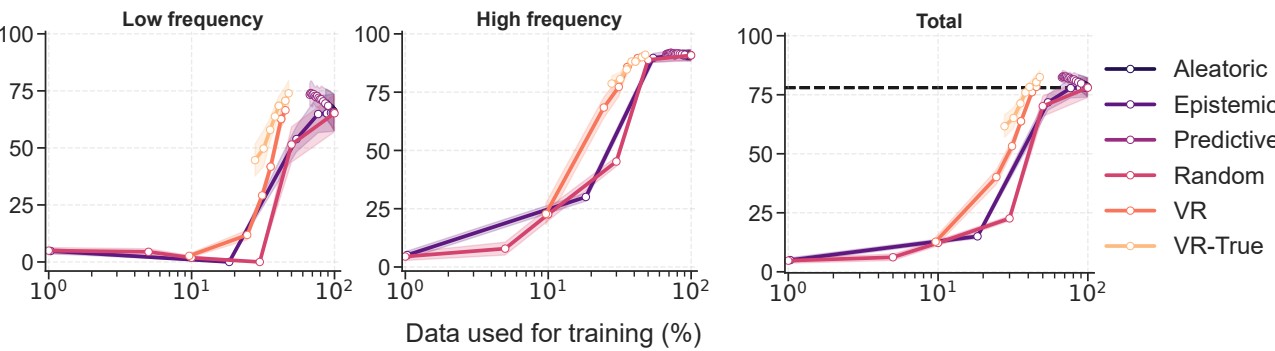

*Figure 9.* Animals dataset under imbalanced active learning using MESU. Performance as a function of the queried-label fraction, illustrating uncertainty-based acquisition with real-valued Bayesian Gaussian weights.

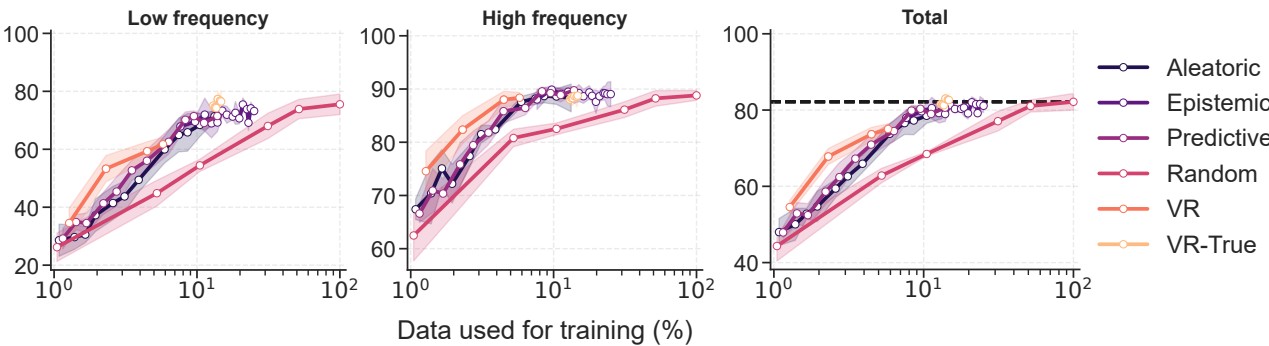

*Figure 10.* Animals dataset under imbalanced active learning using MESU with feature standardisation. Standardisation improves stability and performance across labeling budgets.

Finally, Fig. 11 illustrates the behavior of STE. STE achieves competitive early performance, its acquisition strategy exhibits higher sensitivity to class imbalance at low percentage of labeled data, resulting in consistent drops in accuracy when threshold are increased, leading to a poor accuracy/labeled data ratio compared to BiMU and MESU.

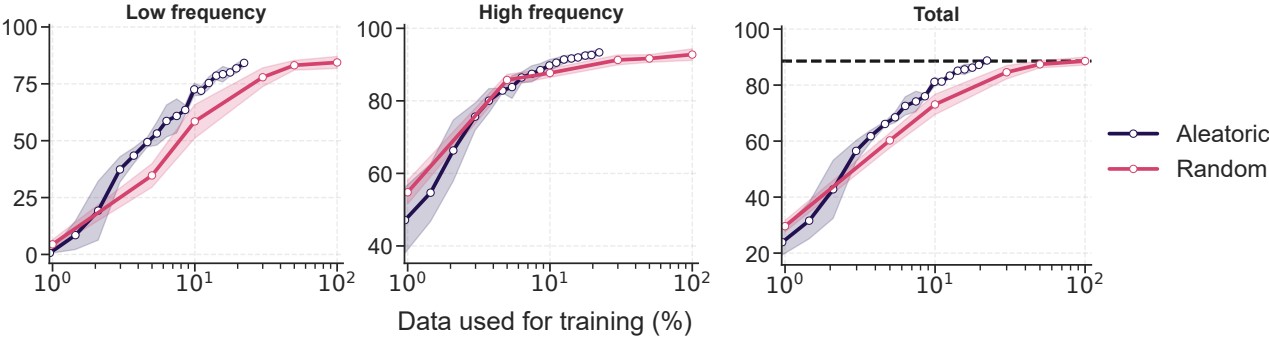

*Figure 11.* Animals dataset under imbalanced active learning using STE. Accuracy as a function of the queried-label fraction when acquisition is driven by aleatoric uncertainty in binary weights setup.

Overall, these complementary figures highlight qualitative differences in acquisition dynamics that are not fully captured by summary metrics alone, especially for BayesBiNN, which, while having reliable OOD detection estimates, does not capture

as well what examples are important and which are not.

## J. OpenLORIS-Object: Effect of standardization on the output features of VGG19

The goal of this appendix experiment is to quantify how much of the performance gap observed in the main results between Bayesian binary algorithms and real-valued algorithms is attributable to input preprocessing rather than to the learning algorithms themselves. To this end, we repeat the OpenLORIS-Object experiment using standardized 8,192-dimensional VGG19 features, where each feature is normalized using the mean and variance estimated from the training data. This contrasts with the streaming setting of Table 2, where no such normalization is applied.

*Table 15.* OpenLORIS-Object (OCL; linear head) with standardized 8,192-dimensional VGG19 features. Mean accuracy is averaged over the 12 tasks after training on all tasks. OOD AUC is computed using the held-out `toy` class. Results are mean±std over 5 runs.

| METHOD | MEAN ACC. (%) | ALEATORIC (AUC) | EPISTEMIC (AUC) |
|---|---|---|---|
| *Binary neural networks* | | | |
| BIMU | **90.98 ± 0.38** | **1.00 ± 0.00** | **1.00 ± 0.00** |
| BAYESBINN | 90.20 ± 0.13 | **1.00 ± 0.00** | **1.00 ± 0.00** |
| STE | 77.69 ± 2.29 | – | – |
| SYNAPTIC METAPLASTICITY | 79.28 ± 0.85 | – | – |
| *Real-valued neural networks* | | | |
| MESU | **92.55 ± 0.24** | 0.99 ± 0.00 | **0.99 ± 0.00** |
| EWC ONLINE | 92.34 ± 0.32 | **1.00 ± 0.00** | – |
| SYNAPTIC INTELLIGENCE | 92.23 ± 0.30 | **1.00 ± 0.00** | – |
| SGD | 92.31 ± 0.31 | **1.00 ± 0.00** | – |

Standardization consistently improves performance across all methods, but the gains are markedly larger for real-valued neural networks. In particular, MESU improves from $87.01\%$ to $92.55\%$ mean accuracy, while EWC Online and Synaptic Intelligence increase from $87.18\%$ and $86.44\%$ to $92.34\%$ and $92.23\%$, respectively. These substantial gains indicate that real-valued continual learning methods strongly benefit from well-conditioned inputs and rely on stable feature statistics induced by mean–variance normalization. Their optimization dynamics and regularization mechanisms are therefore closely tied to offline preprocessing assumptions.

Binary Bayesian methods also benefit from standardization, but to a lesser extent. BiMU improves from $89.19\%$ to $90.98\%$ accuracy, and BayesBiNN from $86.93\%$ to $90.20\%$, while maintaining near-perfect epistemic uncertainty separation (ROC-AUC $\approx 1.00$). The more modest performance increase suggests that binary models are comparatively less sensitive to feature scaling.

When input statistics are known in advance and preprocessing based on the full training distribution is feasible, real-valued methods achieve higher final accuracy than binary approaches. However, the main results focus on realistic continual learning scenarios for edge and robotic systems, where data arrive as an unbounded stream and reliable estimation of global feature statistics is not possible online. In this regime, the robustness of binary Bayesian methods – and in particular BiMU – to unnormalized inputs and shifting distributions provides a practical advantage, despite their more constrained parameterization.

**Hyperparameter optimization.**

Hyperparameters are computed on the validation set of OpenLORIS-Object. They are computed through the custom hyperparameter tuning cost function $\mathcal{H}$ (see Appendix B.2).

*Table 16.* Hyperparameter configurations for all evaluated methods on the OpenLORIS dataset (standardized, 8192 features).

| METHOD | LEARNING RATE(S) | ADDITIONAL PARAMETERS | MC SAMPLES |
|---|---|---|---|
| *Binary Neural Networks* | | | |
| BIMU | 1.2 | $\alpha_{\mathrm{MAX}} = 0.07$, $\beta_{\mathrm{L}} = 184$, $\beta_{\mathrm{KL}} = 0.14$, $N = 7{,}900$ | 10 |
| BAYESBINN | 1.59 | PRIOR STRENGTH $= 1.2 \times 10^{-6}$ | 10 |
| SYNAPTIC METAPLASTICITY | $4.5 \times 10^{-5}$ | METAPLASTICITY $= 13.7$, WEIGHT DECAY $= 8 \times 10^{-12}$ | 1 |
| STE (BASELINE) | $1 \times 10^{-4}$ | – | 1 |
| *Real-Valued Neural Networks* | | | |
| MESU | $\alpha_\mu = 4.0$, $\alpha_\sigma = 5.0$ | $\sigma_{\mathrm{PRIOR}} = 0.76$, $N = 5{,}000{,}000$, CLAMP GRAD $= 0.53$ | 10 |
| ONLINE EWC | $4 \times 10^{-4}$ | IMPORTANCE $= 0.124$, DOWNWEIGHTING $= 0.7$ | 1 |
| SYNAPTIC INTELLIGENCE | $4 \times 10^{-4}$ | COEFF. $= 4.5 \times 10^{-5}$, DAMPING $= 0.05$ | 1 |
| SGD (BASELINE) | $3 \times 10^{-4}$ | – | 1 |

## K. OpenLORIS-Object Imbalanced Continual Active Learning: Complementary Figures

This section provides additional results on the OpenLORIS-Object benchmark, focusing on the imbalanced continual active learning setting. The figures presented here complement the main results and allow for a detailed inspection of each method's behavior over time and across acquisition steps.

Fig. 12 presents the results for BayesBiNN on the OpenLORIS dataset in imbalanced continual active learning setting. As in the Animals setting, all active learning strategies perform below random acquisition, highlighting the limited usefulness of the uncertainty estimates produced by the model. Although OpenLORIS contains a larger total number of training examples (442,194 images), this corresponds to only approximately seven Permuted MNIST–scale tasks. This short task horizon limits the degree of weight divergence that emerge during training. In the absence of a metaplastic learning rate mechanism, BayesBiNN is therefore unable to leverage these examples efficiently enough to consolidate task-relevant weights. As a consequence, the model struggles to differentiate between low-frequency and high-frequency samples. While this results in high overall uncertainty and yields strong out-of-distribution detection, the resulting uncertainty remains poorly structured, as predictions are dominated by the most recently observed samples rather than reflecting statistics accumulated over the full dataset, unlike approaches that explicitly incorporate metaplasticity such as BiMU.

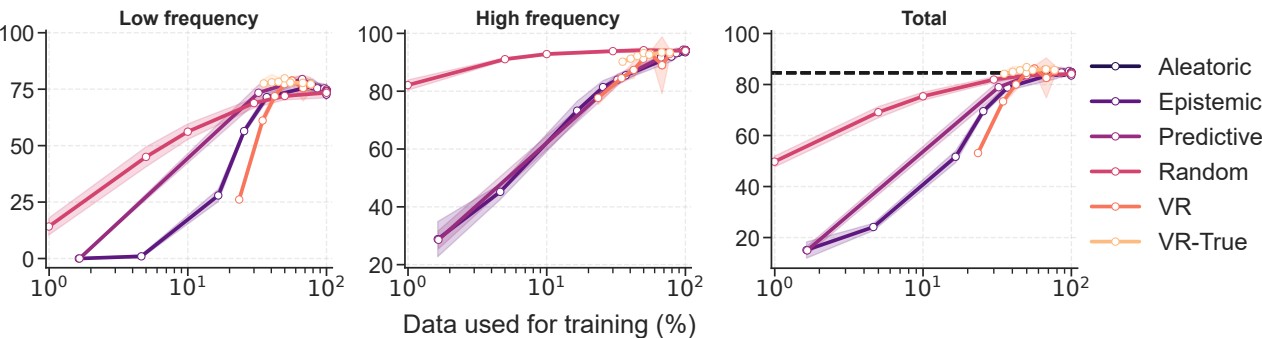

*Figure 12.* OpenLORIS-Object (OCL; linear head) with 8,192-dimensional VGG19 features under imbalanced continual active learning using BayesBiNN. Evolution of classification accuracy across labeling budgets in the presence of distribution shifts for Bayesian binary weights without forgetting.

Fig. 13 and Fig. 14 report MESU's performance with and without input standardization. In contrast to the Animals benchmark, performance degradation without standardization is less severe, owing to the substantially larger number of samples available to accumulate information about the importance of each parameter, leading to more reliable uncertainty estimates. Nevertheless, standardization consistently improves robustness and reduces variability across acquisition steps. In particular, standardization yields an accuracy gain of roughly 10 percentage points at the 3% labeling budget for VR. Under standardized features, aleatoric, epistemic, and predictive uncertainty criteria exhibit similar behavior, indicating reliable

discrimination of informative samples throughout the stream leading to strong efficiency of uncertainty estimates to select which data to integrate to increase accuracy.

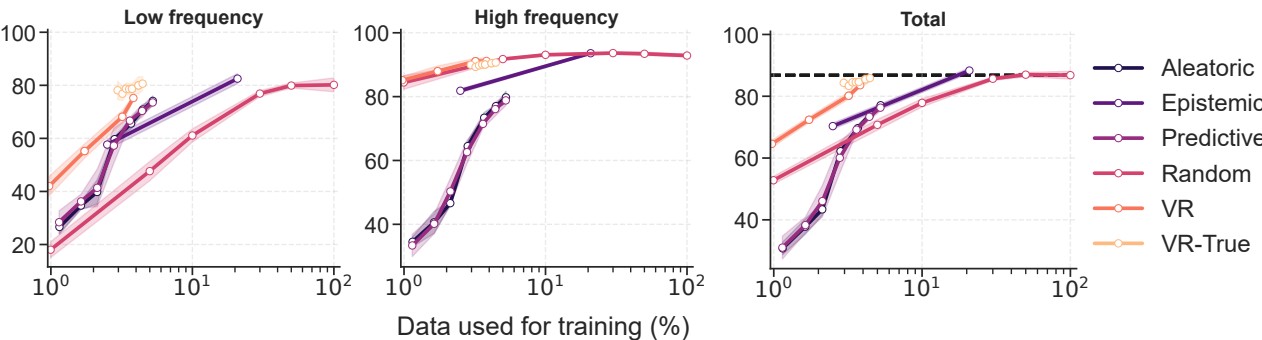

*Figure 13.* OpenLORIS-Object (OCL; linear head) with 8,192-dimensional VGG19 features under imbalanced continual active learning using MESU. Performance under uncertainty-driven querying with real-valued Bayesian weights.

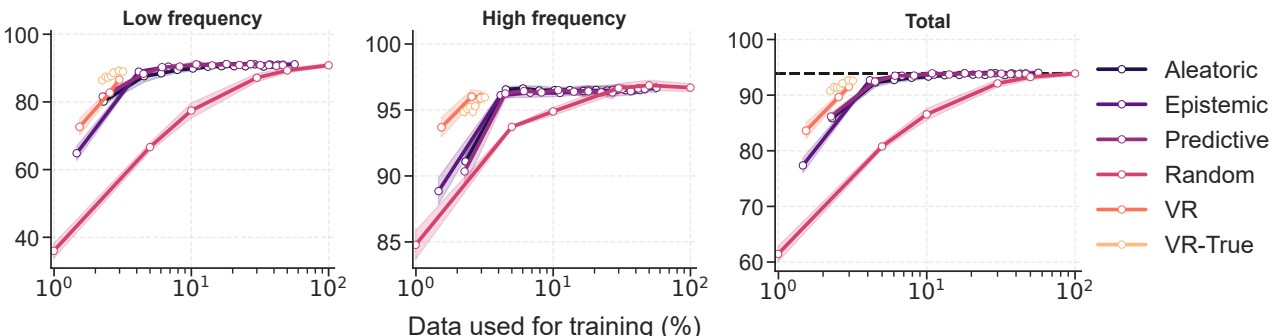

*Figure 14.* OpenLORIS-Object (OCL; linear head) with 8,192-dimensional VGG19 features under imbalanced continual active learning using MESU with feature standardisation. Standardisation improves robustness under continual distribution shifts.

Fig. 15 shows the results obtained with STE. STE's performance degrades more noticeably when confronted with successive task shifts, yielding poor aleatoric measurements that are unable to capture efficient examples to learn on.

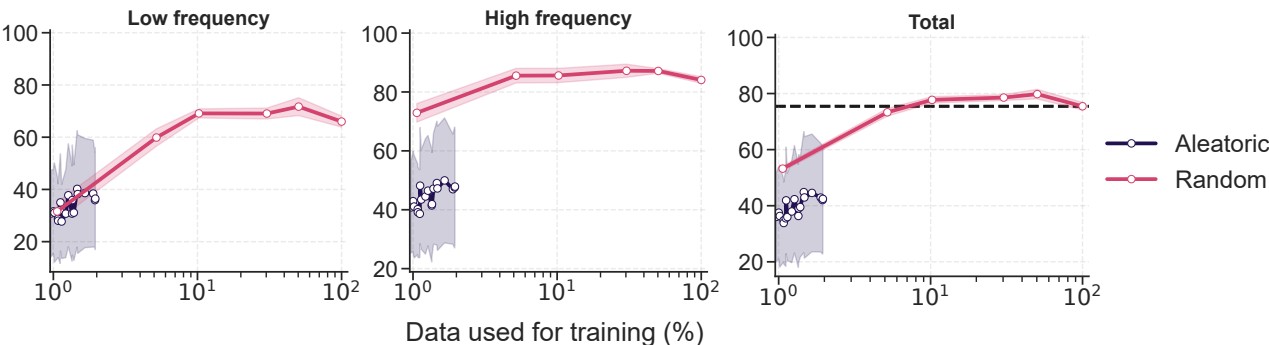

*Figure 15.* OpenLORIS-Object (OCL; linear head) with standardized 8,192-dimensional VGG19 features under imbalanced continual active learning using STE. Performance degradation under long-tailed distributions and sequential task shifts.

These complementary figures further support the main conclusions of the paper: that reliable epistemic uncertainty is critical for effective active continual learning under class imbalance, that its quality is strongly influenced by input conditioning in real-valued Bayesian methods, and that preserving uncertainty over long, non-stationary streams enables substantial reductions in labeling and update costs without sacrificing accuracy.

## L. OpenLORIS-Object Imbalanced Continual Active Learning: Number of predictors for variation ratio

In this appendix, we investigate the performance of variation ratio when limiting or increasing the number of predictors associated with the disagreement in the Online Continual Learning setting. Even when using few samples, variation ratio achieves strong predictive performance. For instance, using only 2 samples with 3.3% of the data achieves $89.3\% \pm 0.88$ accuracy, while 3 samples with 3.87% of the data reach $90.61\% \pm 0.53$. Low-sample regimes already provide high accuracy even without a large range of thresholds, denoting the ability of BiMU to distinguish well useful samples from non-informative ones.

Increasing the number of samples generally improves accuracy and allows finer control via the threshold parameter. For example, with 10 samples and 3.97% of the data, accuracy reaches $90.91\% \pm 0.98$, and with 25 samples using 4.81–5.63% of the data, accuracy increases up to $91.74\% \pm 0.23$. These higher-sample regimes provide additional thresholds for flexible performance trade-offs, but require more forward passes to estimate disagreement, leading to more computational overhead. High number of predictors also induce a higher variance at low data usage due to the difference of training examples selected through seeds.

*Table 17.* Ablation study of the number of samples used to compute the variation ratio uncertainty measure on OpenLORIS-Object (OCL; linear head) with 8,192-dimensional VGG19 features. Rows with identical samples, accuracy, and data usage are merged, and the thresholds column lists all thresholds for that configuration. Low sample regimes achieve high accuracy under strong compression. Additional samples yield better accuracy and offer more thresholds at the cost of more forward passes. Data used now shows mean $\pm$ standard deviation across final updates. Results are averaged over five runs.

| SAMPLES | ACCURACY (%) | DATA USED (%) | THRESHOLD |
|---|---|---|---|
| 2 | $89.30 \pm 0.88$ | $3.30 \pm 0.04$ | 0.50 |
| 3 | $90.61 \pm 0.53$ | $3.87 \pm 0.05$ | 0.33 |
| 3 | $82.26 \pm 1.15$ | $1.62 \pm 0.03$ | 0.66 |
| 5 | $88.85 \pm 0.92$ | $2.91 \pm 0.16$ | 0.20 |
| 5 | $80.86 \pm 1.90$ | $1.47 \pm 0.16$ | $0.40 - 0.60$ |
| 5 | $52.21 \pm 23.71$ | $0.34 \pm 0.19$ | 0.80 |
| 10 | $90.91 \pm 0.98$ | $3.97 \pm 0.03$ | 0.10 |
| 10 | $88.70 \pm 1.94$ | $3.10 \pm 0.31$ | $0.20 - 0.30$ |
| 10 | $79.27 \pm 4.00$ | $1.37 \pm 0.24$ | $0.40 - 0.50$ |
| 10 | $44.02 \pm 32.02$ | $0.42 \pm 0.39$ | 0.60 |
| 10 | $9.37 \pm 8.22$ | $0.02 \pm 0.04$ | 0.70 |
| 10 | $5.26 \pm 0.00$ | $0.00 \pm 0.00$ | $0.80 - 0.90$ |
| 25 | $91.34 \pm 0.50$ | $5.63 \pm 0.09$ | 0.04 |
| 25 | $91.11 \pm 0.44$ | $4.40 \pm 0.04$ | 0.12 |
| 25 | $90.93 \pm 0.70$ | $4.03 \pm 0.05$ | 0.16 |
| 25 | $90.23 \pm 0.68$ | $3.65 \pm 0.14$ | 0.20 |
| 25 | $89.00 \pm 0.64$ | $3.27 \pm 0.23$ | 0.24 |
| 25 | $87.16 \pm 1.88$ | $2.93 \pm 0.36$ | 0.28 |
| 25 | $86.40 \pm 2.68$ | $2.61 \pm 0.44$ | 0.32 |
| 25 | $79.14 \pm 13.18$ | $1.96 \pm 0.78$ | 0.36 |
| 25 | $66.41 \pm 30.90$ | $1.48 \pm 0.84$ | 0.40 |
| 25 | $51.53 \pm 37.93$ | $1.03 \pm 0.88$ | 0.44 |
| 25 | $34.82 \pm 36.25$ | $0.52 \pm 0.65$ | 0.48 |
| 25 | $19.95 \pm 29.37$ | $0.23 \pm 0.45$ | 0.52 |
| 25 | $19.16 \pm 27.79$ | $0.17 \pm 0.34$ | 0.56 |
| 25 | $18.48 \pm 26.44$ | $0.13 \pm 0.26$ | 0.60 |
| 25 | $17.01 \pm 23.49$ | $0.10 \pm 0.19$ | 0.64 |
| 25 | $16.07 \pm 21.61$ | $0.07 \pm 0.14$ | 0.68 |
| 25 | $5.26 \pm 0.00$ | $0.00 \pm 0.00$ | $> 0.72$ |
| BASELINE | $87.76 \pm 0.19$ | 100 | - |

# M. OpenLORIS-Object Imbalanced Continual Active Learning: Querying dynamics and temporal distribution of active updates

In this appendix, we further analyze when active updates occur during OpenLORIS-Object. We want to assert whether variation ratio querying reacts to distributional changes or if it selects samples uniformly throughout each task. For each task, we divide the online stream into 100 parts, with three temporal regions: the first quarter (0–24), the middle part of the task (25–74), and the last quarter (74–99). As labels are queried only when the variation ratio score exceeds the fixed threshold, the number of queried samples is also the number of backpropagation updates.

Table 18 reports the accuracy at several checkpoints within each task, together with the percentage of updates occurring in each temporal region. The main trend is consistent across tasks: most updates occur shortly after the distribution shift. On average, $45.85\%$ of all updates are performed in the first quarter of a task, while only $12.56\%$ occur in the last quarter. Querying is concentrated when the stream changes then progressively decreases as the model adapts to the current factor.

Importantly, the update rate does not collapse to zero at the end of the task. The last quarter still accounts for $12.56\%$ of the updates on average, indicating that the posterior remains sufficiently uncertain to select informative samples after the initial adaptation phase. The model reacts strongly after a shift and does not become completely rigid once the current task has been partially learned.

*Table 18.* Temporal distribution of variation ratio updates on OpenLORIS-Object (OCL; linear head) with 8,192-dimensional VGG19 features. Accuracy is reported at fixed checkpoints within each task. Update percentages indicate the fraction of updates occurring in the first quarter, middle part, and last quarter of the task. Most updates occur immediately after the task shift, while a smaller but non-zero fraction remains available later in the task.

| (A) ACCURACY CHECKPOINTS | | | | |
|---|---|---|---|---|
| TASK | ACC@0 (%) | ACC@25 (%) | ACC@50 (%) | ACC@75 (%) | ACC@99 (%) |
| TASK 1 | $5.26 \pm 0.00$ | $86.40 \pm 12.42$ | $90.53 \pm 6.55$ | $92.54 \pm 4.96$ | $90.26 \pm 4.30$ |
| TASK 2 | $5.26 \pm 0.00$ | $92.72 \pm 3.96$ | $92.50 \pm 4.16$ | $93.03 \pm 2.93$ | $91.01 \pm 2.79$ |
| TASK 3 | $8.73 \pm 6.93$ | $94.21 \pm 1.19$ | $89.82 \pm 0.58$ | $89.47 \pm 1.68$ | $88.11 \pm 1.78$ |
| TASK 4 | $8.90 \pm 7.28$ | $75.53 \pm 3.06$ | $94.61 \pm 2.59$ | $95.96 \pm 1.19$ | $91.93 \pm 1.32$ |
| TASK 5 | $12.68 \pm 9.56$ | $55.18 \pm 2.04$ | $95.44 \pm 0.93$ | $94.65 \pm 0.97$ | $90.00 \pm 1.28$ |
| TASK 6 | $11.45 \pm 10.22$ | $37.85 \pm 3.13$ | $93.51 \pm 1.74$ | $91.54 \pm 2.68$ | $81.14 \pm 1.21$ |
| TASK 7 | $18.60 \pm 17.69$ | $66.10 \pm 4.36$ | $80.88 \pm 3.11$ | $91.89 \pm 0.31$ | $90.31 \pm 0.78$ |
| TASK 8 | $18.25 \pm 16.71$ | $68.33 \pm 2.48$ | $80.44 \pm 2.69$ | $96.45 \pm 0.95$ | $93.60 \pm 0.97$ |
| TASK 9 | $12.32 \pm 8.70$ | $39.61 \pm 2.42$ | $53.25 \pm 2.05$ | $91.54 \pm 1.36$ | $80.96 \pm 1.06$ |
| TASK 10 | $19.78 \pm 17.86$ | $68.82 \pm 3.38$ | $70.79 \pm 2.00$ | $83.25 \pm 1.57$ | $89.78 \pm 1.40$ |
| TASK 11 | $21.32 \pm 19.67$ | $64.61 \pm 5.02$ | $72.06 \pm 2.27$ | $76.18 \pm 1.36$ | $92.37 \pm 1.08$ |
| TASK 12 | $14.87 \pm 11.93$ | $42.11 \pm 2.14$ | $41.58 \pm 2.92$ | $42.37 \pm 1.25$ | $94.25 \pm 2.00$ |
| AVERAGE | $13.12 \pm 5.29$ | $65.96 \pm 18.73$ | $79.62 \pm 16.68$ | $86.57 \pm 14.40$ | $89.48 \pm 4.10$ |

| (B) TEMPORAL DISTRIBUTION OF UPDATES | | |
|---|---|---|
| TASK | UPDATES 0–24 (%) | UPDATES 25–73 (%) | UPDATES 74–99 (%) |
| TASK 1 | $25.43 \pm 31.15$ | $40.19 \pm 29.00$ | $13.08 \pm 12.57$ |
| TASK 2 | $41.00 \pm 20.55$ | $41.82 \pm 11.01$ | $15.66 \pm 9.49$ |
| TASK 3 | $53.47 \pm 1.72$ | $33.85 \pm 1.05$ | $10.95 \pm 1.35$ |
| TASK 4 | $58.65 \pm 1.75$ | $29.72 \pm 1.47$ | $9.81 \pm 1.37$ |
| TASK 5 | $61.35 \pm 1.65$ | $29.07 \pm 1.63$ | $7.67 \pm 1.26$ |
| TASK 6 | $56.65 \pm 3.22$ | $32.93 \pm 2.51$ | $8.26 \pm 1.02$ |
| TASK 7 | $42.37 \pm 2.68$ | $40.33 \pm 2.30$ | $14.96 \pm 0.79$ |
| TASK 8 | $41.28 \pm 4.12$ | $42.43 \pm 3.22$ | $14.60 \pm 1.55$ |
| TASK 9 | $42.83 \pm 2.84$ | $40.16 \pm 1.31$ | $14.76 \pm 2.30$ |
| TASK 10 | $40.73 \pm 1.84$ | $43.60 \pm 1.14$ | $13.65 \pm 2.15$ |
| TASK 11 | $39.53 \pm 2.58$ | $42.64 \pm 1.02$ | $15.13 \pm 2.02$ |
| TASK 12 | $46.93 \pm 2.25$ | $39.01 \pm 1.72$ | $12.19 \pm 1.45$ |
| AVERAGE | $45.85 \pm 9.70$ | $37.98 \pm 4.95$ | $12.56 \pm 2.66$ |

## N. Budget-driven adaptive thresholding for variation ratio

In the main experiments, active continual learning uses a fixed one-pass threshold: an incoming example is queried if its uncertainty score exceeds $\tau$. While a fixed threshold is simple and hardware-friendly, deployment may require tighter control over the realized query rate as we do not know in advance the budget of updates that variation ratio will use.

In this appendix, we propose a fully online budget-driven controller that adapts $\tau$ from the discrepancy between the current query rate and a target budget, set in advance as a hyperparameter.

Let $B \in [0, 1]$ denote the target fraction of samples to query, and let $Q_{\text{rate}}(t)$ be the fraction of samples queried up to time $t$. The signed budget error is

$$\epsilon_t = Q_{\text{rate}}(t) - B. \tag{67}$$

A positive value indicates that the current stream has exceeded the target budget, whereas a negative value indicates that the controller is below budget. We compute a continuous threshold

$$\tau_{\text{cont}}(t) = B + \text{sign}(\epsilon_t) \, |\epsilon_t|^{\gamma}, \tag{68}$$

where $\gamma > 0$ controls the adaptation speed. When $Q_{\text{rate}}(t) > B$, the threshold is increased, making the controller more selective. Conversely, when $Q_{\text{rate}}(t) < B$, the threshold is decreased, making the controller more permissive.

For variation ratio querying, the uncertainty score is discrete due to being computed from a finite number of Monte Carlo predictors. With $K$ posterior samples, variation ratio can only take values in a finite set. We quantize the controller output onto the same set:

$$n_t = \text{round}(K \tau_{\text{cont}}(t)), \tag{69}$$
$$n_t = \min(\max(n_t, 0), K), \tag{70}$$
$$\tau_t = \frac{n_t}{K}. \tag{71}$$

The controller is memory-free; it only requires the current cumulative query rate, the target budget, and the finite-$K$ quantization level.

We evaluate this controller on OpenLORIS-Object under the same active continual learning setting as in Sec. 4.5: sequential factor analysis, frozen VGG19 features, an online linear head, 8,192 randomly selected features, class imbalance, and one-pass training without replay. We use variation ratio querying with $K = 10$ posterior samples. The queried fraction is reported as the percentage of samples that trigger both a label request and a parameter update.

Adaptive thresholding can recover the performance of the best fixed-threshold variation ratio while explicitly targeting an update budget. With a $3\%$ target budget and $\gamma = 0.50$, the controller reaches $90.79 \pm 0.61\%$ accuracy while querying $3.96 \pm 0.03\%$ of the stream. This is close to the best fixed-threshold variation ratio result, which reaches $90.91\%$ accuracy at $3.97\%$ queried data. Budget-driven adaptive thresholding is feasible in the fully online setting.

However, the controller is sensitive to the operating regime. At very low budgets, performance degrades sharply for small $\gamma$. For example, at $B = 0.5\%$ and $\gamma = 0.01$, the controller queries exactly $0.50\%$ of the stream but reaches only $52.68 \pm 2.29\%$ accuracy. Here, the budget is too limited and variation ratio stays at its minimal threshold and avoids taking updates. Hence, matching the target budget does not allow to preserve enough queried samples to learn the task correctly.

To sum up, we proposed an algorithm enabling online adaptation of the querying threshold with a constraint parameter setting how tight the budget should be kept and demonstrated results on par to static thresholding.

*Table 19.* Budget-driven adaptive variation ratio thresholding on OpenLORIS-Object (OCL; linear head) with 8,192-dimensional VGG19 features. The target budget $B$ is the desired queried fraction. The exponent $\gamma$ controls how aggressively the threshold reacts to the signed budget error. Queried samples are also the only samples used for backpropagation updates. Results are averaged over five runs.

| METHOD | CONTROLLER SETTING | DATA USED (%) | ACCURACY (%) |
|---|---|---|---|
| RANDOM QUERYING | FULL STREAM | 100 | $87.76 \pm 0.19$ |
| BUDGETED VR | $B = 0.5\%, \gamma = 0.01$ | $0.50 \pm 0.00$ | $52.68 \pm 2.29$ |
| BUDGETED VR | $B = 0.5\%, \gamma = 0.10$ | $1.11 \pm 0.03$ | $79.84 \pm 0.65$ |
| BUDGETED VR | $B = 0.5\%, \gamma = 0.50$ | $3.34 \pm 0.05$ | $89.62 \pm 0.43$ |
| BUDGETED VR | $B = 1.0\%, \gamma = 0.01$ | $1.00 \pm 0.00$ | $61.95 \pm 1.95$ |
| BUDGETED VR | $B = 1.0\%, \gamma = 0.10$ | $1.22 \pm 0.01$ | $79.53 \pm 1.40$ |
| BUDGETED VR | $B = 1.0\%, \gamma = 0.50$ | $3.34 \pm 0.05$ | $89.88 \pm 0.57$ |
| BUDGETED VR | $B = 2.0\%, \gamma = 0.01$ | $2.00 \pm 0.01$ | $70.20 \pm 1.72$ |
| BUDGETED VR | $B = 2.0\%, \gamma = 0.10$ | $2.03 \pm 0.01$ | $85.64 \pm 1.21$ |
| BUDGETED VR | $B = 2.0\%, \gamma = 0.50$ | $3.70 \pm 0.02$ | $89.97 \pm 0.74$ |
| BUDGETED VR | $B = 3.0\%, \gamma = 0.01$ | $3.00 \pm 0.01$ | $73.90 \pm 0.60$ |
| BUDGETED VR | $B = 3.0\%, \gamma = 0.10$ | $3.00 \pm 0.01$ | $86.79 \pm 0.71$ |
| BUDGETED VR | $B = 3.0\%, \gamma = 0.50$ | $3.96 \pm 0.03$ | $\mathbf{90.79 \pm 0.61}$ |
| BUDGETED VR | $B = 4.0\%, \gamma = 0.01$ | $4.00 \pm 0.01$ | $76.97 \pm 0.76$ |
| BUDGETED VR | $B = 4.0\%, \gamma = 0.10$ | $4.00 \pm 0.01$ | $87.94 \pm 0.30$ |
| BUDGETED VR | $B = 4.0\%, \gamma = 0.50$ | $4.23 \pm 0.03$ | $90.56 \pm 0.30$ |
| BUDGETED VR | $B = 5.0\%, \gamma = 0.01$ | $5.00 \pm 0.01$ | $78.34 \pm 0.60$ |
| BUDGETED VR | $B = 5.0\%, \gamma = 0.10$ | $5.00 \pm 0.01$ | $87.66 \pm 0.85$ |
| BUDGETED VR | $B = 5.0\%, \gamma = 0.50$ | $5.01 \pm 0.01$ | $90.87 \pm 0.51$ |

# O. Algorithm: Binary Metaplasticity from Uncertainty

---

**Algorithm 1** Binary Metaplasticity from Uncertainty

---

**Input:** $M$ Batches $\{\mathcal{D}_t\}_{t=1}^{M}$, Monte Carlo samples $K$, number of synapses $s$, temperature $T$, prior parameter $\boldsymbol{\lambda}_{\text{prior}}$, memory window $N$, maximum learning rate $\alpha_{\max}$

**for** $t = 1$ **to** $M$ **do**

    **for** $k = 1$ **to** $K$ **do**

        Sample $\epsilon_k^{(1)}, \ldots, \epsilon_k^{(s)} \overset{i.i.d.}{\sim} \mathcal{U}(0, 1)$

        $\boldsymbol{\delta}_k = \frac{1}{2}\left(\log(\boldsymbol{\epsilon}_k) - \log(1 - \boldsymbol{\epsilon}_k)\right)$

        Compute the Gumbel-softmax trick $\boldsymbol{\omega}_k = \tanh\left(\frac{1}{T}(\boldsymbol{\lambda}_{t-1} + \boldsymbol{\delta}_k)\right)$

        Compute loss $\mathcal{L}(\boldsymbol{\omega}_k, \mathcal{D}_t)$

        Compute gradient of the loss function $\frac{\partial \mathcal{L}(\boldsymbol{\omega}_k, \mathcal{D}_t)}{\partial \boldsymbol{\omega}_k}$

        Compute gradient over $\boldsymbol{\lambda}_{t-1}$

        $\frac{\partial \mathcal{L}(\boldsymbol{\omega}_k, \mathcal{D}_t)}{\partial \boldsymbol{\lambda}_{t-1}} \leftarrow \frac{\partial \mathcal{L}(\boldsymbol{\omega}_k, \mathcal{D}_t)}{\partial \boldsymbol{\omega}_k} \odot \frac{1 - \boldsymbol{\omega}_k^2}{T}$

    **end for**

    Estimate expected gradients:

    $\left.\frac{\partial \mathcal{L}(\boldsymbol{\lambda}, \mathcal{D}_t)}{\partial \boldsymbol{\lambda}}\right|_{\boldsymbol{\lambda} = \boldsymbol{\lambda}_{t-1}} \leftarrow \frac{1}{K} \sum_{k=1}^{K} \left[\frac{\partial \mathcal{L}(\boldsymbol{\omega}_k, \mathcal{D}_t)}{\partial \boldsymbol{\lambda}_{t-1}}\right]$

    Compute adaptative learning rate:

    $\eta(\boldsymbol{\lambda}_{t-1}) = \dfrac{1}{\frac{1}{\cosh^2(\boldsymbol{\lambda}_{t-1})} + 2\tanh(\boldsymbol{\lambda}_{t-1}) \left.\frac{\partial \mathcal{L}(\boldsymbol{\lambda}, \mathcal{D}_t)}{\partial \boldsymbol{\lambda}}\right|_{\boldsymbol{\lambda} = \boldsymbol{\lambda}_{t-1}} + \frac{1}{\alpha_{\max}} + 2\left|\left.\frac{\partial \mathcal{L}(\boldsymbol{\lambda}, \mathcal{D}_t)}{\partial \boldsymbol{\lambda}}\right|_{\boldsymbol{\lambda} = \boldsymbol{\lambda}_{t-1}}\right|}$

    Update parameters:

    $\boldsymbol{\lambda}_t = \boldsymbol{\lambda}_{t-1} - \eta(\boldsymbol{\lambda}_{t-1}) \odot \left(\left.\frac{\partial \mathcal{L}(\boldsymbol{\lambda}, \mathcal{D}_t)}{\partial \boldsymbol{\lambda}}\right|_{\boldsymbol{\lambda} = \boldsymbol{\lambda}_{t-1}} + \frac{(\boldsymbol{\lambda}_{t-1} - \boldsymbol{\lambda}_{\text{prior}})}{N \cdot \cosh^2(\boldsymbol{\lambda}_{t-1})}\right)$

**end for**

---

## P. Computational Cost Evaluation

The proposed active learning strategy presented in this paper relies on the fact that in binary neural networks, inference is substantially cheaper than gradient-based updates. BiMU exploits this asymmetrical cost by using repeated low-cost Bayesian inference passes to estimate uncertainty, while triggering expensive backpropagation updates only for informative samples. In this appendix, we ask whether the overhead induced by Monte Carlo Bayesian sampling remains smaller than the compute saved by avoiding most training updates.

We evaluate this trade-off directly on an embedded device. We measure the inference and training costs of BiMU, BayesBiNN, STE, and Synaptic Metaplasticity on a microcontroller unit (MCU) implementation, and analyze the effective compute cost of active continual learning under realistic querying rates.

More specifically, we address two claims: **(i)** binary inference is substantially cheaper than online training updates, even in Bayesian settings requiring multiple stochastic forward passes; **(ii)** BiMU reduces overall compute despite the overhead of Monte Carlo samples to estimate uncertainty.

We implemented all methods on an STMicroelectronics STM32 NUCLEO-64 MCU board running at 216 MHz. Forward passes were implemented using 32-bit signed integers, while backward passes used 32-bit floating-point operations due to gradient estimation requiring non-integer values. The measurements in Table 20 were obtained on a reference layer with $N_{in} = 512$ and $N_{out} = 16$.

To estimate the runtime of the OpenLORIS experiments, we project these measurements to the architecture used in the main benchmark, with $N_{in} = 8192$ and $N_{out} = 19$. This corresponds to a $19\times$ increase in the number of layer operations relative to the measured reference layer. As all compared methods use the same classifier structure and the same MCU implementation constraints, this projection preserves the relative compute ratios between inference and training while reflecting the substantially larger dimensionality of the OpenLORIS feature space.

We note that binary inference benefits from sub-byte operations and popcount instructions, whereas training remains dominated by floating-point computations and memory writes. Consequently, the gap measured here between inference and training cost in this appendix is likely conservative with respect to dedicated binary hardware accelerators.

*Table 20.* NUCLEO-64 benchmark results (216 MHz). Latency and CPU cycles correspond to the mean $\pm$ standard deviation over 400 iterations for a layer with $N_{in} = 512$ and $N_{out} = 16$.

| | INFERENCE | | TRAINING | |
|---|---|---|---|---|
| ALGORITHM | TIME ($\mu s$) | CYCLES | TIME ($\mu s$) | CYCLES |
| BiMU | $717.93 \pm 0.37$ | 155,170 | $13991.27 \pm 3.72$ | 3,022,211 |
| BayesBiNN | $718.01 \pm 0.37$ | 155,195 | $13117.61 \pm 5.27$ | 2,833,522 |
| STE | $310.85 \pm 0.83$ | 67,233 | $3643.02 \pm 2.69$ | 786,999 |
| Syn. Meta. | $310.91 \pm 0.81$ | 67,243 | $7723.78 \pm 3.82$ | 1,668,432 |

Table 20 supports claim **(i)** by showing that, for BiMU specifically, the forward pass requires only $C_{\text{inference}} = 0.72$ ms, whereas the backward update requires $C_{\text{train}} = 14.0$ ms, making inference approximately $19.4\times$ cheaper than training. Similar trends are observed across all binary methods, confirming that the dominant computational cost arises from parameter updates rather than inference itself.

*Table 21.* Projected compute cost on the OpenLORIS-8192 architecture ($N_{in} = 8192$, $N_{out} = 19$).

| ALGORITHM | INFERENCE ($ms$) | TRAINING ($ms$) | EXPECTED COMPUTE PER DATA POINT ($ms$) | ACCURACY (%) |
|---|---|---|---|---|
| BiMU (ACTIVE LEARNING) | 13.7 | 266.0 | 45.0 | 89.30 |
| BiMU | 13.7 | 266.0 | 559.4 | 90.98 |
| BayesBiNN | 13.6 | 249.2 | 525.6 | 84.18 |
| STE | 5.9 | 62.2 | 62.2 | 77.69 |
| Syn. Meta. | 5.9 | 146.7 | 146.7 | 79.28 |

We next evaluate whether the reduction in update frequency enabled by active learning compensates for the additional Monte Carlo forward passes required for uncertainty estimation. We consider the OpenLORIS-8192 setting used in the main text. Table 17 shows that BiMU reaches $89.30\%$ accuracy using only $K = 2$ Monte Carlo samples while querying and updating on only $B_{\text{rate}} = 3.3\%$ of the stream. We also consider $K = 2$ Monte Carlo samples for the backward pass of BiMU and BayesBiNN.

Under this setting, the expected compute cost per incoming data point for BiMU with active learning is

$$C_{\text{AL}} = K \times (C_{\text{inference}} + B_{\text{rate}} C_{\text{train}}), \tag{72}$$

where $K$ is the number of Monte Carlo predictors and $B_{\text{rate}}$ the fraction of queried samples triggering backpropagation. By contrast, BiMU and BayesBiNN full online training performs one inference and one update for every Monte Carlo predictor:

$$C_{\text{B}} = K \times (C_{\text{inference}} + C_{\text{train}}). \tag{73}$$

Finally, STE and Synaptic Metaplasticity only require their backward cost.

Table 21 supports claim **(ii)**. Despite performing multiple stochastic forward passes per data point, BiMU with active learning reduces the expected compute cost to $45.0$ ms, compared to $559.4$ ms for full BiMU training and $525.6$ ms for BayesBiNN. This corresponds to approximately a $12.4\times$ reduction relative to full BiMU training while maintaining comparable accuracy ($89.30\%$ versus $90.98\%$). STE remains computationally very cheap because it does not provide uncertainty estimates: this comes at a substantial accuracy degradation in accuracy ($77.69\%$).

Overall, the results show that the low cost of binary inference is sufficient to amortize the overhead of Bayesian uncertainty estimation when active querying significantly reduces the number of gradient updates, supporting claims **(i)** and **(ii)**.

