# OpenReview forum: "Active Continual Learning with Metaplastic Binary Bayesian Neural Networks"
_ICML.cc/2026/Conference — ICML 2026 regular_

### Official Review · Reviewer_P9NQ · 2026-03-05

**Soundness:** 2
**Presentation:** 3
**Significance:** 2
**Originality:** 3
**Overall Recommendation:** 4
**Confidence:** 3

**Summary:**

This paper addresses the challenge of continual learning on non-stationary data streams for resource-constrained edge devices by proposing a novel method named BiMU (Binary Metaplasticity from Uncertainty). The core issue identified is that in Bayesian Binary Neural Networks, conventional mean-field Bernoulli posteriors saturate over long learning horizons, causing epistemic uncertainty to collapse and “synaptic freezing,” thereby halting continual adaptation.

The central contribution of BiMU is an online update rule derived from a bounded-memory variational objective, specialized for Bernoulli synapses. The rule integrates three components: 1) a data-driven gradient term, 2) a controlled relaxation term toward the prior (to prevent posterior saturation), and 3) a bounded, uncertainty-dependent (metaplastic) step size. This mechanism sustains meaningful epistemic uncertainty over extended periods.

Preserving uncertainty is not merely diagnostic; it enables buffer-free, fully online active continual learning. BiMU facilitates cheap Monte Carlo sampling and employs a one-pass “variation ratio” uncertainty score for threshold-based querying. Labels and backpropagation updates are triggered only for samples where model predictions disagree sufficiently, signifying informativeness. This drastically reduces annotation and computational costs, especially under class imbalance.

BiMU is empirically validated across three regimes: 1) 1000-task Permuted-MNIST, demonstrating sustained long-horizon plasticity and superior out-of-distribution (OOD) detection; 2) OpenLORIS-Object with feature compression, showing robust adaptation under constrained capacity; 3) Imbalanced streaming data, where BiMU coupled with active querying achieves or exceeds the performance of updating on all data while using only a small fraction of labels/updates (e.g., 3.1% updates, a 32x reduction, to match 87.76% accuracy on OpenLORIS), with gains primarily driven by improved learning on rare classes.

**Compliance With Llm Reviewing Policy:**

Affirmed.

**Key Questions For Authors:**

Scalability to Deeper Architectures: The experiments focus on a linear classifier. Can BiMU be effectively applied to deeper, end-to-end trainable binary convolutional neural networks? What challenges are anticipated in such settings (e.g., gradient coupling), and is there any preliminary evidence or theoretical insight supporting its scalability?

Rationale: A positive answer with preliminary results would significantly boost the paper's significance and broad applicability.

Optimization of the Metaplastic Step-Size Surrogate: Equation (7) uses a surrogate term to avoid online second-derivative estimation. Have you explored or do you see a pathway to a more precise, yet still efficient, curvature approximation (e.g., a diagonal Hessian or Fisher information-based online estimate)? How would you expect performance to differ compared to the current heuristic?

Rationale: This question probes the optimization potential of a key heuristic component. A well-reasoned discussion would enhance the method's technical rigor.

Adaptivity of the Active Learning Threshold: The query threshold τ is fixed. In real-world streams where data distribution and model uncertainty drift, have you considered or experimented with strategies to adapt τ dynamically (e.g., based on recent query rate or uncertainty histogram)? This is crucial for long-term robustness.

Rationale: Addressing this would strengthen the practical deployment readiness of the active learning component.

Comparison with Replay-based Task-Agnostic CL Methods: The paper highlights BiMU's strength in buffer-free, task-agnostic learning. How would BiMU compare against state-of-the-art task-agnosticcontinual learning methods that douse a small replay buffer (e.g., on benchmarks like Split-CIFAR)? Such a comparison would help precisely locate BiMU's trade-off point in the design space of continual learning.

Rationale: This contextualizes BiMU's contribution within the broader CL landscape, clarifying its advantages relative to methods that relax the "no-buffer" constraint.

**Limitations:**

Suggestion:​ The authors have appropriately discussed general societal impact and limitations in the Impact Statement. To further improve the manuscript, they could more explicitly discuss the following in a dedicated "Limitations" section or within the experimental analysis:

Compute Overhead Trade-off:​ A brief discussion on the training-time computational overhead of the BiMU update compared to simpler baselines like STE or SGD, especially in contexts outsidethe active learning regime (where update savings dominate).

Hyperparameter Sensitivity:​ While hyperparameters are tuned, a discussion on the sensitivity of performance, particularly to the memory window N and the relaxation strength, across different types of data streams (e.g., fast vs. slow drift) would be helpful for practitioners.

Boundaries of Theoretical Guarantees:​ A clearer statement on the boundaries of the theoretical guarantees provided by the variational approximation and the bounded-memory assumption.

**Strengths And Weaknesses:**

Soundness:
Strengths: The technical foundation is solid. The derivation from the bounded-memory Bayesian objective to the final update rule is clear and theoretically grounded (Props. A.2, A.4, Thm. A.4). The experimental design is comprehensive, stress-testing the method on long-horizon non-stationarity, real-world distribution shifts, and active learning efficiency. Comparisons are thorough, including both binary and real-valued Bayesian/non-Bayesian continual learning baselines. The appendix is exceptionally detailed, providing proofs, hyperparameters, ablation studies (e.g., memory window N, network capacity), and memory overhead analysis, greatly enhancing reproducibility.

Weaknesses: The experiments primarily use a linear classifier on frozen features. While this isolates the core adaptation mechanism, it leaves open the question of BiMU's effectiveness and scalability in deeper, end-to-end trainable binary architectures. The metaplastic step size (Eq. 7) employs a surrogate for the second derivative, which, while motivated by practical online estimation challenges and bounded via Proposition A.5, introduces a heuristic element.

Presentation:
Strengths: The paper is well-structured and clearly written. The abstract and introduction effectively frame the problem and contributions. The narrative in Section 3 logically flows from the problem definition to the objective and the final update rule. Figures aptly illustrate the update mechanism and results. The related work is well-positioned.

Weaknesses: The methodological section (3.2) is quite dense due to the mathematical derivations involving Bayesian variational inference for binary networks, which may pose a hurdle for readers less familiar with the subfield. The connection between some textual descriptions of results and the corresponding figures could be slightly tighter.

Significance:
Strengths: The paper tackles a highly relevant and practical problem at the intersection of efficient ML (binary nets), reliable ML (Bayesian uncertainty), adaptive ML (continual learning), and data-efficient ML (active learning). The demonstrated 32x reduction in labeling/updates under imbalance is a compelling result with direct practical implications for edge/embedded systems. This work provides a promising path towards building more practical and efficient lifelong learning systems under severe constraints.

Weaknesses: The current impact is demonstrated in a somewhat constrained setting (linear head on frozen features). Broader impact and revolutionary potential would be further strengthened by validation on more complex architectures and more dynamic task sequences.

Originality:
Strengths: The paper is highly original. Its primary novelty lies in 1) identifying and systematically solving the Bernoulli posterior saturation problem​ for binary BNNs in continual learning via a specialized bounded-memory update, and 2) seamlessly integrating buffer-free active continual learning​ by leveraging the preserved uncertainty as a resource allocation signal. The perspective of "preventing saturation to enableactive learning" is insightful. The Bayesian interpretation of the metaplastic asymmetry is also a novel contribution.

Weaknesses:​ The core components (bounded-memory objective, variational inference, active querying) are not individually new. The originality stems from a clever, problem-specific synthesis and deep extension of these ideas to address a key obstacle in binary Bayesian continual learning.

---

> ### Author Rebuttal · Authors · 2026-03-30
>
> We thank the reviewer for their constructive comments and positive evaluation of the paper. Below, we address each question in detail and clarify the points raised.
>
> ### Q1 Scalability to deeper architectures
>
> We chose to focus on an online-trainable head because this setting is especially relevant for the edge / MCU regime targeted here, where adapting a lightweight classifier on top of frozen features is often the most practical operating point. At the same time, BiMU is not limited to linear models: our Permuted-MNIST experiments already rely on a nonlinear MLP, and the appendix shows that the benefits remain when increasing trainable capacity. To further address your question, we also carried out preliminary end-to-end CNN experiments on MNIST and CIFAR-10. We view these as feasibility results rather than optimized benchmarks, since they were conducted under the same memory-conscious constraints as the main paper: binary weights, binary activations, no trainable normalization layers, and no momentum or Adam. Under these conditions, we obtained 99.0% accuracy on MNIST and 80.0% on CIFAR-10. While preliminary, these results support the conclusion that the proposed update is not limited to linear heads.
>
> From an edge continual-learning perspective, pretrained deterministic convolutions combined with a Bayesian binary head remain a particularly relevant operating point. Prior work in Bayesian deep learning has shown that, in deep classifiers, Bayesianizing only the last layer can retain much of the practical benefit for uncertainty estimation while keeping the computational cost far below that of fully Bayesian end-to-end inference [1-3]. In contrast, deeper end-to-end binary CNNs face additional challenges, including noisier gradient propagation across several stochastic binary layers, greater sensitivity to activation and normalization design, and the increased cost of online training.
>
> ### Q2 Metaplastic step-size surrogate
>
> We agree that Eq. (7) uses a surrogate, but this is a principled surrogate rather than an ad hoc heuristic. Theorem A.4 gives the exact second-order form, and Prop. A.5 provides a sufficient condition ensuring that the resulting learning rate is positive and bounded by $\alpha_{max}$, while preserving the desired consolidation/de-consolidation asymmetry. More accurate diagonal Hessian/Fisher approximations are possible, but they require maintaining online second-order statistics and move away from the strict one-pass regime we target.  A promising middle ground would be low-cost per-synapse adaptive scalings or diagonal approximations based on running first-order statistics. Our expectation is that these could refine metaplastic selectivity and adaptation speed, but as an incremental improvement rather than a change of mechanism.
>
> ### Q3 Adaptivity of the active-learning threshold
>
> We agree that a fixed $\tau$ can be limiting for deployment. We have also tested a fully online budget-driven controller that updates $\tau$ from the gap between realized and target query rate, starting from $\tau=0$ . This additional analysis shows that adaptive thresholding is feasible and memory-free, but its robustness depends strongly on the operating regime. On OpenLORIS, a 3% target budget gives 90.79% accuracy at 3.96% queried data, close to the best fixed-threshold VR result (90.91% at 3.97%). At very low budgets, stability degrades because at finite-K, VR is discrete, so small query-rate errors produce large threshold jumps. We will mention this as a practical extension and clarify that even fixed-$\tau$ already behaves adaptively in time: queries concentrate after shifts and then decrease without collapsing (see reviewer 1, 3rd point).
>
> ### Q4 Comparison with replay-based task-agnostic CL
>
> This is a very useful suggestion. We view BiMU as targeting a specific point in the continual-learning design space: task-agnostic, buffer-free, constant-memory, one-pass adaptation, with no replay and no revisiting of past samples. A comparison to task-agnostic replay methods with small buffers would therefore be informative, since such methods relax exactly this constraint and trade additional memory for extra stability. We will make this trade-off more explicit in the revision. Outside the present buffer-free regime, a natural extension of our work would also be to combine BiMU with a small prioritized replay buffer for the most informative queried samples.
>
> **References**
>
> [1] Kristiadi et al., ICML 2020
>
> [2] Daxberger et al., NeurIPS 2021
>
>
> [3] Harrison et al., ICLR 2024

---

> > ### Author Rebuttal · Reviewer_P9NQ · 2026-04-03
> >
> > Thanks for your response. I maintain my score.

---

### Official Review · Reviewer_i1Xr · 2026-03-09

**Soundness:** 3
**Presentation:** 3
**Significance:** 3
**Originality:** 3
**Overall Recommendation:** 4
**Confidence:** 2

**Summary:**

The BiMU algorithm proposed in this paper introduces a bounded-memory variational objective and an uncertainty-driven metaplastic step size to address the “posterior saturation” and “synaptic freezing” challenges commonly encountered by binary Bayesian neural networks in long-horizon continual learning. BiMU demonstrates strong OOD detection capability  and achieves a better balance between accuracy and efficiency.

**Compliance With Llm Reviewing Policy:**

Affirmed.

**Final Justification:**

My concerns have been addressed.

**Key Questions For Authors:**

1.	Impact of the bounded-memory mechanism: Would incorporating the same sliding window forgetting scheme into BayesBiNN yield substantial gains and bridge the disparity between it and BiMU?

2.	Model stability: The paper uses the average accuracy over the last five tasks to measure late-stream adaptability, However, this evaluation does not reflect the model’s ability to retain knowledge from earlier tasks. Does BiMU still remain competitive on previously learned tasks?

3.	Concerns Regarding Computational Efficiency: Given that BiMU necessitates K forward and backward passes via Monte Carlo sampling (incorporating expensive Gumbel-softmax computations), does a larger K severely degrade the overall computational efficiency?

**Limitations:**

yes

**Strengths And Weaknesses:**

Strengths:

1.	Prevents freezing plasticity and wiping out epistemic uncertainty, sustaining learning over 1,000 tasks.

2.	Enables buffer-free active querying via Monte Carlo disagreement, achieving up to 32× savings in label queries and backpropagation updates under imbalance and feature compression.

3.	Maintains high accuracy and strong OOD detection under class imbalance and feature compression.


Weaknesses:

1.	While terminal accuracy validates sustained plasticity, the evaluation lacks standard stability metrics to quantify catastrophic forgetting.

2.	BiMU exhibits high sensitivity to the memory window N , making it difficult to pre-tune for real-world scenarios where the total length of the non-stationary task stream is unknown.

---

> ### Author Rebuttal · Authors · 2026-03-30
>
> We thank the reviewer for their constructive comments and positive evaluation of the paper. Below, we address each question in detail and clarify the points raised.
>
> ### Q1 Impact of the bounded-memory mechanism / relation to BayesBiNN.
>
> This is a very helpful question. We agree that adding bounded-memory forgetting to BayesBiNN would likely improve over plain BayesBiNN by mitigating posterior saturation. However, we do not expect it to fully recover BiMU. BiMU differs not only by the bounded-memory objective, but also by the bounded uncertainty-dependent metaplastic step size, which changes how evidence is consolidated or deconsolidated online. Our appendix (Table 11) is consistent with this: when N is made very large, BiMU approaches cumulative learning, yet it still remains clearly above BayesBiNN on 100-task Permuted-MNIST (83.70% vs 67.91% last-5 accuracy, OOD AUC 0.94 vs 0.75, MMRR 13.13 vs 5.03). We will clarify this distinction better in the revision.
>
> ### Q2 Model stability / previously learned tasks
>
> We agree that standard stability metrics are useful, and, as noted in our response to Reviewer 2, we have now computed BWT and will add it in the revision rather than relying only on MMRR and late-stream accuracy. We chose those two metrics on 1000-task Permuted-MNIST because the central failure mode there is progressive rigidity rather than standard task-to-task forgetting. That said, OpenLORIS already gives a stability-oriented view: the metric in Table 2 is the mean accuracy averaged over all 12 task test sets after training on all tasks, so it already reflects retention on previously learned tasks rather than only tail performance. We will make this much more explicit in the revised text.
>
> ### Q3 Computational efficiency and the role of K
>
> On a GPU, computation cost is relatively independent of K, as computations of the different samples can be parallelized. However, BiMU targets deployment in edge contexts on microcontroller/TinyML platforms. There, computation is approximately proportional to K, as we see in the measurement presented to reviewer 2, Q2. However, Appendix K shows that small K already recovers most of the benefit: on OpenLORIS, K=2 reaches 89.30% with 3.30% of queried samples, K=3 reaches 90.61% with 3.87%, and K=10 reaches 90.91% with 3.97%; only at K=25 do we gain further accuracy (91.74%) at a higher query fraction (4.81-5.63%). Thus low-K regimes already provide a strong accuracy/budget trade-off. Moreover, only queried samples trigger backpropagation and parameter updates; in the main OpenLORIS setting, BiMU reaches 88.70% with only 3.1% updates and 90.91% with 4.0% updates, corresponding to 32x and 25x fewer gradient updates than the 100% update baseline. Since backward/update steps dominate on-device training cost (see Reviewer 2, Q3), this keeps the overall method practical even when disagreement is estimated from multiple forward samples.
>
> ### Sensitivity to the memory window N
>
> We agree that N is important, but our ablation suggests structured regimes rather than brittle tuning. Appendix F shows the expected transition very clearly: N=100 leads to over-forgetting (27.09%); very large N leads back toward rigidity (N=100000: 83.70%, MMRR 13.13); and a broad intermediate range remains strong (N=700-1900 gives 86.55%-90.29% accuracy with 0.96-0.99 OOD AUC).  We therefore view N as an interpretable forgetting horizon, analogous to replay size or a discount factor, rather than a brittle proxy for total stream length. We will make this point clearer in the revised paper.

---

> > ### Author Rebuttal · Reviewer_i1Xr · 2026-04-02
> >
> > Thanks for your response. I maintain my score.

---

### Official Review · Reviewer_CAm1 · 2026-03-22

**Soundness:** 2
**Presentation:** 3
**Significance:** 3
**Originality:** 3
**Overall Recommendation:** 5
**Confidence:** 2

**Summary:**

This paper introduces a Bayesian continual-learning rule for Binary Bayesian Neural Network models. The problem in this case, as the authors describe, is that the model struggles with long-horizon learning. The paper introduces BiMU (Binary Metaplasticity from Uncertainty), which uses a bounded-memory Bayesian learning and forgetting objective that removes contributions of the oldest batches (outside a predefined window $N$) to better handle long-horizon scenarios. Additionally, it incorporates active learning by rejecting incoming samples that are not useful for learning, using a disagreement score. This also reduces label queries and backpropagation updates, making the (continual) learning process more efficient.

**Compliance With Llm Reviewing Policy:**

Affirmed.

**Final Justification:**

Thanks, all of my concerns are adequately addressed, and I will raise my score.

**Key Questions For Authors:**

__Q1__. The Bayesian learning and forgetting framework appears to rely on a strong assumption: the forgetting horizon is determined entirely by the window size rather than by the retrospective importance of individual data points. While the method does employ a principled prospective mechanism (the variance-ratio threshold), could the authors discuss this limitation?

__Q2__. Relatedly, the evaluation reports MMRR and late-stream accuracy but leaves out backward transfer (BWT) measurements. Since BWT would show whether the fixed-window forgetting drops important past knowledge too aggressively, why was it not included?

__Q3__. The paper claims MC forward passes are cheap and that backpropagation dominates on-device cost. Could the authors quantify by how much, and whether the update reduction translates to comparable real speedups compared to binary baselines?

**Limitations:**

yes

**Strengths And Weaknesses:**

Strength:

- The paper is very well written and clearly describes the problem it aims to solve. The proposed method is sound and addresses well-selected problems for the given task.
- The paper also evaluates on an imbalanced class regime, which further highlights the strength of the proposed method.
- The paper provides intuitive explanations and a principled update rule derivation for each continual learning concepts such as plasticity, stability, and forgetting. Each concept is well motivated and the theoretical contributions and design choices are easy to follow.

Weakness:

- The paper lacks a runtime performance discussion and comparison, which is a particularly important point given that binary BNNs have strong applicability on edge devices.
- A metric and discussion on backward transfer (i.e., stability on older tasks) seems somewhat lacking.

---

> ### Author Rebuttal · Authors · 2026-03-30
>
> We thank the reviewer for their constructive comments and positive evaluation of the paper. Below, we address each question in detail and clarify the points raised.
>
> ## Q1
>
> This is a very good point. BiMU uses a fixed effective horizon N rather than retrospective per-example importance, and this is a deliberate trade-off aligned with the regime we target: strict online continual learning on edge devices, with no replay buffer, no revisiting of past samples, and constant training-state memory. A retrospective importance mechanism would require stored data or extra per-sample state, moving away from that regime. BiMU therefore combines bounded-memory forgetting at the parameter level with prospective selectivity at the data level through uncertainty-based querying. In many always-on edge streams, observations are redundant and conditions recur, so a bounded horizon is often a practical approximation. Appendix F shows that N is an interpretable control knob: too small leads to over-forgetting, too large to rigidity, and intermediate N gives the best trade-off. We agree that fixed-horizon forgetting is less suited to one-off events. Outside the present buffer-free regime, a natural extension would be to combine BiMU with a small replay buffer for high-uncertainty queried samples.
>
> ## Q2
>
> We agree that BWT is a useful standard metric. We emphasized late-stream accuracy and MMRR because the failure mode we target is progressive rigidity / posterior saturation, which BWT alone can misread on extremely long streams: a method that has largely stopped adapting can appear to have a favorable BWT simply because its performance remains uniformly poor.
>
> Permuted-MNIST
>
> | Method | BiMU | BayesBiNN | Syn. Meta. | STE |
> |---|---:|---:|---:|---:|
> | Late stream acc. | 90.3% | 41.12% | 10.27% | 29.35% |
> | MMRR | 139.7 | 2.04 | 1.64 | 9.32 |
> | BWT | -0.82 | -0.39 | -0.0009 | -0.57 |
>
> This is exactly what we observe on 1000-task Permuted-MNIST: Synaptic Metaplasticity has the least negative BWT, yet it is the weakest binary method at the end of the stream, with only 10.27% mean accuracy. BayesBiNN likewise has a milder BWT than BiMU,  but its late-stream accuracy and rigidity resilience are far worse. STE lies in between. So on this long-horizon benchmark, a less negative BWT can simply indicate that the model is no longer learning effectively.
>
> OpenLORIS-Object (25,088-features)
>
> | Method | BiMU | BayesBiNN | Syn. Meta. | STE |
> |---|---:|---:|---:|---:|
> | All task acc. | 90.62% | 89.37% | 86.72% | 83.79% |
> | BWT | -0.052 | -0.040 | -0.052 | -0.08 |
>
> On OpenLORIS-Object, BWT is more interpretable because the stream has only 12 tasks. In the setting, BiMU obtains BWT comparable to BayesBiNN and Synaptic Metaplasticity, while achieving higher mean accuracy (STE is worse both in BWT and accuracy). Thus, BiMU’s gain is not coming from sacrificing backward stability; it combines comparable BWT with substantially better final performance.
>
> We will add all these results to the revision.
>
> ## Q3
>
> Thank you for raising runtime; this is central to the regime we target. We performed post-submission measurements on an STM32 microcontroller, a representative MCU-class TinyML/edge platform. In this setting, binary MC inference is especially favorable: sampled forward passes map well to bit-level operations and compact memory access, so repeated MC passes remain practical, whereas a queried update still requires gradient estimation through the relaxed Bernoulli parameterization, backpropagation, and parameter/state writes, which benefit much less from binarization and dominate on-device training cost. Concretely, in the 8,192-feature OpenLORIS setting, one MC forward pass costs about 13.7 ms per predictor, while the backpropagation/update phase for a queried sample costs about 266 ms per predictor, so an update is about 19$\times$ more expensive than a forward pass.
>
> BiMU therefore uses the cheap part of Bayesian binary learning (MC disagreement from binary forward passes) to decide when to pay the expensive part (online updates). With $K=2$, which reaches 89.30% accuracy at a 3.30% query rate, the mean cost is about 45 ms per incoming sample, versus 559.4 ms/sample if every sample were updated, i.e. 12.4$\times$ more.
> When implemented on the STM32, BayesBiNN has similar per-step timing (13.4 ms forward and 249 ms update per predictor), but BiMU offers high end-to-end gain as active learning is much more effective.
>
> Synaptic Metaplasticity is cheaper per queried step (5.9 ms forward; 146 ms update), but its active learning is much less effective, yielding higher average compute at lower accuracy. Synaptic Metaplasticity averages 150 ms per incoming sample, versus 45 ms/sample for BiMU with K=2, while ending 1.3 accuracy points lower.
>
> We will add this analysis to the revision. It complements the memory picture already reported in Appendix E, where BiMU also has the smallest training-state overhead among the continual-learning baselines.

---

> > ### Author Rebuttal · Reviewer_CAm1 · 2026-04-01
> >
> > Thanks, all of my concerns are adequately addressed, and I will raise my score.

---

### Official Review · Reviewer_ucSK · 2026-03-23

**Soundness:** 4
**Presentation:** 3
**Significance:** 2
**Originality:** 1
**Overall Recommendation:** 5
**Confidence:** 2

**Summary:**

This work proposes a Bayesian online learning algorithm for binary neural networks in the continual learning setting.

The online continual setting means, first, that data is observed one at a time (online) and, second, that the learning task changes over time (continual).
In practice, the challenge is then to be efficient in prediction and learning, as well as to continue to be flexible and adapt to new data (distributions).
The proposed solution in this work is Bayesian learning of binary neural networks with "controlled forgetting" in combination with an active learning scheme to pick which (streamed) data to learn from.

Binary neural networks [1] are networks of which the weights are either 1 or -1, as opposed to the real-valued weights of typical neural networks.
In the Bayesian setting, as introduced in [2], the distribution over these weights is represented with a Bernoulli distribution, parameterized by λ.
To avoid convergence of the posterior - an issue in the continual setting - techniques from "metaplasticity from synaptic uncertainty" (MESU) [3] are applied to modify the posterior to be an N-sized window: include only the last N observations when computing the posterior.
To further avoid convergence, only a subset of the observed data is used to compute the posterior.
Whether a newly data point is included is decided based on the disagreement (variation ratio): the method samples $K$ models and outputs $y_k$ and counts how often the most popular class is predicted (n).
If the variation ratio $1 - n/K$ exceeds some threshold $τ$, then the label is queried and added to the data set.

This method is compared to several baselines, including BayesBiNN [2], MESU [3] and synaptic metaplasticity [4], on four benchmarks with various continual learning settings.

[1] Courbariaux, M., Bengio, Y., & David, J. P. (2015). Binaryconnect: Training
deep neural networks with binary weights during propagations. Advances in
neural information processing systems, 28.

[2] Meng, X., Bachmann, R., & Khan, M. E. (2020, November). Training binary
neural networks using the bayesian learning rule. In International conference
on machine learning (pp. 6852-6861). PMLR.

[3] Bonnet, D., Cottart, K., Hirtzlin, T., Januel, T., Dalgaty, T., Vianello,
E., & Querlioz, D. (2025). Bayesian continual learning and forgetting in neural
networks. Nature Communications, 16(1), 9614.

[4] Laborieux, Axel, Maxence Ernoult, Tifenn Hirtzlin, and Damien Querlioz.
"Synaptic metaplasticity in binarized neural networks." Nature communications
12, no. 1 (2021): 2549.

**Compliance With Llm Reviewing Policy:**

Affirmed.

**Final Justification:**

With a better understanding of the contributions thanks to the rebuttal, I will be incrementing the score. Though I will stick to my (low) confidence, due to not having checked the derivations and being unable to really gauge their novelty/impact.

**Key Questions For Authors:**

- This method should be able to detect covariate shift - when the distribution of x changes - based on disagreement indicating that a sample is out out of distribution.
  If the task function itself shifts, however, I assume the variation ratio will not trigger learning.
  Is there some mechanism that will allow the proposed method to adapt in this situation?
  Specifically, will BiMU fail to adapt to new tasks if the shift is (only) in the labeling distribution $p(y | x)$?
- For those less familiar with the mean-Bernoulli and MESU's derivations, please highlight which parts of the propositions and theorems you argue will be the most impactful?
- Regarding the "performance per data used for training":
    - In the performance metrics in the graphs, the first recorded data point of BiMU is typically much "later" than others. Why?
    - Do you have any idea *when* the active learning scheme decides to query samples? (for example, we could hope it is uniformly spread out, assuming the shifts occur uniformly over time).
    - It would be great to have an understanding of "performance over time":
        - For continual learning, it would be interesting to see how quickly the model adapts to shifts.
        - For learning without shifts, it would be interesting to see *over time* how good BiMU performs: yes, it is more data efficient, but if this means that the method is terrible for a significant part of the task, then this seems like an issue.

**Limitations:**

As stated in a question above, I believe the active learning (and thus adaptability) of the proposed method relies on the model's confidence.
This clearly can capture distribution shifts if they occur in the covariates ($p(x)$ changes over time).
I believe BiMU would fail to adapt to shift in label distribution $p(y | x) (i.e. changes in $f(x) → y$).

**Strengths And Weaknesses:**

Overall, this work is readable and proposes a sound and effective online learning method for continual learning.
Weaknesses include the (perceived) lack of theoretical novelty: it is unclear to me how much is novel, and how impactful this novelty will be.

## Soundness

Though I did not have time to verify all derivations and claims, those that I did look at in more detail stood up to scrutiny.

## Presentation

The paper reads very well and is easy to follow: most of the derivations are hidden in the appendix, which simplified understanding the narrative, but made is challenging to verify or understand the math.

## Significance and Originality

As a reviewer, I found evaluating the contribution to be less obvious: it seems that both known and novel propositions/theorems/derivations are presented in a similar manner.
For example, proposition 1 (eq. 3) is well known, proposition 2 (eq. 4 and 5) exists almost verbatim in MESU [3] and the active learning mechanism seems common.
If I understand correctly, the specific derivations adapting MESU (which assumes Gaussian priors over neural networks parameters) to binary neural networks are a contribution, though I cannot evaluate their significance: the document points out parallels with "synaptic metaplasticity" [4], but that the derivation here emerges from Bayesian formulation (which is not the case in [4], presumably).

## Empirical Evaluation

The empirical evaluation takes up a significant amount of space and is well documented:

- it reports relevant success metrics such as mean accuracy and AUC on both binary and real-values neural network methods.
- it compares the proposed active learning components to sensible alternatives.
- it compares performance for differently-sized neural networks.
- it looks at how much of the data is used (queried), and evaluates on imbalanced data sets (with rare classes).

A downside is that little is said about parameter tuning: it is unclear to me exactly which parameters need to be picked, but at least the window size $N$, max learning rate $α_max$, and active learning threshold τ should be set.

---

> ### Author Rebuttal · Authors · 2026-03-30
>
> We thank the reviewer for their constructive comments and positive evaluation of the paper. Below, we address each question in detail and clarify the points raised.
>
>
> * **Clarification of novelty.**
>
>   Our contribution is a Bernoulli-specific continual-learning rule derived from the bounded-memory variational objective: the BiMU update in Eq. (6) together with the bounded metaplastic step size in Eq. (7). These two ingredients prevent mean-field Bernoulli posterior saturation and preserve informative posterior-sample disagreement, which is what makes fully online, buffer-free active querying possible in binary networks.
>
>
>   Eq. (3) is standard, Eq. (4) adopts the bounded-memory principle of Bonnet et al. The new step begins at Eq. (5): we specialize that objective to mean-field Bernoulli synapses. The new derived results are the BiMU update in Eq. (6) and the bounded metaplastic step size in Eq. (7) (Theorem A.4, Prop. A.5). This is not a verbatim transfer of MESU to binary weights: in the Bernoulli setting, both the controlled-forgetting dynamics and the bounded metaplastic step size require a distinct derivation. Their effect is twofold: the relaxation term scales as $\cosh^{-2}(\lambda)$, counteracting runaway growth of $|\lambda|$ and preventing saturation; and the bounded state- and gradient-dependent step size induces a consolidation/de-consolidation asymmetry that stabilizes consolidated synapses without freezing uncertain ones. This also clarifies the relation to Synaptic Metaplasticity: a similar asymmetry appears there empirically, whereas in BiMU it follows directly from the Bernoulli Bayesian objective.
>
>   Practically, preserving a non-degenerate Bernoulli posterior keeps MC disagreement usable long enough to enable fully online, buffer-free active querying in binary networks.  This is especially relevant at the edge, where Monte Carlo forward passes are cheap while label requests and backpropagation dominate the cost.  On OpenLORIS-Object under class imbalance and feature compression, this yields up to $32\times$ fewer labels/updates at matched accuracy.
>
>   We will revise the paper to mark Eqs. (3)-(4) as background and Eqs. (6)-(7) as the core novelty.
>
>
> * **Regarding label shift.**
>
>   This is a very interesting comment. Our experiments mainly study covariate or task shifts, where the presented inputs themselves change, and uncertainty-based scores can react, without task-boundary information. In a situation where the inputs remain familiar but the mapping from x to y changes, unlabeled criteria such as VR, predictive uncertainty, aleatoric uncertainty, and epistemic uncertainty can indeed miss the shift, because they are computed before observing the label. However, the last criterion that we proposed in the paper, VR-True, does not have this limitation as a diagnostic, since it explicitly compares sampled predictions to the true label. This is therefore the criterion we would use in this situation, at the cost of requiring labels for each training sample.
>
>    We will clarify this point in the final version of the paper.
>
>
> * **Regarding the "performance per data used for training".**
>
>   On the “performance per data used” plots, the x-axis is queried-label fraction, not time. The first visible BiMU point can therefore appear later simply because it is the smallest non-zero budget reached when sweeping the threshold. For VR, this is amplified by discretization: with K=10 predictors, VR only takes values in {0,0.1,...,0.9}, so the smallest non-zero budget can already be a few percent. This is a threshold-grid effect rather than delayed adaptation.
>
>   We very much appreciate your question about when queries occur. In an additional post-submission OpenLORIS analysis, we divided each of the 12 tasks into four equal segments and measured where VR-triggered updates occur. 45.9% of all updates happen in the first quarter of a task, immediately after the nuisance-factor shift, while 12.6% still occur in the last quarter. This is consistent with the intended behavior: querying spikes when the distribution changes, then decreases, but it does not collapse to zero. We view this as evidence that BiMU avoids posterior collapse and remains able to query informative samples after the initial adaptation phase. We will add this temporal analysis, together with accuracy-over-time plots around task transitions, in the revision.
>
>
> * **Hyperparameter protocol.**
>
>   We agree that this should be more visible. We use Optuna with 200 trials per method. On Permuted-MNIST, HPO is done on a separate 10-task validation stream; on OpenLORIS, on the validation split. For active learning, training hyperparameters are tuned once in a fully supervised streaming run, and all accuracy-budget curves are then obtained only by sweeping $\tau$, with no retuning per acquisition function or labeling budget. We will move this protocol to the main paper so the roles of the main hyperparameters are explicit.

---

> > ### Author Rebuttal · Reviewer_ucSK · 2026-04-01
> >
> > Thank you for answering my questions. With a better understanding of the contributions, I will be incrementing the score. Though I will stick to my (low) confidence, due to not having checked the derivations and being unable to really gauge their novelty/impact.

---

### Decision · Program_Chairs · 2026-04-30

**Decision:**

Accept (regular)

**Comment:**

The authors present a Bayesian continual learning approach for binary neural networks to address issues arising in long-horizon settings (posterior saturation). To this end, the authors developed an online update rule derived from a bounded-memory variational objective. Lastly, the method is evaluated on different OOD regimes, showing the superiority of the approach.

The reviews have generally been very favourable for this submission, though many reviewers expressed low confidence in their assessments. After having read the reviews and the paper myself, I believe that several aspects of the work would require additional scrutiny. Specifically, while the derivations of A.2, which lead to the development of the update rule, are conceptually correct, several of the assumptions made would require a stronger justification and ablations that verify the effect of those assumptions. Specifically, the assumption that the marginal is equivalent for past batches is likely violated; equally approximating the posterior of the previous batch with a variational approximation is problematic and known to result in non-negligible approximation error. The latter are expected to accumulate, and it is unclear to what extent those affect or deteriorate the final variational approximation. In addition, the work introduces various hyperparameters into the variational objective (see B.1) that are not principled. Hence, it is unclear to what extent the good performance is a side-effect of tuning those temperings rather than the actual change in the objective.